# How Does Label Noise Gradient Descent Improve Generalization in the Low SNR Regime?

## Abstract

The capacity of deep learning models is often large enough to both learn the underlying statistical signal and overfit to noise in the training set. This noise memorization can be harmful especially for data with a low signal-to-noise ratio (SNR), leading to poor generalization. Inspired by prior observations that label noise provides implicit regularization that improves generalization, in this work, we investigate whether introducing label noise to the gradient updates can enhance the test performance of neural network (NN) in the low SNR regime. Specifically, we consider the learning of a two-layer NN with a simple label noise gradient descent (GD) algorithm, in an idealized signal-noise data setting. We prove that adding label noise during training suppresses noise memorization, preventing it from dominating the learning process; consequently, label noise GD enjoys rapid signal growth while the overfitting remains controlled, thereby achieving good generalization despite the low SNR. In contrast, we also show that NN trained with standard GD tends to overfit to noise in the same low SNR setting and establish a non-vanishing lower bound on its test error, thus demonstrating the benefit of label noise injection in gradient-based training.

## 1 Introduction

The success of deep learning across various domains (LeCun et al., 2015; Silver et al., 2016; Brown, 2020) is often attributed to their ability to extract useful features (Girshick et al., 2014; Devlin, 2018) via gradient-based training (Damian et al., 2022; Ba et al., 2022). One desirable property of gradient-based feature learning is the algorithmic regularization that prioritizes learning of the underlying signal instead of overfitting to noise: real-world data contains noise due to mislabeling, data corruption, or inherent ambiguity, yet despite having the capacity to memorize noise, neural networks (NNs) trained by gradient descent (GD) tend to identify informative features and "low-complexity" solutions that generalize (Zhang et al., 2021; Rahaman et al., 2019). To understand this behavior, recent theoretical works considered data models that partition the features into signal and noise components (Ghorbani et al., 2020; Ben Arous et al., 2022; Wang et al., 2024), and studied the performance of gradient-based training under different signal-to-noise conditions.

Among existing theoretical settings, the signal-noise model proposed in Allen-Zhu & Li (2020); Cao et al. (2022) has been extensively studied in the feature learning theory literature. In this model, input features are constructed by combining a label-dependent *signal* with label-independent *noise*. The signal represents meaningful patterns relevant to the predictive task while the noise component captures background features unrelated to the learning task. This idealized setting has shed light on how various algorithms, neural network architectures, and other factors influence optimization and generalization of neural networks, depending on the signal-to-noise ratio (SNR) (Frei et al., 2022; Zou et al., 2023; Jelassi & Li, 2022; Huang et al., 2023; Xu et al., 2023; Chen et al., 2022).

In the signal-noise model, it is known that the SNR dictates a transition from *benign overfitting* to *harmful overfitting*. In the high SNR regime, gradient-based feature learning prioritizes signal learning over noise memorization; hence upon convergence, the trained NN recovers the signal and generalizes to unseen data despite some degree of noise memorization, a phenomenon known as benign overfitting (Bartlett et al., 2020; Tsigler & Bartlett, 2023; Li et al., 2023b; Sanyal et al., 2020; Shamir, 2023). In contrast, when the SNR is low, noise memorization dominates the training

dynamics, and the network fails to identify useful features before the training loss becomes small, leading to harmful overfitting (Cao et al., 2022; Kou et al., 2023b).

Motivated by this observation, recent works have explored algorithmic modifications that either enhance signal learning or suppress noise memorization, to improve generalization in the challenging low SNR regime. Huang et al. (2023) showed that the smoothing effect of graph convolution in graph neural networks mitigates overfitting to noise; however, this approach requires the graph to be sufficiently dense and exhibits high homophily. Chen et al. (2024) found that the sharpness-aware minimization (SAM) method (Foret et al., 2020) prevents noise memorization in early stages of training, thereby promoting effective feature learning; this being said, SAM has higher computational cost than standard GD due to the two forward and backward passes per step, and it involves more complex hyperparameter tuning. The goal of this work is to address the following question.

*Is there a simple modification of GD with no computational overhead that achieves small generalization error in low SNR settings where standard GD fails to generalize?*

### 1.1 OUR CONTRIBUTIONS

We provide an affirmative answer to the question above by introducing **random label noise** to the training dynamics as a form of regularization, inspired by the label noise stochastic gradient descent (SGD) method (Blanc et al., 2020; HaoChen et al., 2021; Shallue et al., 2019; Szegedy et al., 2016). Specifically, we consider the learning of a two-layer convolutional neural network in a binary classification problem studied in Cao et al. (2022), and show that by randomly flipping the labels of a small proportion of training samples at each iteration, noise memorization can be suppressed despite the low SNR, whereas signal learning experiences a period of fast growth. As a result, neural network trained by label noise GD attains good generalization performance in regimes where standard GD fails, as summarized in the following informal theorem:

**Theorem 1.1** (Informal). *Given $n$ training samples drawn from the distribution in Definition 2.1 in the low SNR regime where $n^{-1}\mathrm{SNR}^{-2} = \tilde{\Omega}(1)$. Then for any $\epsilon > 0$, after a polynomial number of training steps $t$ (depending on $\epsilon$), with high probability we have,*

- ***Standard GD*** *minimizes the logistic training loss to $L_S^{(t)} \leq \epsilon$, but the generalization error (0-1 loss) remains large, i.e., $L_D^{(t)} = \Omega(1)$.*

- ***Label noise GD*** *cannot reduce the logistic training loss to a small value $L_S^{(t)} = \Omega(1)$, but achieves small generalization error (0-1 loss), i.e., $L_D^{(t)} = o(1)$.*

We make the following remarks on our main results.

- **Improved Generalization due to Label Noise.** The theorem provides an upper bound on the test error of label noise GD, as well as a lower bound on the error of standard GD. This demonstrates that incorporating label noise into the gradient descent updates improves generalization in the low SNR regime. We note that our conditions on label noise GD learnability are weaker than those required for SAM as specified in Chen et al. (2024), even though our studied algorithm is arguably simpler and more computationally efficient – see Section 3 for more comparisons.

- **Analysis of Feature Learning Dynamics.** We establish the main theorem via a refined characterization of the training dynamics of label noise GD on a two-layer convolutional NN with squared ReLU activation. A key observation in our analysis is that label noise introduces regularization to the noise memorization process, preventing it from growing beyond a constant level; meanwhile, signal learning continues to exhibit a rapid growth rate, allowing the model to identify the informative features and avoid harmful overfitting in low SNR regimes.

### 1.2 ADDITIONAL RELATED WORKS

**Label Noise SGD.** Recent works have empirically shown that label noise stochastic gradient descent (SGD) through label smoothing exhibits favorable generalization properties due to the regularization effect of the injected noise (Shallue et al., 2019; Szegedy et al., 2016; Wen et al., 2019).

Furthermore it has been argued that label flipping approach adopted in this work can be cast as label smoothing methods (Li et al., 2020). From a theoretical standpoint, label noise SGD has been primarily explored in the context of linear regression or shallow neural networks, particularly in regression settings (Blanc et al., 2020; Damian et al., 2021; HaoChen et al., 2021; Huh & Rebeschini, 2024; Li et al., 2021; Vivien et al., 2022; Takakura & Suzuki, 2024); these studies have highlighted the implicit regularization benefits of label noise in SGD. For instance, Takakura & Suzuki (2024) illustrated the implicit regularization of label noise in mean-field neural networks, while Li et al. (2021); Damian et al. (2021) proved that label noise introduces bias towards flat minima. In contrast to these existing literature, our work focuses on the binary classification setting specified by the signal-noise model, providing a quantitative analysis of the training dynamics and the generalization benefits of label noise GD in the low SNR regime.

**Signal-Noise Data Models.** Recent theoretical works have studied the signal-noise model in various contexts, including $(i)$ *optimization algorithms*, such as Adam (Zou et al., 2021), momentum (Jelassi & Li, 2022), sharpness-aware minimization (Chen et al., 2023), large learning rates (Lu et al., 2023); $(ii)$ *learning paradigms*, such as ensembling and knowledge distillation (Allen-Zhu & Li, 2020), semi-and self-supervised learning (Kou et al., 2023a; Wen & Li, 2021), Mixup (Zou et al., 2023; Chidambaram et al., 2023), adversarial training (Allen-Zhu & Li, 2022), and prompt tuning (Oymak et al., 2023); and $(iii)$ *neural network structures*, such as convolutional neural network (Cao et al., 2022; Kou et al., 2023b), vision transformer (Jelassi et al., 2022; Li et al., 2023a), graph neural network (Huang et al., 2023; Li et al., 2024). Our work is in line with Chen et al. (2022); Huang et al. (2023), with the goal of showing that a simple algorithmic modification (label noise GD) facilitates feature learning in the challenging low SNR regime.

## 1.3 NOTATION

We use bold-faced letters for vectors and matrices. For a vector $v$, its $\ell_2$-norm is denoted as $\|v\|_2$. For a matrix $A$, we use $\|A\|_2$ to denote its spectral norm and $\|A\|_F$ its Frobenius norm. We employ standard asymptotic notations $O(\cdot), o(\cdot), \Omega(\cdot),$ and $\Theta(\cdot)$ to track the limiting scaling, and $\widetilde{O}(\cdot), \widetilde{\Omega}(\cdot),$ and $\widetilde{\Theta}(\cdot)$ to hide polylogarithmic factors. We denote $[n] = 1, 2, \ldots, n$, and $[a, b] = a, a + 1, \ldots, b$, where $b \geq a$, $a \neq 1$, and $a, b \in \mathbb{N}$.

## 2 PROBLEM SETUP

In this section, we describe the signal-noise data model, the neural network architecture used for training, and the label noise gradient descent algorithm considered in this work.

**Data generating process.** We consider the signal-noise data model from Cao et al. (2022); Chen et al. (2023). Let $\mu \in \mathbb{R}^d$ be a fixed signal vector, and for each data point $(x, y)$, the feature $x$ is composed of two patches, denoted as $x = \{x^{(1)}, x^{(2)}\} \in \mathbb{R}^{2d}$. The target variable $y$ is a binary label, taking values in $\{\pm 1\}$. Then the data is generated according to the following process.

**Definition 2.1.** *We consider the following generating process for $(x, y)$,*

1. *The true label $y$ is drawn from a Rademacher distribution, i.e., $\mathbb{P}[y = 1] = \mathbb{P}[y = -1] = 1/2$.*

2. *One of the patches, $x^{(1)}$ or $x^{(2)}$ is randomly selected to be $y\mu$ (representing the signal), while the other is set to be $\xi \sim \mathcal{N}(0, \sigma_p^2(I_d - \mu\mu^\top \|\mu\|_2^{-2}))$ (representing the noise). Here, $\sigma_p^2$ denotes the strength of the noise vector.*

We make the following remarks on the data distribution.

- The data model simulates a setting where the input features are composed of both signal and noise components. Specifically, each data point is divided into two patches, and one of these patches contains meaningful information (signal) related to the classification label, while the other patch only contains random noise that is independent of the label. The noise covariance $\sigma_p^2(I_d - \mu\mu^\top \|\mu\|_2^{-2})$ is set to ensure that the noise vector is orthogonal to the signal vector for simplicity.

- This setup is designed to reflect real-world scenarios where data contains a mix of relevant and irrelevant features (see (Allen-Zhu & Li, 2020, Appendix A) for discussions). Note that in high dimensions ($n \ll d$), the NN can achieve small training loss just by overfitting to the noise component. Therefore, the challenge for the learning algorithm in the low SNR regime is to identify and learn the signal patch while ignoring the noisy patch.

- We use the minimum number of patches in the multi-patch model for concise presentation. Our results can be extended to more general cases where the number of patches is greater than 2; see Allen-Zhu & Li (2020); Shen et al. (2022) for such extension.

**Neural network and loss function.**   Following Cao et al. (2022), we consider a two-layer convolutional neural network with squared ReLU activation and shared filters applied separately to each patch. The network is defined as $f(\boldsymbol{W}, \boldsymbol{x}) = F_{+1}(\boldsymbol{W}_{+1}, \boldsymbol{x}) - F_{-1}(\boldsymbol{W}_{-1}, \boldsymbol{x})$, where

$$F_j(\boldsymbol{W}_j, \boldsymbol{x}) = \frac{1}{m} \sum_{r=1}^{m} \sum_{p=1}^{2} \sigma(\langle \boldsymbol{w}_{j,r}, \boldsymbol{x}^{(p)} \rangle) = \frac{1}{m} \sum_{r=1}^{m} \Big( \sigma(\langle \boldsymbol{w}_{j,r}, y\boldsymbol{\mu} \rangle) + \sigma(\langle \boldsymbol{w}_{j,r}, \boldsymbol{\xi} \rangle) \Big),$$

in which $m$ denotes the size of the hidden layer, and $\sigma(z) = (\max\{0, z\})^2$. Note that $j \in \{-1, +1\}$ corresponds to the fixed second-layer. The symbol $\boldsymbol{W}_j$ represents the collection of weight vectors in the first layer, i.e.,

$$\boldsymbol{W}_j = [\boldsymbol{w}_{j,1}, \boldsymbol{w}_{j,2}, \dots, \boldsymbol{w}_{j,m}] \in \mathbb{R}^{d \times m},$$

where $\boldsymbol{w}_{j,r} \in \mathbb{R}^d$ is the weight vector of the $r$-th neuron. Here, $j \in \{-1, +1\}$ indicates the fixed value in the second layer. The initial weights $\boldsymbol{W}_{\pm 1}$ has entries sampled from $\mathcal{N}(0, \sigma_0^2)$.

**Remark 2.1.** *Since we do not optimize the 2nd-layer parameters, we expect the 2-homogeneous squared ReLU activation to mimic the behavior of training both layers simultaneously in a ReLU network; such higher-order homogeneity amplifies feature learning (e.g., see (Chizat & Bach, 2020; Glasgow, 2023)) and creates more significant gap between signal learning and noise memorization. A similar effect can be achieved by a smoothed ReLU activation with local polynomial growth as in Allen-Zhu & Li (2020); Shen et al. (2022).*

We use the logistic loss computed over $n$ training samples, denoted as $S = \{(\boldsymbol{x}_i, y_i)\}_{i \in [n]}$:

$$L_S(\boldsymbol{W}) = \frac{1}{n} \sum_{i \in [n]} \ell(y_i f(\boldsymbol{W}, \boldsymbol{x}_i)), \quad \text{where } \ell(z) = \log(1 + \exp(-z)).$$

To evaluate the generalization performance of the trained network, we measure its expected 0-1 loss on unseen data, defined as

$$L_{\mathcal{D}}^{0-1}(\boldsymbol{W}) = \mathbb{E}_{(\boldsymbol{x}, y) \sim \mathcal{D}}[\mathbb{1}(y \neq \text{sign}(f(\boldsymbol{W}, \boldsymbol{x})))], \tag{1}$$

where $\mathcal{D}$ denotes the data distribution specified in Definition 2.1, and $\mathbb{1}(\cdot)$ is the indicator function.

**Label noise GD for binary classification.**   We train the above neural network by gradient descent on either (i) the original loss function (standard GD), or (ii) the loss function with label-flipping noise defined as

$$L_S^{\epsilon}(\boldsymbol{W}^{(t)}) \triangleq \frac{1}{n} \sum_{i \in [n]} \ell(\epsilon_i^{(t)} y_i f(\boldsymbol{W}^{(t)}, \boldsymbol{x}_i)).$$

Here $\epsilon_i^{(t)}$ is a random variable that takes value 1 with probability $1 - p$ and $-1$ with probability $p$, represented by $\epsilon_i^{(t)} \sim \text{Rademacher}(1 - p, p)$. In other words, the sign of the labels is flipped with probability $p$ independently at each step. The generalization benefit of this label-flipping strategy has been studied both theoretically (Damian et al., 2021) and empirically (Xie et al., 2016; HaoChen et al., 2021) as an extension of label noise GD to classification settings.

The label noise GD update is then given as follows:

$$\boldsymbol{w}_{j,r}^{(t+1)} = \boldsymbol{w}_{j,r}^{(t)} - \frac{\eta}{nm} \sum_{i=1}^{n} \tilde{\ell}_i^{'(t)} \sigma'(\langle \boldsymbol{w}_{j,r}^{(t)}, y_i \boldsymbol{\mu} \rangle) \epsilon_i^{(t)} j\boldsymbol{\mu} - \frac{\eta}{nm} \sum_{i=1}^{n} \tilde{\ell}_i^{'(t)} \sigma'(\langle \boldsymbol{w}_{j,r}^{(t)}, \boldsymbol{\xi}_i \rangle) \epsilon_i^{(t)} y_i j\boldsymbol{\xi}_i, \tag{2}$$

---

**Algorithm 1** Label noise gradient descent

---

1: Initialize $\boldsymbol{W}_0$, step size $\eta$, flipping probability $p \in [0, 1]$
2: **for** $t = 0, ..., T - 1$ **do**
3:     Sample $\epsilon_i^{(t)} \sim \text{Rademacher}(1 - p, p)$, $\forall i \in [n]$.
4:     $\boldsymbol{W}^{(t+1)} = \boldsymbol{W}^{(t)} - \eta \nabla_{\boldsymbol{W}} L_S^\epsilon(\boldsymbol{W}^{(t)})$, where $L_S^\epsilon(\boldsymbol{W}^{(t)}) = \frac{1}{n} \sum_{i \in [n]} \ell\big(\epsilon_i^{(t)} y_i f(\boldsymbol{W}^{(t)}, \boldsymbol{x}_i)\big)$.
5: **end for**

---

where $\eta$ is the learning rate, and we defined $\tilde{\ell}_i^{'(t)} = \ell'(\epsilon_i^{(t)} y_i f(\boldsymbol{W}^{(t)}, \boldsymbol{x}_i))$ as the derivative of the loss function. This *label noise GD* training procedure is outlined in Algorithm 1. Observe that the proposed algorithm is *computationally efficient*, as the introduced label noise does not modify the original gradient descent framework. Hence this method is simple to implement, does not add significant computational overhead, and requires no complex hyperparameter tuning.

## 3 MAIN RESULTS

In this section, we quantify the benefits of label noise gradient descent by comparing its generalization performance against standard gradient descent (GD) training without label noise. We begin by outlining the assumptions that apply to both label noise GD and standard GD.

**Assumption 3.1.** *Define* $\text{SNR} = \frac{\|\boldsymbol{\mu}\|_2}{\sigma_p \sqrt{d}}$. *We consider the following setting for both algorithms:*

(i) *data dimension* $d = \tilde{\Omega}(\max\{n^2, n\|\boldsymbol{\mu}\|_2^2/\sigma_p^2\})$; *signal-to-noise ratio* $\text{SNR} = \tilde{O}(1/\sqrt{n})$.

(ii) *network width* $m = \tilde{\Omega}(1)$; *number of training samples* $n = \tilde{\Omega}(1)$.

(iii) *learning rate* $\eta \leq \tilde{O}(\sigma_p^{-2} d^{-1})$.

(iv) *initialization variance* $\tilde{O}(n \sigma_p^{-1} d^{-3/4}) \leq \sigma_0 \leq \tilde{O}(\min\{\|\boldsymbol{\mu}\|_2^{-1} d^{-5/8}, \sigma_p^{-1} d^{-1/2}\})$.

(v) *flipping rate of label noise* $0 < p < 1/C$, *where* $C$ *is a sufficient large constant.*

We make the following remarks on the above assumption.

- The high-dimensional assumption $(i)$ is standard in the benign overfitting analysis of NNs (e.g., see Cao et al. (2022); Frei et al. (2022)). The low SNR condition is derived from the comparison between the magnitude of signal learning and noise memorization – see Section 4.1; similar conditions has been established in Cao et al. (2022); Kou et al. (2023b) for different activations.

- The requirements on the hidden layer size $m$ and the sample size $n$ being at least polylogarithmic in the dimension $d$ ensure that certain statistical properties regarding weight initialization and the training data hold with high probability at least $1 - 1/d$.

- The upper bound on the learning rate $\eta$ ensures that the iterates in (4-6) remain bounded, which is required for standard GD to reach low training loss; see Proposition 4.1 for details.

- The upper bound on the initialization scale $\sigma_0$ is used to ensure convergence of GD, whereas the lower bound is used for anti-concentration upon initialization. Similar requirements on $\sigma_0$ can be found in (Cao et al., 2022, Condition 4.2).

- The upper bound on label flipping rate $p$ prevents the label noise from dominating the true signal.

We first state the negative result for standard gradient descent (GD) without label noise.

**Theorem 3.1** (GD fails to generalize under low SNR). *Under Assumption 3.1, for any $\epsilon > 0$, there exists $t = \Theta\big(\frac{nm \log(1/(\sigma_0 \sigma_p \sqrt{d}))}{\eta \sigma_p^2 d} + \frac{m^3 n}{\eta \epsilon \sigma_p^2 d}\big)$, such that with probability at least $1 - d^{-1/4}$, it holds that*

- *The training error converges, i.e.,* $L_S(\boldsymbol{W}^{(t)}) \leq \epsilon$.

- *The test error is large, i.e.,* $L_{\mathcal{D}}(\boldsymbol{W}^{(t)}) \geq 0.24$.

Theorem 3.1 indicates that even though standard GD can minimize the training error to an arbitrarily small value, the generalization performance remains poor. This is mainly because the neural network overfits to the noise components in the input data instead of learning the useful features.

Next, we present the positive result for label noise gradient descent.

**Theorem 3.2** (Label Noise GD generalizes under low SNR). *Under Assumption 3.1, there exists* $t = \Theta\left(\frac{nm \log(1/(\sigma_0 \sigma_p \sqrt{d}))}{\eta \sigma_p^2 d} + \frac{m \log(6/(\sigma_0 \|\boldsymbol{\mu}\|_2))}{\eta \|\boldsymbol{\mu}\|_2^2}\right)$ *and constants* $C > 0$, *such that with probability at least* $1 - d^{-1/4}$, *it holds that*

- *The training error is at constant order, i.e.,* $L_S(\boldsymbol{W}^{(t)}) = \Theta(1)$.

- *The test error is small, i.e.,* $L_{\mathcal{D}}(\boldsymbol{W}^{(t)}) \leq 2\exp\left(-\frac{Cd}{n^2}\right)$.

Theorem 3.2 shows that label noise GD achieves vanishing generalization error when the input dimensionality is large (i.e., $d = \Omega(n^2)$) despite the low SNR.

**Remark 3.1.** *Theorems 3.2 and 3.1 present contrasting outcomes for standard GD and label noise GD in the low SNR regime. In particular,*

- *Standard GD minimizes the training error effectively but does so by primarily overfitting to noise in the training data. This significant noise memorization leads to harmful overfitting.*

- *In contrast, label noise GD introduces a regularization effect through label noise, which prevents the network from fully memorizing the noise components. This allows the network to focus on learning the true signal, resulting in a phase of accelerated signal learning. Consequently, the model generalizes even though the training loss does not vanish (due to noise injection).*

**Comparison with sharpness-aware minimization (Chen et al., 2023).** We briefly discuss the differences between our findings and those in Chen et al. (2023) for the sharpness-aware minimization (SAM) method, where the authors established conditions on the SNR under which SAM can generalize better than stochastic gradient descent (SGD). However, their analysis requires the additional condition that the signal norm satisfies $\|\boldsymbol{\mu}\|_2 \geq \tilde{\Omega}(1)$, indicating the necessity of a sufficiently strong signal. In contrast, we show that label noise GD enjoys good generalization without this strong signal condition. This highlights the robustness of label noise GD in low SNR regimes (even when the signal strength is considerably weaker compared to the noise).

## 4 PROOF SKETCH

In this section, we give an overview of of our analysis of the optimization dynamics of standard GD and label noise GD . Our key technical contributions are summarized as follows:

- **Boundary characterization in low SNR regimes.** Unlike previous studies Cao et al. (2022); Kou et al. (2023b); Chen et al. (2024) that focus on the higher polynomial or standard ReLU activation, we analyze the 2-homogeneous squared ReLU activation, leading to a different boundary characterization of the low SNR regime for standard GD – see Section 4.2.

- **Upper bound via supermartingale.** We introduce a novel application of supermartingale arguments combined with Azuma's inequality to analyze the boundedness of noise memorization for label noise GD. This probabilistic approach provides high-probability guarantees on the training dynamics that were not previously established in this context.

### 4.1 SIGNAL-NOISE DECOMPOSITION

To analyze the training dynamics, we adopt a parameter decomposition technique from (Cao et al., 2022; Kou et al., 2023b): there exist $\{\gamma_{j,r}^{(t)}\}$ and $\{\rho_{j,r,i}^{(t)}\}$ such that

$$\boldsymbol{w}_{j,r}^{(t)} = \boldsymbol{w}_{j,r}^{(0)} + j\gamma_{j,r}^{(t)}\|\boldsymbol{\mu}\|_2^{-2}\boldsymbol{\mu} + \sum_{i=1}^{n} \rho_{j,r,i}^{(t)}\|\boldsymbol{\xi}_i\|_2^{-2}\boldsymbol{\xi}_i. \tag{3}$$

This decomposition originates from the observation that the gradient descent update always evolves in the direction of $\boldsymbol{\mu}$ and $\boldsymbol{\xi}_i$ for $i \in [n]$. In particular, $\gamma_{j,r}^{(t)} \approx \langle \boldsymbol{w}_{j,r}^{(t)}, \boldsymbol{\mu} \rangle$ serves as the *signal learning* coefficient, whereas $\rho_{j,r,i}^{(t)} \approx \langle \boldsymbol{w}_{j,r}^{(t)}, \boldsymbol{\xi}_i \rangle$ characterizes the *noise memorization* during training.

Next we let $\overline{\rho}_{j,r,i}^{(t)} = \rho_{j,r,i}^{(t)} \mathbb{1}(y_i = j)$ and $\underline{\rho}_{j,r,i}^{(t)} = \rho_{j,r,i}^{(t)} \mathbb{1}(y_i = -j)$. Combined with the gradient descent update given by Equation (2), we obtain the iteration rules for these coefficients:

$$\gamma_{j,r}^{(t+1)} = \gamma_{j,r}^{(t)} - \frac{\eta}{nm} \sum_{i=1}^{n} \tilde{\ell}_i^{'(t)} \sigma'(\langle \boldsymbol{w}_{j,r}^{(t)}, y_i \boldsymbol{\mu} \rangle) \|\boldsymbol{\mu}\|_2^2 \epsilon_i^{(t)}, \tag{4}$$

$$\overline{\rho}_{j,r,i}^{(t+1)} = \overline{\rho}_{j,r,i}^{(t)} - \frac{\eta}{nm} \tilde{\ell}_i^{'(t)} \sigma'(\langle \boldsymbol{w}_{j,r}^{(t)}, \boldsymbol{\xi}_i \rangle) \|\boldsymbol{\xi}_i\|_2^2 \epsilon_i^{(t)} \mathbb{1}(y_i = j), \tag{5}$$

$$\underline{\rho}_{j,r,i}^{(t+1)} = \underline{\rho}_{j,r,i}^{(t)} + \frac{\eta}{nm} \tilde{\ell}_i^{'(t)} \sigma'(\langle \boldsymbol{w}_{j,r}^{(t)}, \boldsymbol{\xi}_i \rangle) \|\boldsymbol{\xi}_i\|_2^2 \epsilon_i^{(t)} \mathbb{1}(y_i = -j). \tag{6}$$

where the initial values of the coefficients are given by $\gamma_{j,r}^{(0)} = 0$ and $\rho_{j,r,i}^{(0)} = 0$ for all $i \in [n]$, $j \in \{-1, 1\}$ and $r \in [m]$.

To analyze the optimization trajectory, we track the dynamics of signal learning coefficients $(\gamma_{j,r}^{(t)})$ and noise memorization coefficients $(\rho_{j,r,i}^{(t)})$ using the iteration rules in Equations (4-6). To facilitate a detailed analysis, we first provide upper bounds on the absolute value of both the signal learning and noise memorization coefficients throughout the entire training process.

**Proposition 4.1.** *Given Assumption 3.1 and $\epsilon > 0$. Let $\beta = 2 \max_{j,r,i} \{|\langle \boldsymbol{w}_{j,r}^{(0)}, \boldsymbol{\mu} \rangle|, |\langle \boldsymbol{w}_{j,r}^{(0)}, \boldsymbol{\xi}_i \rangle|\}$ and $\alpha = 4 \log(T^*)$. For $0 \le t \le T^*$, where $T^* = \eta^{-1} \mathrm{poly}(n, m, d, \|\boldsymbol{\mu}\|_2^{-1}, (\sigma_p^2 d)^{-1}, \sigma_0^{-1}, \epsilon^{-1})$, for all $i \in [n]$, $r \in [m]$ and $j \in \{-1, 1\}$, it holds that*

$$0 \le \gamma_{j,r}^{(t)} \le \alpha, \quad 0 \le \overline{\rho}_{j,r,i}^{(t)} \le \alpha, \tag{7}$$

$$0 \ge \underline{\rho}_{j,r,i}^{(t)} \ge -\beta - 16\sqrt{\frac{\log(4n^2/\delta)}{d}} n\alpha \ge -\alpha. \tag{8}$$

The proof can be found in the Appendix B. Proposition 4.1 indicates that during the entire training stage, there is a logarithmic upper bound on the absolute values of both the signal learning and noise memorization coefficients. This result is crucial for a detailed stage-wise characterization of the training dynamics. Note that the upper bound provided in this proposition holds for both standard GD and label noise GD.

## 4.2 PROOF SKETCH FOR THEOREM 3.1

We first establish the negative result for standard GD based on a two-stage analysis. As previously mentioned, we consider the 2-homogeneous $\sigma(z) = \mathrm{ReLU}^2(z)$ which differs from Cao et al. (2022); Kou et al. (2023b); Chen et al. (2023). This leads to a key difference in the boundary characterization of the low SNR regime.

**First stage.** Notice that starting from small initialization, the loss derivative remains close to a constant. Based on this observation, we establish the difference in magnitude between the coefficients of signal learning and noise memorization.

According to the update rule for the signal learning coefficient given by Equation (4) and by setting $\epsilon_i^{(t)} = 1$ for all $t$ and $i \in [n]$ (i.e., no label flipping), the upper bound of signal learning can be achieved as $\gamma_{j,r}^{(t)} + |\langle \boldsymbol{w}_{j,r}^{(0)}, \boldsymbol{\mu} \rangle| \le \exp\left(\frac{2\eta\|\boldsymbol{\mu}\|_2^2}{m}t\right)|\langle \boldsymbol{w}_{j,r}^{(0)}, \boldsymbol{\mu} \rangle|$. Meanwhile, the bounds for the noise memorization coefficients can be derived from the update rules (5) and (6). The results are given as $\max_{j,r} |\underline{\rho}_{j,r,i}^{(t)}| \le \frac{3\eta\sigma_p^2 td}{nm} \sqrt{\log(8mn/\delta)} \sigma_0 \sigma_p \sqrt{d}$, and $\max_{j,r} \overline{\rho}_{j,r,i}^{(t)} \ge \exp\left(\frac{\eta C_1 \sigma_p^2 d}{2nm}t\right) \sigma_0 \sigma_p \sqrt{d}/4 - 0.6\overline{\beta}$, for all $i \in [n]$, where we define $\overline{\beta} = \min_{i \in [n]} \max_{r \in [m]} \langle \boldsymbol{w}_{y_i,r}^{(0)}, \boldsymbol{\xi}_i \rangle$, and use $|\tilde{\ell}_i^{'(t)}| \ge C_1$. In the low SNR setting, where $\sigma_p \sqrt{d}$ is much larger than $\|\boldsymbol{\mu}\|_2$, we observe that noise memorization dominates the feature learning process during the first stage, as shown in the following lemma.

**Lemma 4.1.** *Under the same condition as Theorem 3.1, and let $T_1 = \Theta(\frac{nm \log(1/(\sigma_0 \sigma_p \sqrt{d}))}{\eta \sigma_p^2 d})$, the following results hold with high probability at least $1 - d^{-1}$: (i) $\max_{j,r} \overline{\rho}_{j,r,i}^{(T_1)} \geq 1$, for all $i \in [n]$; (ii) $\max_{j,r,i} |\underline{\rho}_{j,r}^{(t)}| \leq \tilde{O}(\sigma_0 \sigma_p \sqrt{d})$, for all $t \in [T_1]$; (iii) $\max_{j,r} \gamma_{j,r}^{(t)} \leq \tilde{O}(\sigma_0 \|\boldsymbol{\mu}\|_2)$, for all $t \in [T_1]$.*

Lemma 4.1 indicates that when the SNR is sufficiently low, i.e., $\text{SNR} = \tilde{O}(1/\sqrt{n})$, noise memorization dominates the training dynamics during the early phase of standard GD optimization. We highlight that this "low-SNR" condition differs from that of Cao et al. (2022); Kou et al. (2023b) due to the choice of activation function. In particular, Cao et al. (2022) assumed $\sigma(z) = (\max\{0, z\})^q$ with $q > 2$ and established a low-SNR boundary $n^{-1}\text{SNR}^{-q} = \tilde{\Omega}(1)$, whereas Kou et al. (2023b) considered the ReLU activation and derived the condition $n\frac{\|\boldsymbol{\mu}\|_2^4}{\sigma_p^4 d} \leq O(1)$.

**Second stage.** After the first stage, the loss derivative is no longer bounded by a constant value. To prove convergence of the training loss $L(t) \leq \epsilon$, we build upon the analysis from the first stage and define $\boldsymbol{w}_{j,r}^* = \boldsymbol{w}_{j,r}^{(0)} + 2m \log(2/\epsilon) \sum_{i=1}^n \|\boldsymbol{\xi}_i\|_2^{-2} \boldsymbol{\xi}_i$. We show that, as gradient descent progresses, the distance between $\boldsymbol{W}^{(t)}$ and $\boldsymbol{W}^*$ decreases until $L(t) \leq \epsilon$: $\|\boldsymbol{W}^{(t)} - \boldsymbol{W}^*\|_F^2 - \|\boldsymbol{W}^{(t+1)} - \boldsymbol{W}^*\|_F^2 \leq \eta L_S(t) - \eta \epsilon$. Moreover, we show that the difference between signal learning and noise memorization still holds in the second stage, as summarized below.

**Lemma 4.2.** *Let $T_2 = \eta^{-1}\sigma_p^{-2}d^{-1}nm \log(1/(\sigma_0 \sigma_p d)) + \eta^{-1}\epsilon^{-1}m^3 n \sigma_p^{-2} d^{-1}$. Under the same assumptions as Theorem 3.1, for training step $t \in [T_1, T_2]$, it holds that $\gamma_{j,r}^{(t)} \leq \tilde{O}(\sigma_0 \|\boldsymbol{\mu}\|_2)$, $|\underline{\rho}_{j,r,i}^{(t)}| \leq \tilde{O}(\sigma_0 \sigma_p \sqrt{d})$, and $\overline{\rho}_{j,r,i}^{(t)} \geq 1$. Besides, there exists a step $t \in [T_1, T_2]$, such that $L_S(t) < \epsilon$.*

Lemma 4.2 shows that standard GD achieves low training error after polynomially many steps, and noise memorization dominates the entire training process, which results in harmful overfitting.

### 4.3 Proof Sketch for Theorem 3.2

We also divide the training dynamics of label noise GD into two phases. In the first phase, both signal learning and noise memorization increase exponentially despite the presence of random label noise. In the second phase, label noise suppresses the growth of noise memorization, causing it to oscillate within a constant range; meanwhile, signal learning continues to grow exponentially until stabilizing at constant value, which leads to beneficial feature learning and low generalization error.

**First stage.** Leveraging the fact that the derivative of the loss function remains within a constant range due to small initialization, we demonstrate that both signal learning and noise memorization exhibit exponential growth rates, even in the presence of label noise. According to the iterative update of the signal learning coefficient in Equation (4), the upper and lower bounds are given as $\gamma_{j,r}^{(t)} + |\langle \boldsymbol{w}_{j,r}^{(0)}, \boldsymbol{\mu} \rangle| \leq \exp\left(\frac{2\eta \|\boldsymbol{\mu}\|_2^2}{m} t\right)|\langle \boldsymbol{w}_{j,r}^{(0)}, \boldsymbol{\mu} \rangle|$, and $\max_{r \in [m]}\{\gamma_{j,r}^{(t)} + j\langle \boldsymbol{w}_{j,r}^{(0)}, \boldsymbol{\mu} \rangle\} \geq \exp(\frac{C_0 \eta \|\boldsymbol{\mu}\|_2^2}{8m})\left(\max_{r \in [m]}\langle \boldsymbol{w}_{j,r}^{(0)}, \boldsymbol{\mu} \rangle\right)$, respectively. Here $C_0$ is the lower bound for the absolute loss derivative. These bounds indicate that signal learning grows exponentially with the number of training iterations. On the other hand, from the update equation (5), we characterize the behavior of noise memorization. Despite the injected label noise, we can show a lower bound on the noise memorization rate: $\max_{j,r}\{\overline{\rho}_{j,r,i}^{(t)} + 0.6|\langle \boldsymbol{w}_{j,r}^{(0)}, \boldsymbol{\xi}_i \rangle|\} \geq \exp(\frac{\eta C_0 \sigma_p^2 d}{2nm})|\langle \boldsymbol{w}_{j,r}^{(0)}, \boldsymbol{\xi}_i \rangle|$. The main results for the first stage are summarized in the following lemma.

**Lemma 4.3.** *Under the same condition with Theorem 3.2, and let $T_1 = \Theta(\frac{nm \log((1/\sigma_0 \sigma_p d))}{\eta \sigma_p^2 d})$. Then the following holds with probability at least $1 - d^{-1}$: (i) $\max_{j,r} \overline{\rho}_{j,r,i}^{(T_1)} \geq 0.1$, for all $i \in [n]$; (ii) $\max_{j,r,i} |\underline{\rho}_{j,r}^{(t)}| \leq \tilde{O}(\sigma_0 \sigma_p \sqrt{d})$, for all $t \in [T_1]$. (iii) $\max_{j,r} \gamma_{j,r}^{(t)} \geq \tilde{O}(\sigma_0 \|\boldsymbol{\mu}\|_2)$, for all $t \in [T_1]$.*

Lemma 4.3 states that both signal learning and noise memorization grow exponentially during the first stage. For the analysis of label noise GD, one additional technical challenge is the instability of training dynamics caused by the injected random noise, which we address as follows. For signal learning, we make use of the small label flipping rate $p$ and aggregate information across all sam-

ples via concentration. Whereas for noise memorization (which is tied to individual samples), we leverage the broad range of time steps in the first stage to establish the overall increment rate.

**Second stage.** As shown in Lemma 4.3, at the end of the first phase, noise memorization has reached a significant level, dominating the model's output. However, label noise introduces randomness in the labels, which affects the updates of noise memorization coefficients. We track the evolution of $\overline{\rho}_{j,r,i}^{(t)}$ via the following approximation. Define $\iota_i^{(t)} \triangleq \frac{1}{m} \sum_{r=1}^m \overline{\rho}_{y_i,r,i}^{(t)}$. The evolution of noise memorization under label noise GD can be approximated as

$$\iota_i^{(t+1)} \approx \begin{cases} (1 + \frac{\eta \sigma_p^2 d}{(1+\exp((\iota_i^{(t)})^2))nm})\iota_i^{(t)}, & \text{with prob } 1-p. \\ (1 - \frac{\eta \sigma_p^2 d}{(1+\exp(-(\iota_i^{(t)})^2))nm})\iota_i, & \text{with prob } p. \end{cases}$$

Unlike conventional approaches such as (Cao et al., 2022; Kou et al., 2023b), we analyze this process using a supermartingale argument and apply Azuma's inequality with a union bound over the second-stage training period. Via a martingale argument, we show that noise memorization remains at a constant level with high probability. While noise memorization stabilizes, signal learning continues to grow exponentially. This discrepancy enables signal learning to eventually dominate the generalization. The analysis of the second stage is summarized by the following lemma.

**Lemma 4.4.** *Under the same condition as Theorem 3.2, during $t \in [T_1, T_2]$ with $T_2 = T_1 + \log(6/(\sigma_0\|\boldsymbol{\mu}\|_2))4m(1 + \exp(c_2))\eta^{-1}\|\boldsymbol{\mu}\|_2^{-2}$, there exist a sufficient large positive constant $C_\iota$ and a constant $\iota_i^*$ depending on sample index $i$ such that the following results hold with probability at least $1 - 1/d$: (i) $|\iota_i^{(t)} - \iota_i^*| \le C_\iota$; (ii) $\gamma_{j,r}^{(t)} \le 0.1$ for all $j \in \{-1, 1\}$ and $r \in [m]$ (iii)$\frac{1}{2m}(\sum_{r=1}^m \overline{\rho}_{y_i,r,i}^{(t)})^2 \le f_i^{(t)} \le \frac{2}{m}(\sum_{r=1}^m \overline{\rho}_{y_i,r,i}^{(t)})^2$ and (iv) $\max_{r \in [m]}(\gamma_{j,r}^{(t)} + |\langle \boldsymbol{w}_{j,r}^{(0)}, \boldsymbol{\mu} \rangle|) \ge \exp\left(\frac{\eta\|\boldsymbol{\mu}\|_2^2}{16m}(t - T_1)\right)\max_{r \in [m]}|\gamma_{j,r}^{(T_1)} + \langle \boldsymbol{w}_{j,r}^{(0)}, \boldsymbol{\mu} \rangle|$.*

Lemma 4.4 demonstrates that label noise introduces a regularizing effect preventing the noise memorization coefficients from growing unchecked, while simultaneously allowing signal learning to grow to a sufficiently large value. Building on this result, we show that both signal learning and noise memorization reach a constant order of magnitude. Consequently, the population loss can be bounded by $L_{\mathcal{D}}(\boldsymbol{W}^{(t)}) \le 2\exp\left(-\frac{Cd}{n^2}\right)$, corresponding to the second bullet point of Theorem 3.2.

# 5 SYNTHETIC EXPERIMENTS

We conduct experiments using synthetic data to validate our theoretical results. The samples are generated according to Definition 2.1. The number of training and test sample is $n = 200$ and $n_{\text{test}} = 2000$, respectively, and the input dimension is set to $d = 2000$. The label noise flip rate is $p = 0.1$. We train the two-layer network with squared ReLU activation using standard GD and label noise GD for $t = 2000$ steps. The network width is $m = 20$ and the learning rate is $\eta = 0.5$. The signal vector is defined as $\boldsymbol{\mu} = [2, 0, 0, \ldots, 0] \in \mathbb{R}^d$ and the noise variance is set to $\sigma_p^2 = 0.25$.

**Dynamics of signal and noise coefficients.** In Figure 1, we present the feature learning coefficients defined in Section 4.1, the training loss and test accuracy for both algorithms. We observe that GD successfully minimizes the training loss to a near-zero value; however, noise memorization ($\rho$) significantly exceeds signal learning ($\gamma$), leading to poor test performance. In contrast, label noise GD does not fully minimize the training loss, as it oscillates around 0.5; consistent with our theoretical analysis, this behavior causes noise memorization to remain constant in the second stage, while signal learning continues to grow rapidly. As a result, the test accuracy of label noise GD steadily improves in the second stage.

**Heatmap of generalization error.** Next we explore a range of SNR values from 0.03 to 0.10 and sample sizes $n$ ranging from 100 to 700. For each combination of SNR and sample size $n$, we train the NN for 1000 steps with $\eta = 1.0$ using standard GD or label noise GD. The resulting test error is visualized in Figure 2. Observe that standard GD (left) fails to generalize when $\text{SNR} = O(n^{-1/2})$, which is consistent with our theoretical prediction in Theorem 3.1. On the other hand, label noise GD (right) achieves perfect test accuracy across a broader range of SNR, which agrees with Theorem 3.2.

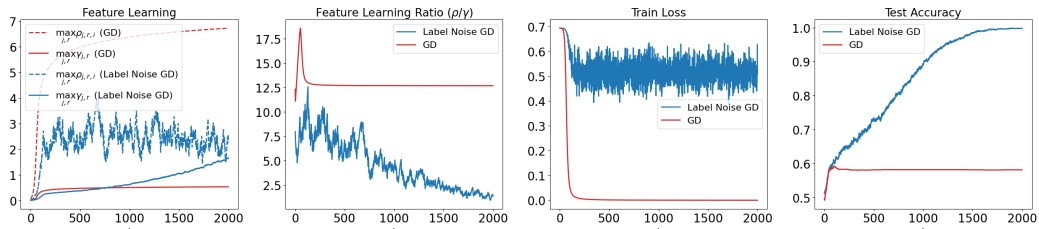

Figure 1: Ratio of noise memorization over signal learning, training loss, and test accuracy, of standard GD and label noise GD. See Section 4.1 for definitions of signal learning ($\gamma$) and noise memorization ($\rho$).

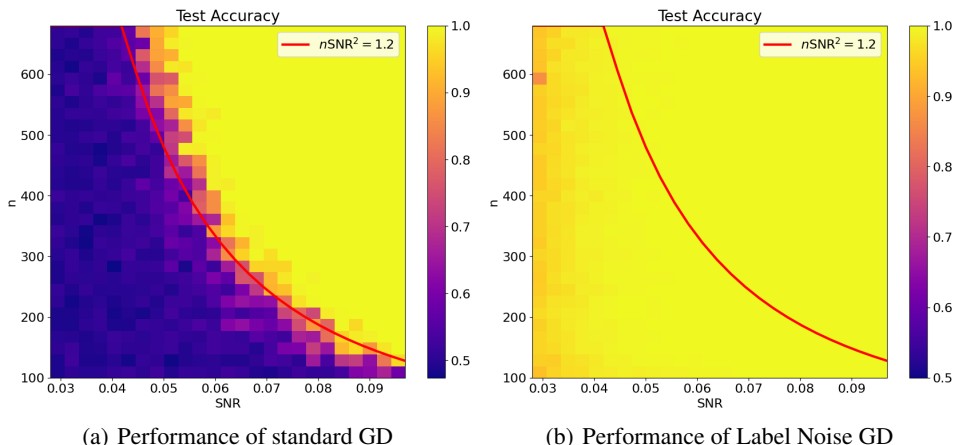

(a) Performance of standard GD          (b) Performance of Label Noise GD

Figure 2: Test accuracy heatmap of standard GD (left) and Label Noise GD (right) after training.

## 6 CONCLUSION AND LIMITATION

We presented a theoretical analysis of gradient-based feature learning in the challenging low SNR regime. Our main contribution is to demonstrate that label noise gradient descent (GD) can effectively enhance signal learning while suppressing noise memorization; this implicit regularization mechanism enables label noise GD to generalize in low SNR settings where standard GD suffers from harmful overfitting. Our theoretical findings are supported by experiments on synthetic data.

**Limitations.** We highlight a few limitations and future directions. Our current analysis applies to a specific choice of activation function (squared ReLU) and architecture (two-layer convolutional neural network); it would be interesting to extend this framework to more complex architectures such as deeper neural networks. Additionally, analyzing label noise GD under other optimization schemes, such as stochastic gradient descent (SGD) and adaptive optimizers like Adam, could provide a deeper understanding of the implicit regularization effects in practical settings.

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

## APPENDIX CONTENTS

## A  PRELIMINARY LEMMAS

**Lemma A.1** (Cao et al. (2022)). *Suppose that $\delta > 0$ and $d = \Omega(\log(4n/\delta))$. Then with probability $1 - \delta$,*

$$\sigma_p^2 d/2 \leq \|\boldsymbol{\xi}_i\|_2^2 \leq 3\sigma_p^2 d/2,$$
$$|\langle \boldsymbol{\xi}_i, \boldsymbol{\xi}_{i'} \rangle| \leq 2\sigma_p^2 \sqrt{d \log(4n^2/\delta)},$$

*for all $i, i' \neq i \in [n]$.*

**Lemma A.2** (Cao et al. (2022)). *Suppose that $d \geq \Omega(\log(mn/\delta))$, $m = \Omega(\log(1/\delta))$. Then with probability at least $1 - \delta$, it satisfies that for all $r \in [m], j \in \{\pm 1\}, i \in [n]$,*

$$|\langle \boldsymbol{w}_{j,r}^{(0)}, \boldsymbol{\mu} \rangle| \leq \sqrt{2\log(8m/\delta)}\sigma_0\|\boldsymbol{\mu}\|_2$$
$$|\langle \boldsymbol{w}_{j,r}^{(0)}, \boldsymbol{\xi}_i \rangle| \leq 2\sqrt{\log(8mn/\delta)}\sigma_0\sigma_p\sqrt{d}$$

*and for all $j \in \{\pm 1\}, i \in [n]$*

$$\sigma_0\|\boldsymbol{\mu}\|_2/2 \leq \max_{r \in [m]} j\langle \boldsymbol{w}_{j,r}^{(0)}, \boldsymbol{\mu} \rangle \leq \sqrt{2\log(8m/\delta)}\sigma_0\|\boldsymbol{\mu}\|_2,$$
$$\sigma_0\sigma_p\sqrt{d}/4 \leq \max_{r \in [m]} j\langle \boldsymbol{w}_{j,r}^{(0)}, \boldsymbol{\xi}_i \rangle \leq 2\sqrt{\log(8mn/\delta)}\sigma_0\sigma_p\sqrt{d}.$$

**Lemma A.3.** *Let $\mathcal{S}_\pm^{(t)} = \{i : \epsilon_i^{(t)} = \pm 1\}$ and $\mathcal{S}_j = \{i : y_i = j\}$. Then $\forall t \geq 0$, we have following with probability at least $1 - \delta$,*

*1.* $||\mathcal{S}_+^{(t)}| - n(1-p)| \leq \sqrt{\frac{n}{2}\log\left(\frac{4T^*}{\delta}\right)}$, *and* $||\mathcal{S}_-^{(t)}| - np| \leq \sqrt{\frac{n}{2}\log\left(\frac{4T^*}{\delta}\right)}$.

*2. The size of set follows, $\forall j \in \{\pm 1\}$*

$$\left||\mathcal{S}_+^{(t)} \cap \mathcal{S}_j| - \frac{(1-p)n}{2}\right| \leq \sqrt{\frac{n}{2}\log\left(\frac{8T^*}{\delta}\right)}, \quad \left||\mathcal{S}_-^{(t)} \cap \mathcal{S}_j| - \frac{pn}{2}\right| \leq \sqrt{\frac{n}{2}\log\left(\frac{8T^*}{\delta}\right)}.$$

*Suppose $n \geq \frac{8\log(8T^*/\delta)}{p^2} \geq \frac{8\log(8T^*/\delta)}{(1-p)^2}$, we have*

$$|\mathcal{S}_+^{(t)} \cap \mathcal{S}_j| \in \left[\frac{(2-3p)n}{4}, \frac{(2-p)n}{4}\right], \quad |\mathcal{S}_-^{(t)} \cap \mathcal{S}_j| \in \left[\frac{pn}{4}, \frac{3pn}{4}\right].$$

*Proof of Lemma A.3.* By independence, we have $\mathbb{E}|\mathcal{S}_+^{(t)}| = (1-p)n$ and $\mathbb{E}|\mathcal{S}_-^{(t)}| = pn$. By Hoeffding's inequality, we have for arbitrary $\tau > 0$,

$$\mathbb{P}\left(||\mathcal{S}_+^{(t)}| - (1-p)n| \geq \tau\right) \leq 2\exp\left(-\frac{2\tau^2}{n}\right), \quad \mathbb{P}\left(||\mathcal{S}_-^{(t)}| - pn| \geq \tau\right) \leq 2\exp\left(-\frac{2\tau^2}{n}\right).$$

Setting $\tau = \sqrt{(n/2)\log(4/\delta)}$ and taking the union bound over $[T^*]$ gives

$$||\mathcal{S}_+^{(t)}| - (1-p)n| \leq \sqrt{\frac{n}{2}\log\left(\frac{4T^*}{\delta}\right)}, \quad ||\mathcal{S}_-^{(t)}| - pn| \leq \sqrt{\frac{n}{2}\log\left(\frac{4T^*}{\delta}\right)},$$

which holds with probability at least $1 - \delta$.

Similarly, by the same argument, we can show the result for $|\mathcal{S}_+^{(t)} \cap \mathcal{S}_j|$ and $|\mathcal{S}_-^{(t)} \cap \mathcal{S}_j|$.

Suppose $n \geq \frac{8\log(8T^*/\delta)}{p^2} \geq \frac{8\log(8T^*/\delta)}{(1-p)^2}$, then we have with probability at least $1 - \delta$, we have $|\mathcal{S}_+^{(t)} \cap \mathcal{S}_j| \in \left[\frac{(2-3p)n}{4}, \frac{(2-p)n}{4}\right], \quad |\mathcal{S}_-^{(t)} \cap \mathcal{S}_j| \in \left[\frac{pn}{4}, \frac{3pn}{4}\right]$. $\square$

**Lemma A.4.** *Let $\mathcal{S}_{i,\pm}^{(t)} := \{s \leq t : \epsilon_i^{(s)} = \pm 1\}$. Then for any $i \in [n]$, $t > 0$, with probability at least $1 - \delta$,*

1. $||\mathcal{S}_{i,+}^{(t)}| - (1-p)t| \leq \sqrt{\frac{t}{2}\log(\frac{4n}{\delta})}$ and $||\mathcal{S}_{i,-}^{(t)}| - pt| \leq \sqrt{\frac{t}{2}\log(\frac{4n}{\delta})}$.

2. *In addition, suppose* $t \geq \frac{2\log(4n/\delta)}{p^2}$, *we have* $|\mathcal{S}_{i,+}^{(t)}| \in [\frac{(2-3p)t}{2}, \frac{(2-p)t}{2}]$, $|\mathcal{S}_{i,-}^{(t)}| \in [\frac{pt}{2}, \frac{3pt}{2}]$.

*Proof of Lemma A.4.* By independence, we have $\mathbb{E}|\mathcal{S}_{i,+}^{(t)}| = (1-p)t$ and $\mathbb{E}|\mathcal{S}_{i,-}^{(t)}| = pt$. By Hoeffding's inequality, we have for arbitrary $\tau > 0$,

$$\mathbb{P}\big(||\mathcal{S}_{i,+}^{(t)}| - (1-p)t| \geq \tau\big) \leq 2\exp\big(-\frac{2\tau^2}{t}\big), \quad \mathbb{P}\big(||\mathcal{S}_{i,-}^{(t)}| - pt| \geq \tau\big) \leq 2\exp\big(-\frac{2\tau^2}{t}\big).$$

Setting $\tau = \sqrt{(t/2)\log(4/\delta)}$ and taking the union bound gives

$$||\mathcal{S}_{i,+}^{(t)}| - (1-p)t| \leq \sqrt{\frac{t}{2}\log\Big(\frac{4n}{\delta}\Big)}, \quad ||\mathcal{S}_{i,-}^{(t)}| - pt| \leq \sqrt{\frac{t}{2}\log\Big(\frac{4n}{\delta}\Big)},$$

which holds with probability at least $1-\delta$.

Suppose $t \geq \frac{2\log(4n/\delta)}{p^2} \geq \frac{2\log(4n/\delta)}{(1-p)^2}$, then we have with probability at least $1-\delta$, we have $|\mathcal{S}_{i,+}^{(t)}| \in [\frac{(2-3p)t}{2}, \frac{(2-p)t}{2}], |\mathcal{S}_{i,-}^{(t)}| \in [\frac{pt}{2}, \frac{3pt}{2}]$. $\qquad\square$

# B PROOF OF PROPOSITION 4.1

In this section, we provide a proof for Proposition 4.1, which establishes upper bounds for the absolute values of the signal learning and noise memorization coefficients throughout the entire training stage. Additionally, we present some preliminary lemmas that will be used in the proof of Proposition 4.1 as well as in other results in the subsequent sections.

**Lemma B.1.** *Suppose that inequalities (7) and (8) hold for all* $r \in [m]$, $j \in \{-1, 1\}$, $i \in [n]$ *and* $t \in [0, T^*]$. *For any* $\delta > 0$, *with probability at least* $1-\delta$, *it holds that*

$$|\langle \boldsymbol{w}_{j,r}^{(t)} - \boldsymbol{w}_{j,r}^{(0)}, \boldsymbol{\xi}_i\rangle - \rho_{j,r,i}^{(t)}| \leq 8\sqrt{\frac{\log(4n^2/\delta)}{d}}n\alpha,$$

$$|\langle \boldsymbol{w}_{j,r}^{(t)} - \boldsymbol{w}_{j,r}^{(0)}, j\boldsymbol{\mu}\rangle - \gamma_{j,r}^{(t)}| = 0.$$

*Proof of Lemma B.1.* From the signal-noise decomposition of $\boldsymbol{w}_{j,r}^{(t)}$, we have

$$|\langle \boldsymbol{w}_{j,r}^{(t)} - \boldsymbol{w}_{j,r}^{(0)}, \boldsymbol{\xi}_i\rangle - \rho_{j,r,i}^{(t)}| \overset{(a)}{=} |j\gamma_{j,r}^{(t)}\langle \boldsymbol{\mu}, \boldsymbol{\xi}_i\rangle\|\boldsymbol{\mu}\|_2^{-2} + \sum_{i' \neq i} \rho_{j,r,i}^{(t)}\langle \boldsymbol{\xi}_{i'}, \boldsymbol{\xi}_i\rangle\|\boldsymbol{\xi}_{i'}\|_2^{-2}|$$

$$\overset{(b)}{\leq} 8\sqrt{\frac{\log(4n^2/\delta)}{d}}n\alpha,$$

where (a) follows from the weight decomposition (see Equation 3), and inequality (b) is due to Lemma A.1 and the upper bound of $\rho_{j,r,i}^{(t)}$ based on inequalities (7) and (8).

Next, for the projection of the weight difference onto the signal vector, we have:

$$|\langle \boldsymbol{w}_{j,r}^{(t)} - \boldsymbol{w}_{j,r}^{(0)}, j\boldsymbol{\mu}\rangle - \gamma_{j,r}^{(t)}| = |\sum_{i=1}^{n} \rho_{j,r,i}^{(t)}\|\boldsymbol{\xi}_i\|_2^{-2}\langle \boldsymbol{\xi}_i, \boldsymbol{\mu}\rangle| = 0,$$

where the equality holds because $\langle \boldsymbol{\xi}_i, \boldsymbol{\mu}\rangle = 0$ for $i \in [n]$ due to the covariance property of the noise vector distribution. $\qquad\square$

With Lemma B.1 in place, we are now prepared to prove Proposition 4.1. The general proof strategy follows the approach outlined in Cao et al. (2022). However, we present a complete proof here for the sake of clarity and to provide a unified analysis for both gradient descent and label noise GD.

*Proof of Proposition 4.1.* The proof uses induction and covers both gradient descent and label noise gradient descent.

At $t = 0$, it is straightforward that the results hold for all coefficients, as they are initialized to zero. Now, assume that there exists a time step $\hat{T}$ such that for $t \in [1, \hat{T}]$ the following inequalities hold:

$$0 \le \gamma_{j,r}^{(t)} \le \alpha, \quad 0 \le \overline{\rho}_{j,r,i}^{(t)} \le \alpha,$$

$$0 \ge \underline{\rho}_{j,r,i}^{(t)} \ge -\beta - 16\sqrt{\frac{\log(4n^2/\delta)}{d}}n\alpha \ge -\alpha.$$

To complete the induction, we need to show that the above inequalities hold for $t = \hat{T} + 1$. First, we examine $\underline{\rho}_{j,r,i}^{(\hat{T}+1)}$ for $j = -y_i$, since $\underline{\rho}_{j,r,i}^{(\hat{T}+1)} = 0$ when $j = y_i$ by definition. Using Lemma B.1, if $\underline{\rho}_{j,r,i}^{(\hat{T})} \le -0.5\beta - 8\sqrt{\frac{\log(4n^2/\delta)}{d}}n\alpha$, we have

$$\langle \boldsymbol{w}_{j,r}^{(\hat{T})}, \boldsymbol{\xi}_i \rangle \le \underline{\rho}_{j,r,i}^{(\hat{T})} + 8\sqrt{\frac{\log(4n^2/\delta)}{d}}n\alpha + \langle \boldsymbol{w}_{j,r}^{(0)}, \boldsymbol{\xi}_i \rangle \le 0.$$

Thus,

$$\underline{\rho}_{j,r,i}^{(\hat{T}+1)} = \underline{\rho}_{j,r,i}^{(\hat{T})} + \frac{\eta}{nm}\ell_i^{'(\hat{T})}\sigma'(\langle \boldsymbol{w}_{j,r}^{(\hat{T})}, \boldsymbol{\xi}_i \rangle)\|\boldsymbol{\xi}_i\|_2^2\epsilon_i^{(\hat{T})}$$

$$= \underline{\rho}_{j,r,i}^{(\hat{T})} \ge -\beta - 16\sqrt{\frac{\log(4n^2/\delta)}{d}}n\alpha,$$

where we have used $\sigma'(\langle \boldsymbol{w}_{j,r}^{(\hat{T})}, \boldsymbol{\xi}_i \rangle) = 0$. On the other hand, if $\underline{\rho}_{j,r,i}^{(\hat{T})} \ge -0.5\beta - 8\sqrt{\frac{\log(4n^2/\delta)}{d}}n\alpha$, the update function implies:

$$\underline{\rho}_{j,r,i}^{(\hat{T}+1)} \overset{(a)}{\ge} \underline{\rho}_{j,r,i}^{(\hat{T})} + \frac{\eta}{nm}\ell_i^{'(\hat{T})}\langle \boldsymbol{w}_{j,r}^{(\hat{T})}, \boldsymbol{\xi}_i \rangle\|\boldsymbol{\xi}_i\|_2^2$$

$$\overset{(b)}{\ge} -0.5\beta - 8\sqrt{\frac{\log(4n^2/\delta)}{d}}n\alpha - \frac{3\eta\sigma_p^2 d}{2nm}(0.5\beta + 8\sqrt{\frac{\log(4n^2/\delta)}{d}}n\alpha)$$

$$\overset{(c)}{\ge} -\beta - 16\sqrt{\frac{\log(4n^2/\delta)}{d}}n\alpha,$$

where (a) is due to choosing $\epsilon_i^{(\hat{T})} = 1$ and $\langle \boldsymbol{w}_{j,r}^{(\hat{T})}, \boldsymbol{\xi}_i \rangle > 0$, follows from Lemma A.1, and (c) holds when $\eta \le \frac{2nm}{3\sigma_p^2 d}$.

Next, consider $\overline{\rho}_{j,r,i}^{(\hat{T}+1)}$ for $j = y_i$. Let $\hat{T}_1$ to be the last time that $\overline{\rho}_{j,r,i}^{(t)} \le 0.5\alpha$. By propagation, we have:

$$\overline{\rho}_{j,r,i}^{(\hat{T}+1)} = \overline{\rho}_{j,r,i}^{(\hat{T}_1)} - \frac{\eta}{nm}\ell_i^{'(\hat{T}_1)}\sigma'(\langle \boldsymbol{w}_{j,r}^{(\hat{T}_1)}, \boldsymbol{\xi}_i \rangle)\|\boldsymbol{\xi}_i\|_2^2\epsilon_i^{(\hat{T}_1)} - \sum_{\hat{T}_1 < t \le \hat{T}} \frac{\eta}{nm}\ell_i^{'(t)}\sigma'(\langle \boldsymbol{w}_{j,r}^{(t)}, \boldsymbol{\xi}_i \rangle)\|\boldsymbol{\xi}_i\|_2^2\epsilon_i^{(t)}$$

$$\overset{(a)}{\le} 0.5\alpha + \frac{\eta}{nm}\langle \boldsymbol{w}_{j,r}^{(\hat{T}_1)}, \boldsymbol{\xi}_i \rangle\|\boldsymbol{\xi}_i\|_2^2 + \sum_{\hat{T}_1 < t \le \hat{T}} \frac{\eta}{nm}\ell_i^{'(t)}\langle \boldsymbol{w}_{j,r}^{(t)}, \boldsymbol{\xi}_i \rangle\|\boldsymbol{\xi}_i\|_2^2$$

$$\overset{(b)}{\le} 0.5\alpha + \frac{3\eta\sigma_p^2 d}{2nm}(0.5\alpha + \beta + 16\sqrt{\frac{\log(4n^2/\delta)}{d}}n\alpha)$$

$$+ \sum_{\hat{T}_1 < t \le \hat{T}} \exp(-4\alpha^2 + 1)\frac{3\eta\sigma_p^2 d}{2nm}(\alpha + \beta + 16\sqrt{\frac{\log(4n^2/\delta)}{d}}n\alpha)$$

$$\overset{(c)}{\le} 0.5\alpha + 0.25\alpha + 0.25\alpha = \alpha,$$

where (a) holds since $\ell_i^{'(\hat{T}_1)} \ge -1$ and $\epsilon_i^{(t)} \le 1$ for all $t \in [\hat{T}_1, \hat{T}]$, (b) is by Lemma A.1, Lemma B.1, and $-\tilde{\ell}_i^{'(t)} \le \exp(-F_{y_i}+1) \le \exp(-4\alpha^2+1)$. Here we have used that $\beta+16\sqrt{\frac{\log(4n^2/\delta)}{d}}n\alpha \le 2\alpha$

with the condition that $d = \tilde{\Omega}(n^2)$ and $\sigma_0 \leq \tilde{O}(1) \min\{\|\boldsymbol{\mu}\|_2^{-1}, \sigma_p^{-1} d^{-1/2}\}$. The final inequality (c) holds because $\eta = O(\frac{nm}{\sigma_p^2 d})$ and $\exp(-4\alpha^2 + 1)\alpha < 1$ with $\alpha = 4\log(T^*)$.

Similarly, we can prove that $\gamma_{j,r}^{(\hat{T}+1)} \leq \alpha$ using $\eta = O(\frac{nm}{\|\boldsymbol{\mu}\|_2^2})$, which completes the induction proof. $\qquad\square$

## C STANDARD GD FAILS TO GENERALIZE WITH LOW SNR

### C.1 PROOF OF LEMMA 4.1

In this section, we provide a proof for the result obtained in the first stage of gradient descent training. Several preliminary lemmas are established to facilitate the analysis.

**Lemma C.1** (Upper bound on $\gamma_{j,r}^{(t)}$)**.** *Under Assumption 3.1, in the first stage, where $0 \leq t \leq T_1 = \frac{nm \log(1/(\sigma_0 \sigma_p \sqrt{d}))}{\eta \sigma_p^2 d}$, there exists an upper bound for $\gamma_{j,r}^{(t)}$, for all $j \in \{-1, 1\}, r \in [m]$:*

$$\gamma_{j,r}^{(t)} + |\langle \boldsymbol{w}_{j,r}^{(0)}, \boldsymbol{\mu} \rangle| \leq \exp\big(\frac{2\eta\|\boldsymbol{\mu}\|_2^2}{m} t\big)|\langle \boldsymbol{w}_{j,r}^{(0)}, \boldsymbol{\mu} \rangle|.$$

*Proof of Lemma C.1.* By the iterative update rule of signal learning, we have:

$$\gamma_{j,r}^{(t+1)} \overset{(a)}{\leq} \gamma_{j,r}^{(t)} + \frac{\eta}{nm} \sum_{i=1}^n \sigma'(\langle \boldsymbol{w}_{j,r}^{(t)}, y_i \boldsymbol{\mu} \rangle)\|\boldsymbol{\mu}\|_2^2$$

$$\overset{(b)}{=} \gamma_{j,r}^{(t)} + \frac{\eta}{nm} \sum_{i=1}^n \sigma'(y_i \langle \boldsymbol{w}_{j,r}^{(0)}, \boldsymbol{\mu} \rangle + j y_i \gamma_{j,r}^{(t)})\|\boldsymbol{\mu}\|_2^2$$

$$\overset{(c)}{\leq} \gamma_{j,r}^{(t)} + \frac{2\eta}{m}(\gamma_{j,r}^{(t)} + |\langle \boldsymbol{w}_{j,r}^{(0)}, \boldsymbol{\mu} \rangle|)\|\boldsymbol{\mu}\|_2^2.$$

where (a) follows from $|\ell_i^{'(t)}| \leq 1$, (b) is derived using Lemma B.1, and (c) is due to the properties of the squared ReLU activation function.

Define $A^{(t)} := \gamma_{j,r}^{(t)} + |\langle \boldsymbol{w}_{j,r}^{(0)}, \boldsymbol{\mu} \rangle|$. Then, we have:

$$A^{(t+1)} \leq \big(1 + \frac{2\eta\|\boldsymbol{\mu}\|_2^2}{m}\big)A^{(t)} \leq \big(1 + \frac{2\eta\|\boldsymbol{\mu}\|_2^2}{m}\big)^{(t)} A^{(0)} \leq \exp\big(\frac{2\eta\|\boldsymbol{\mu}\|_2^2}{m} t\big)A^{(0)},$$

where we use $1 + x \leq \exp(x)$. This suggests:

$$\gamma_{j,r}^{(t)} + |\langle \boldsymbol{w}_{j,r}^{(0)}, \boldsymbol{\mu} \rangle| \leq \exp\big(\frac{2\eta\|\boldsymbol{\mu}\|_2^2}{m} t\big)|\langle \boldsymbol{w}_{j,r}^{(0)}, \boldsymbol{\mu} \rangle|.$$

$\qquad\square$

**Lemma C.2** (Upper bound on $\underline{\rho}_{j,r,i}^{(t)}$)**.** *Under Assumption 3.1, in the first stage, where $0 \leq t \leq T_1 = \frac{nm \log(1/(\sigma_0 \sigma_p \sqrt{d}))}{\eta \sigma_p^2 d}$, there exists an upper bound for $|\underline{\rho}_{j,r,i}^{(t)}|$, for all $j, r, i$:*

$$|\underline{\rho}_{j,r,i}^{(t)}| = \tilde{O}(\sigma_0 \sigma_p \sqrt{d}).$$

*Proof of Lemma C.2.* The proof uses the induction method. By the iterative update rule for noise memorization, we have:

$$|\underline{\rho}_{j,r,i}^{(t+1)}| \overset{(a)}{\leq} |\underline{\rho}_{j,r,i}^{(t)}| + \frac{\eta}{nm} \sigma'(\langle \boldsymbol{w}_{j,r}^{(t)}, \boldsymbol{\xi}_i \rangle)\|\boldsymbol{\xi}_i\|_2^2$$

$$\overset{(b)}{\leq} |\underline{\rho}_{j,r,i}^{(t)}| + \frac{3\eta\sigma_p^2 d}{2nm} \sigma'(\langle \boldsymbol{w}_{j,r}^{(0)}, \boldsymbol{\xi}_i \rangle + 16\sqrt{\frac{\log(4n^2/\delta)}{d}} n\alpha + \underline{\rho}_{j,r,i}^{(t)})$$

$$\overset{(c)}{\leq} |\underline{\rho}_{j,r,i}^{(t)}| + \frac{3\eta\sigma_p^2 d}{nm} \sqrt{\log(8mn/\delta)}\sigma_0 \sigma_p \sqrt{d},$$

where the inequality (a) is by the upper bound on $|\ell_i^{'(t)}| \leq 1$; Inequality (b) is derived using Proposition 4.1, Lemma A.1, and Lemma B.1. Finally, the inequality (c) uses the fact that $\underline{\rho}_{j,r,i}^{(t)} < 0$ and Lemma A.2.

Taking a telescoping sum over $t$ form $0$ to $T_1$, we obtain:

$$|\underline{\rho}_{j,r,i}^{(T_1)}| \leq \frac{3\eta\sigma_p^2 dT_1}{nm}\sqrt{\log(8mn/\delta)}\sigma_0\sigma_p\sqrt{d} = \tilde{O}(\sigma_0\sigma_p\sqrt{d}),$$

where we substituted $T_1 = \Theta\left(\frac{nm\log(1/(\sigma_0\sigma_p\sqrt{d}))}{\eta\sigma_p^2 d}\right)$, thereby completing the proof. $\qquad\square$

**Lemma C.3.** *Let* $\bar{\beta} = \min_{i\in[n]}\max_{r\in[m]}\langle \boldsymbol{w}_{y_i,r}^{(0)}, \boldsymbol{\xi}_i\rangle$. *Suppose that* $\sigma_0 \geq$ $160n\sqrt{\frac{\log(4n^2/\delta)}{d}}(\sigma_p\sqrt{d})^{-1}\alpha$. *Then it holds that* $\bar{\beta} \geq 40n\sqrt{\frac{\log(4n^2/\delta)}{d}}\alpha$.

*Proof of Lemma C.3.* The proof follows directly from Lemma A.2. With high probability, we have: $\bar{\beta} \geq \sigma_0\sigma_p\sqrt{d}/4$. Substituting the condition on $\sigma_0$, we obtain:

$$\bar{\beta} \geq 40n\sqrt{\frac{\log(4n^2/\delta)}{d}}\alpha.$$

$\square$

**Lemma C.4** (Lower bound on $\bar{\rho}_{j,r,i}^{(t)}$). *Under Assumption 3.1, in the first stage, where* $0 \leq t \leq T_1 = \frac{nm\log(1/(\sigma_0\sigma_p\sqrt{d}))}{\eta\sigma_p^2 d}$, *there exists a lower bound for* $\max_{j,r} \bar{\rho}_{j,r,i}^{(t)}$, *for all* $i \in [n]$:

$$\max_{j,r}\bar{\rho}_{j,r,i}^{(t)} + \bar{\beta} \geq \exp\left(\frac{\eta C_1\sigma_p^2 d}{2nm}t\right)\sigma_0\sigma_p\sqrt{d}/4.$$

*Proof of Lemma C.4.* By the iterative update rule for noise memorization, we have:

$$\max_{j,r}\bar{\rho}_{j,r,i}^{(t+1)} \overset{(a)}{\geq} \max_{j,r}\bar{\rho}_{j,r,i}^{(t)} + \max_{j,r}\frac{\eta C_1}{nm}\sigma'(\langle \boldsymbol{w}_{j,r}^{(t)}, \boldsymbol{\xi}_i\rangle)\|\boldsymbol{\xi}_i\|_2^2$$

$$\overset{(b)}{\geq} \max_{j,r}\bar{\rho}_{j,r,i}^{(t)} + \max_{j,r}\frac{\eta\sigma_p^2 dC_1}{2nm}\sigma'\left(\langle \boldsymbol{w}_{j,r}^{(0)}, \boldsymbol{\xi}_i\rangle - 16\sqrt{\frac{\log(4n^2/\delta)}{d}}n\alpha + \bar{\rho}_{j,r,i}^{(t)}\right)$$

$$\overset{(c)}{\geq} \max_{j,r}\bar{\rho}_{j,r,i}^{(t)} + \frac{\eta\sigma_p^2 dC_1}{nm}\left(\max_{j,r}\bar{\rho}_{j,r,i}^{(t)} + \frac{2}{5}\bar{\beta}\right),$$

where the inequality (a) is by the lower bound on $|\ell_i^{'(t)}| \geq C_1$ in the first stage; Inequality (b) is by Lemma A.1 and Lemma B.1. Finally, the inequality (c) is by Lemma C.3.

Define $B_i^{(t)} := \max_{j,r}\bar{\rho}_{j,r,i}^{(t)} + 0.6\bar{\beta}$. Then

$$B_i^{(t+1)} \geq \left(1 + \frac{\eta C_1\sigma_p^2 d}{nm}\right)B_i^{(t)} \geq \left(1 + \frac{\eta C_1\sigma_p^2 d}{nm}\right)^{(t)}B_i^{(0)} \geq \exp\left(\frac{\eta C_1\sigma_p^2 d}{2nm}t\right)B_i^{(0)},$$

where we used $1 + x \geq \exp(x/2)$ for $x \leq 2$. $\qquad\square$

With the above lemmas in place, we are now ready to prove Lemma 4.1.

*Proof of Lemma 4.1.* We choose the end of stage 1 as $T_1 = \frac{4nm}{\eta\sigma_p^2 d}\log(1/(\sigma_0\sigma_p\sqrt{d}))$. Then by Lemma C.4, we conclude that $\max_{j,r}\bar{\rho}_{j,r,i}^{(T_1)} \geq 1$, for all $i \in [n]$. Besides, by Lemma C.2, we directly obtain the result that

$$|\underline{\rho}_{j,r,i}^{(T_1)}| \leq \frac{3\eta\sigma_p^2 dT_1}{nm}\sqrt{\log(8mn/\delta)}\sigma_0\sigma_p\sqrt{d} = \tilde{O}(\sigma_0\sigma_p\sqrt{d}).$$

Finally, Lemma C.1 yields

$$\gamma_{j,r}^{(T_1)} + |\langle \boldsymbol{w}_{j,r}^{(0)}, \boldsymbol{\mu}\rangle| \leq \exp\left(\frac{2\eta\|\boldsymbol{\mu}\|_2^2}{m}\frac{4nm}{\eta\sigma_p^2 d}\log(1/(\sigma_0\sigma_p d))\right)|\langle \boldsymbol{w}_{j,r}^{(0)}, \boldsymbol{\mu}\rangle| \leq 2|\langle \boldsymbol{w}_{j,r}^{(0)}, \boldsymbol{\mu}\rangle|,$$

where we have used the condition of low SNR, namely $n\text{SNR}^2 \leq 1/\log(\sigma_0\sigma_p d)$. By Lemma A.2, we conclude the proof for $\max_{j,r}\gamma_{j,r}^{(T_1)} = \tilde{O}(\sigma_0\|\boldsymbol{\mu}\|_2)$. $\qquad\square$

## C.2 PROOF OF LEMMA 4.2

In this section, we provide a complete proof for Lemma 4.2 based on Lemma 4.1 and an iterative analysis of the training dynamics. We introduce several necessary preliminary lemmas that will be used in the proof for $t \in [T_1, T_2]$ with $T_2 = \eta^{-1}\sigma_p^{-2}d^{-1}nm \log(1/(\sigma_0\sigma_p\sqrt{d})) + \eta^{-1}\epsilon^{-1}m^3n\sigma_p^{-2}d^{-1}$.

**Lemma C.5** (Cao et al. (2022)). *Under the same condition as Theorem 3.1, for all $t \in [T_1, T_2]$ and $i \in [n]$, the following properties hold:*

$$\|\nabla L_S(\boldsymbol{W}^{(t)})\|_F^2 = O(\sigma_p^2 d)L_S(\boldsymbol{W}^{(t)}),$$

$$\|\boldsymbol{W}^{(T_1)} - \boldsymbol{W}^*\|_F = \tilde{O}(m^{3/2}n^{1/2}\sigma_p^{-1}d^{-1/2}),$$

$$y_i\langle\nabla f(\boldsymbol{W}^{(t)}, \boldsymbol{x}_i), \boldsymbol{W}^*\rangle \geq 2\log(2/\epsilon).$$

With the above lemmas at hand, we are now ready to provide the complete proof for Lemma 4.2.

*Proof of Lemma 4.2.* We start by showing the convergence of gradient descent. The key idea is to construct a reference weight matrix $\boldsymbol{W}^*$ defined as $\boldsymbol{w}_{j,r}^* = \boldsymbol{w}_{j,r}^{(0)} + 2m\log(2/\epsilon)\sum_{i=1}^n \|\boldsymbol{\xi}_i\|_2^{-2}\boldsymbol{\xi}_i$.

Summing the above inequality from $\boldsymbol{W}^{(t)}$ and $\boldsymbol{W}^*$:

$$\|\boldsymbol{W}^{(t)} - \boldsymbol{W}^*\|_F^2 - \|\boldsymbol{W}^{(t+1)} - \boldsymbol{W}^*\|_F^2$$

$$= \|\boldsymbol{W}^{(t)} - \boldsymbol{W}^*\|_F^2 - \|\boldsymbol{W}^{(t)} - \eta\nabla L_S(\boldsymbol{W}^{(t)}) - \boldsymbol{W}^*\|_F^2$$

$$= 2\eta\langle\nabla L_S(\boldsymbol{W}^{(t)}), \boldsymbol{W}^{(t)} - \boldsymbol{W}^*\rangle - \eta^2\|\nabla L_S(\boldsymbol{W}^{(t)})\|_F^2$$

$$\overset{(a)}{=} \frac{2\eta}{n}\sum_{i=1}^n \ell_i^{'(t)}[2y_i f(\boldsymbol{W}^{(t)}, \boldsymbol{x}_i) - \langle\nabla f(\boldsymbol{W}^{(t)}, \boldsymbol{x}_i), \boldsymbol{W}^*\rangle] - \eta^2\|\nabla L_S(\boldsymbol{W}^{(t)})\|_F^2$$

$$\overset{(b)}{\geq} \frac{2\eta}{n}\sum_{i=1}^n \ell_i^{'(t)}[2y_i f(\boldsymbol{W}^{(t)}, \boldsymbol{x}_i) - 2\log(2/\epsilon)] - \eta^2\|\nabla L_S(\boldsymbol{W}^{(t)})\|_F^2$$

$$\overset{(c)}{\geq} \frac{4\eta}{n}\sum_{i=1}^n [\ell_i^{(t)} - \epsilon/2] - \eta^2\|\nabla L_S(\boldsymbol{W}^{(t)})\|_F^2$$

$$\overset{(d)}{\geq} 2\eta(L_S(\boldsymbol{W}^{(t)}) - \epsilon),$$

where in equation (a), we have applied the homogeneity property of the squared ReLU activation. The inequality (b) is by $\langle\nabla f(\boldsymbol{W}^{(t)}, \boldsymbol{x}_i), \boldsymbol{W}^*\rangle \geq 2\log(2/\epsilon)$ as stated in Lemma C.5, and the inequality (c) is due to the convexity of the logistic function. Finally, the inequality (d) is by Lemma C.5 and the condition on the learning rate.

Taking a summation over the above inequality from $T_1$ to $T_2$, we have

$$\sum_{t=T_1}^{T_2} L_S(\boldsymbol{W}^{(t)}) \leq \frac{\|\boldsymbol{W}^{(T_1)} - \boldsymbol{W}^*\|_F^2 + \eta\epsilon(T_2 - T_1 + 1)}{2\eta}$$

$$\leq \frac{\|\boldsymbol{W}^{(T_1)} - \boldsymbol{W}^*\|_F^2}{\eta}$$

$$\leq \tilde{O}(\eta^{-1}m^3 n\sigma_p^{-2}d^{-1}), \tag{9}$$

where in the second inequality, we have applied Lemma C.5. Finally, plugging in the $T_2 = \eta^{-1}\epsilon^{-1}m^3 n\sigma_p^{-2}d^{-1} + \eta^{-1}\sigma_p^{-2}d^{-1}nm\log(1/(\sigma_0\sigma_p\sqrt{d}))$, we achieve $L_S(\boldsymbol{W}^{(t)}) \leq \epsilon$.

Next, we provide the lower bound for the noise memorization coefficient $\overline{\rho}_{j,r,i}^{(t)}$ and the upper bound for the signal learning coefficient $\gamma_{j,r}^{(t)}$ in the second stage. For the noise memorization coefficient, using its update equation:

$$\overline{\rho}_{j,r,i}^{(t+1)} = \overline{\rho}_{j,r,i}^{(t)} - \frac{\eta}{nm}\ell_i^{'(t)}\sigma'(\langle\boldsymbol{w}_{j,r}^{(t)}, \boldsymbol{\xi}_i\rangle)\|\boldsymbol{\xi}_i\|_2^2 \geq \overline{\rho}_{j,r,i}^{(t)}.$$

Here, we have used $\ell_i'^{(t)} \geq 0$ and property of the squared ReLU activation. This implies that $\overline{\rho}_{j,r,i}^{(t)}$ never decreases during training. Therefore, we have $\max_{j,r} \overline{\rho}_{j,r,i}^{(t)} \geq 1$, for all $i \in [n]$ and $t \in [T_1, T_2]$.

For the signal learning coefficient, we use the induction method. From Lemma 4.1, we know that $\max_{j,r} \gamma_{j,r}^{(T_1)} = \tilde{O}(\sigma_0 \|\boldsymbol{\mu}\|_2) \triangleq \hat{\beta}$. Suppose that there exists $T \in [T_1, T_2]$ such that $\max_{j,r} \gamma_{j,r}^{(t)} \leq 2\hat{\beta}$ for all $t \in [T_1, T]$. Then we analyze:

$$\gamma_{j,r}^{(T+1)} = \gamma_{j,r}^{(T_1)} - \frac{\eta}{nm} \sum_{t=T_1}^{T} \sum_{i=1}^{n} \ell_i'^{(t)} \sigma'(\langle \boldsymbol{w}_{j,r}, \boldsymbol{\mu} \rangle) \|\boldsymbol{\mu}\|_2^2$$

$$\overset{(a)}{\leq} \gamma_{j,r}^{(T_1)} + \frac{2\eta\hat{\beta}}{nm} \|\boldsymbol{\mu}\|_2^2 \sum_{t=T_1}^{T} \sum_{i=1}^{n} |\ell_i'^{(t)}|$$

$$\overset{(b)}{\leq} \gamma_{j,r}^{(T_1)} + \frac{2\eta\hat{\beta}}{nm} \|\boldsymbol{\mu}\|_2^2 \sum_{t=T_1}^{T} L_S(\boldsymbol{W}^{(t)})$$

$$\overset{(c)}{\leq} \gamma_{j,r}^{(T_1)} + \frac{2\eta\hat{\beta}}{nm} \|\boldsymbol{\mu}\|_2^2 \tilde{O}(\eta^{-1} m^3 n \sigma_p^{-2} d^{-1})$$

$$\leq \gamma_{j,r}^{(T_1)} + \tilde{O}(n\mathrm{SNR}^2) \overset{(d)}{\leq} 2\hat{\beta}.$$

where the inequality (a) is due to Lemma B.1, the inequality (b) is by $|\ell_i'| \leq \ell_i$ for $i \in [n]$, and the inequality (c) is due to the inequality (9). Finally, the inequity (d) is by the condition that $n^{-1}\mathrm{SNR}^{-2} = \tilde{\Omega}(1)$. Similarly, with the induction method, we can show that $|\underline{\rho}_{j,r,i}^{(t)}| \leq \tilde{O}(\sigma_0 \sigma_p \sqrt{d})$.

□

## C.3 PROOF OF THEOREM 3.1

To complete the proof of Theorem 3.1, we provide a proof for the generalization result.

**Lemma C.6.** *Define* $g(\boldsymbol{\xi}) = \frac{1}{m} j \sum_{j,r} \sigma(\langle \boldsymbol{w}_{j,r}^{(t)}, \boldsymbol{\xi} \rangle)$. *Under Assumption 3.1, there exists a fixed vector* $\boldsymbol{v}$ *with* $\|\boldsymbol{v}\|_2 \leq 0.02\sigma_p$ *such that*

$$\sum_{j \in \{\pm 1\}} [g(j\boldsymbol{\xi} + \boldsymbol{v}) - g(\boldsymbol{\xi})] \geq 4\tilde{\Omega}(\sigma_0^2 \|\boldsymbol{\mu}\|_2^2).$$

*Proof of Lemma C.6.* To proceed with the proof, we construct the vector $\boldsymbol{v} \triangleq \lambda \sum_{i:y_i=1} \boldsymbol{\xi}_i$, where $\lambda = 0.01/\sqrt{nd}$. Then we show that

$$\|\boldsymbol{v}\|_2^2 = \|\lambda \sum_{i:y_i=1} \boldsymbol{\xi}_i\|_2^2 = \lambda^2 \langle \sum_{i:y_i=1} \boldsymbol{\xi}_i, \sum_{i:y_i=1} \boldsymbol{\xi}_i \rangle$$

$$= \lambda^2 \sum_{i:y_i=1} \|\boldsymbol{\xi}_i\|_2^2 + 2\lambda^2 \sum_i \sum_{j \neq i} \langle \boldsymbol{\xi}_i, \boldsymbol{\xi}_j \rangle$$

$$\leq \lambda^2 n \sigma_p^2 d + 4n^2 \lambda^2 \sigma_p^2 \sqrt{2d \log(4n^2/\delta)}$$

$$\leq 4\lambda^2 n \sigma_p^2 d = 0.02^2 \sigma_p^2,$$

where the first inequity is by Lemma A.1, the second inequality is by $d \geq \tilde{\Omega}(n^2)$, and the final equality is by $\lambda = 0.01/\sqrt{nd}$, which confirms that $\|\boldsymbol{v}\|_2 \leq 0.02\sigma_p$.

By the convexity property of the squared ReLU function, we have that

$$\sigma(\langle \boldsymbol{w}_{1,r}^{(t)}, \boldsymbol{\xi} + \boldsymbol{v} \rangle) - \sigma(\langle \boldsymbol{w}_{1,r}^{(t)}, \boldsymbol{\xi} \rangle) \geq \sigma'(\langle \boldsymbol{w}_{1,r}^{(t)}, \boldsymbol{\xi} \rangle) \langle \boldsymbol{w}_{1,r}^{(t)}, \boldsymbol{v} \rangle,$$

$$\sigma(\langle \boldsymbol{w}_{1,r}^{(t)}, -\boldsymbol{\xi} + \boldsymbol{v} \rangle) - \sigma(\langle \boldsymbol{w}_{1,r}^{(t)}, -\boldsymbol{\xi} \rangle) \geq \sigma'(\langle \boldsymbol{w}_{1,r}^{(t)}, -\boldsymbol{\xi} \rangle) \langle \boldsymbol{w}_{1,r}^{(t)}, \boldsymbol{v} \rangle.$$

With the above inequalities, we have that almost surely for all $\boldsymbol{\xi}$:

$$\sigma(\langle \boldsymbol{w}_{1,r}^{(t)}, \boldsymbol{\xi} + \boldsymbol{v} \rangle) - \sigma(\langle \boldsymbol{w}_{1,r}^{(t)}, \boldsymbol{\xi} \rangle) + \sigma(\langle \boldsymbol{w}_{1,r}^{(t)}, -\boldsymbol{\xi} + \boldsymbol{v} \rangle) - \sigma(\langle \boldsymbol{w}_{1,r}^{(t)}, -\boldsymbol{\xi} \rangle)$$

$$\geq 4|\langle \boldsymbol{w}_{1,r}^{(t)}, \boldsymbol{\xi} \rangle||\langle \boldsymbol{w}_{1,r}^{(t)}, \boldsymbol{v} \rangle.$$

On the other hand, using the properties of the squared ReLU function and the triangle inequality, we have:

$$\sigma(\langle \boldsymbol{w}_{-1,r}^{(t)}, \boldsymbol{\xi} + \boldsymbol{v} \rangle) - \sigma(\langle \boldsymbol{w}_{-1,r}^{(t)}, \boldsymbol{\xi} \rangle) + \sigma(\langle \boldsymbol{w}_{-1,r}^{(t)}, -\boldsymbol{\xi} + \boldsymbol{v} \rangle) - \sigma(\langle \boldsymbol{w}_{-1,r}^{(t)}, -\boldsymbol{\xi} \rangle)$$

$$\leq (\langle \boldsymbol{w}_{-1,r}^{(t)}, \boldsymbol{\xi} \rangle + |\langle \boldsymbol{w}_{-1,r}^{(t)}, \boldsymbol{v} \rangle|)^2 + (-\langle \boldsymbol{w}_{-1,r}^{(t)}, \boldsymbol{\xi} \rangle + |\langle \boldsymbol{w}_{-1,r}^{(t)}, \boldsymbol{v} \rangle|)^2 - \langle \boldsymbol{w}_{-1,r}^{(t)}, \boldsymbol{\xi} \rangle^2$$

$$\leq |\langle \boldsymbol{w}_{-1,r}^{(t)}, \boldsymbol{\xi} \rangle|^2 + 2|\langle \boldsymbol{w}_{-1,r}^{(t)}, \boldsymbol{v} \rangle|^2.$$

Next, we compare $|\langle \boldsymbol{w}_{1,r}^{(t)}, \boldsymbol{v} \rangle|$ and $|\langle \boldsymbol{w}_{-1,r}^{(t)}, \boldsymbol{v} \rangle|$ with $|\langle \boldsymbol{w}_{1,r}^{(t)}, \boldsymbol{\xi} \rangle|$ and $|\langle \boldsymbol{w}_{-1,r}^{(t)}, \boldsymbol{\xi} \rangle|$. We show that

$$|\langle \boldsymbol{w}_{-1,r}^{(t)}, \boldsymbol{v} \rangle| = \lambda|(\sum_{i:y_i=1} \underline{\rho}_{-1,r,i}^{(t)} + \langle \boldsymbol{w}_{-1,r}^{(0)}, \sum_{i:y_1=1} \boldsymbol{\xi}_i \rangle)|$$

$$\leq \lambda(n\sqrt{\log(12mn/\delta)})\sigma_0 \sigma_p \sqrt{d}) \leq \lambda n/4,$$

where the first inequality is by Lemma A.2 and Lemma 4.2, and the second inequality is by the condition on $\sigma_0$ from Assumption 3.1. Besides,

$$|\langle \boldsymbol{w}_{1,r}^{(t)}, \boldsymbol{v} \rangle| = \lambda|(\sum_{i:y_i=1} \overline{\rho}_{1,r,i}^{(t)} + \langle \boldsymbol{w}_{1,r}^{(0)}, \sum_{i:y_1=1} \boldsymbol{\xi}_i \rangle)|$$

$$\geq \lambda(n - n\sqrt{\log(12mn/\delta)})\sigma_0 \sigma_p \sqrt{d}) \geq \lambda n/2,$$

where the first inequality is by Lemma A.2 and Lemma 4.2; and the second inequality is by the condition on $\sigma_0$ from Assumption 3.1.

Finally, by Lemma A.2, Proposition 4.1, and Lemma A.1 it holds that

$$|\langle \boldsymbol{w}_{1,r}^{(t)}, \boldsymbol{\xi} \rangle| = |\langle \boldsymbol{w}_{1,r}^{(0)}, \boldsymbol{\xi} \rangle + \sum_{i=1}^{n} \rho_{j,r,i}^{(t)} \|\boldsymbol{\xi}_i\|_2^{-2} \langle \boldsymbol{\xi}_i, \boldsymbol{\xi} \rangle|$$

$$\leq \sqrt{\log(12mn/\delta)})\sigma_0 \sigma_p \sqrt{d} + 8\sqrt{\frac{\log(4n^2/\delta)}{d}} \sqrt{n} \alpha.$$

On the other hand, it is observed that $\langle \boldsymbol{w}_{1,r}^{(t)} - \boldsymbol{w}_{1,r}^{(0)}, \boldsymbol{\xi} \rangle \sim \mathcal{N}(0, \sigma_w^2)$, where the variance $\sigma_w$ follows

$$\sigma_w^2 = \sigma_p^2 \sum_{k=1}^{d} (\sum_{i=1}^{n} \rho_{j,r,i}^{(t)} \|\boldsymbol{\xi}_i\|_2^{-2} \xi_{i,k})^2$$

$$\overset{(a)}{\geq} \frac{1}{2} \sigma_p^2 \sum_{k=1}^{d} \sum_{i=1}^{n} (\rho_{j,r,i}^{(t)})^2 \|\boldsymbol{\xi}_i\|_2^{-4} \xi_{i,k}^2$$

$$= \frac{1}{2} \sigma_p^2 \sum_{i=1}^{n} (\rho_{j,r,i}^{(t)})^2 \|\boldsymbol{\xi}_i\|_2^{-2}$$

$$\geq \frac{1}{3d} \sum_{i=1}^{n} (\rho_{j,r,i}^{(t)})^2 \geq \frac{n}{6d},$$

where (a) is by Lemma A.1 and condition on $d$ from Assumption 3.1, (b) is due to Lemma A.1, and (c) is by Lemma 4.2.

By the anti-concentration inequality of Gaussian variance, we have

$$\mathbb{P}(|\langle \boldsymbol{w}_{1,r}^{(t)} - \boldsymbol{w}_{1,r}^{(0)}, \boldsymbol{\xi} \rangle| \leq \tau) \leq 2\text{erf}(\frac{\tau}{\sqrt{2}\sigma_w}) \leq 2\text{erf}(\frac{\tau\sqrt{6d}}{\sqrt{2n}})$$

$$\leq 2\sqrt{1 - \exp(-\frac{12d\tau^2}{\pi n})}.$$

Then with probability at least $1 - \delta$, it holds that

$$|\langle \boldsymbol{w}_{1,r}^{(t)} - \boldsymbol{w}_{1,r}^{(0)}, \boldsymbol{\xi} \rangle| \geq \sqrt{\frac{\pi n}{12d} \log(\frac{1}{1 - (\delta/2)^2})} \geq \sqrt{\frac{\pi n \delta^2}{96d}},$$

where we have used $\log(1+x) \geq \frac{x}{1+x}$ for $x > -1$ and $\delta^2 \leq 1/8$.

Together, we conclude that

$$\sum_{j \in \{\pm 1\}} [g(j\boldsymbol{\xi} + \boldsymbol{v}) - g(\boldsymbol{\xi})] \geq 4|\langle \boldsymbol{w}_{1,r}^{(t)}, \boldsymbol{\xi} \rangle||\langle \boldsymbol{w}_{1,r}^{(t)}, \boldsymbol{v} \rangle| + |\langle \boldsymbol{w}_{-1,r}^{(t)}, \boldsymbol{\xi} \rangle|^2 + 2|\langle \boldsymbol{w}_{-1,r}^{(t)}, \boldsymbol{v} \rangle|^2$$

$$\geq 4(\lambda/2)\sqrt{\frac{\pi n \delta^2}{96d}} \geq 4\tilde{\Omega}(\sigma_0^2 \|\boldsymbol{\mu}\|_2^2),$$

where the final inequality holds by $\sigma_0^2 \leq \tilde{O}(\frac{1}{d^{5/4}\|\boldsymbol{\mu}\|_2^2})$ with $\delta$ chosen as $d^{-1/4}$, thus completing the proof. $\qquad\square$

*Proof of Theorem 3.1.* For the population loss, we expand the expression

$$L_{\mathcal{D}}^{0-1}(\boldsymbol{W}^{(t)}) = \mathbb{E}_{(\boldsymbol{x},y) \sim \mathcal{D}}[\mathbb{1}(y \neq \text{sign}(f(\boldsymbol{W}, \boldsymbol{x})))] = \mathbb{P}(yf(\boldsymbol{W}^{(t)}, \boldsymbol{x}) < 0)$$

$$= \mathbb{P}\Big(\frac{1}{m}\sum_{r=1}^m \sigma(\langle \boldsymbol{w}_{-y,r}^{(t)}, \boldsymbol{\xi} \rangle) - \frac{1}{m}\sum_{r=1}^m \sigma(\langle \boldsymbol{w}_{y,r}^{(t)}, \boldsymbol{\xi} \rangle) \geq$$

$$\frac{1}{m}\sum_{r=1}^m \sigma(\langle \boldsymbol{w}_{y,r}^{(t)}, y\boldsymbol{\mu} \rangle) - \frac{1}{m}\sum_{r=1}^m \sigma(\langle \boldsymbol{w}_{-y,r}^{(t)}, y\boldsymbol{\mu} \rangle)\Big).$$

Recall the weight decomposition:

$$\boldsymbol{w}_{j,r}^{(t)} = \boldsymbol{w}_{j,r}^{(0)} + j\gamma_{j,r}^{(t)}\|\boldsymbol{\mu}\|_2^{-2}\boldsymbol{\mu} + \sum_{i=1}^n \overline{\rho}_{j,r,i}^{(t)}\|\boldsymbol{\xi}_i\|_2^{-2}\boldsymbol{\xi}_i + \sum_{i=1}^n \underline{\rho}_{j,r,i}^{(t)}\|\boldsymbol{\xi}_i\|_2^{-2}\boldsymbol{\xi}_i.$$

Then we conclude that:

$$\langle \boldsymbol{w}_{-y,r}^{(t)}, y\boldsymbol{\mu} \rangle = \langle \boldsymbol{w}_{-y,r}^{(0)}, y\boldsymbol{\mu} \rangle - \gamma_{-y,r}^{(t)},$$

$$\langle \boldsymbol{w}_{y,r}^{(t)}, y\boldsymbol{\mu} \rangle = \langle \boldsymbol{w}_{y,r}^{(0)}, y\boldsymbol{\mu} \rangle + \gamma_{y,r}^{(t)}.$$

First, we provide the bound for the signal learning part:

$$\frac{1}{m}\sum_{r=1}^m \sigma(\langle \boldsymbol{w}_{y,r}^{(t)}, y\boldsymbol{\mu} \rangle) - \frac{1}{m}\sum_{r=1}^m \sigma(\langle \boldsymbol{w}_{-y,r}^{(t)}, y\boldsymbol{\mu} \rangle)$$

$$\leq \frac{1}{m}\sum_{r=1}^m \sigma(\langle \boldsymbol{w}_{y,r}^{(t)}, y\boldsymbol{\mu} \rangle) = \frac{1}{m}\sum_{r=1}^m \sigma(\langle \boldsymbol{w}_{y,r}^{(0)}, y\boldsymbol{\mu} \rangle + \gamma_{y,r}^{(t)})$$

$$\leq (\langle \boldsymbol{w}_{y,r}^{(0)}, y\boldsymbol{\mu} \rangle + \gamma_{y,r}^{(t)})^2$$

$$\leq \tilde{O}(\sigma_0^2 \|\boldsymbol{\mu}\|_2^2),$$

where the first and second inequalities follow from the properties of the squared ReLU function, and the last inequality is by Lemma A.2 and Lemma 4.2.

Denote that $g(\boldsymbol{\xi}) = \frac{1}{m}j\sum_{j,r}\sigma(\langle \boldsymbol{w}_{j,r}^{(t)}, \boldsymbol{\xi} \rangle)$. It follows that:

$$\mathbb{P}(yf(\boldsymbol{W}^{(t)}, \boldsymbol{x}) < 0)$$

$$= \mathbb{P}\Big(\frac{1}{m}\sum_{r=1}^m \sigma(\langle \boldsymbol{w}_{-y,r}^{(t)}, \boldsymbol{\xi} \rangle) - \frac{1}{m}\sum_{r=1}^m \sigma(\langle \boldsymbol{w}_{y,r}^{(t)}, \boldsymbol{\xi} \rangle) \geq \frac{1}{m}\sum_{r=1}^m \sigma(\langle \boldsymbol{w}_{y,r}^{(t)}, y\boldsymbol{\mu} \rangle) - \frac{1}{m}\sum_{r=1}^m \sigma(\langle \boldsymbol{w}_{-y,r}^{(t)}, y\boldsymbol{\mu} \rangle)\Big)$$

$$\geq 0.5\mathbb{P}\Big(|g(\boldsymbol{\xi})| \geq \tilde{\Omega}(\sigma_0^2 \|\boldsymbol{\mu}\|_2^2)\Big).$$

Define the set $\mathcal{A} = \{\boldsymbol{\xi} : |g(\boldsymbol{\xi})| \geq \tilde{\Omega}(\sigma_0^2 \|\boldsymbol{\mu}\|_2^2)\}$. By Lemma C.6, we have:

$$\sum_{j \in \{\pm 1\}} [g(j\boldsymbol{\xi} + \boldsymbol{v}) - g(\boldsymbol{\xi})] \geq 4\tilde{\Omega}(\sigma_0^2 \|\boldsymbol{\mu}\|_2^2).$$

Thus, there must exist at least one of $\boldsymbol{\xi}, \boldsymbol{\xi} + \boldsymbol{v}, -\boldsymbol{\xi}$ and $-\boldsymbol{\xi} + \boldsymbol{v}$ that belongs to $\mathcal{A}$ and the probability is larger than 0.25. Furthermore, we have:

$$|\mathbb{P}(\mathcal{A}) - \mathbb{P}(\mathcal{A} - \boldsymbol{v})| = |\mathbb{P}_{\boldsymbol{\xi} \sim \mathcal{N}(\boldsymbol{0}, \sigma_p^2 \boldsymbol{I})}(\boldsymbol{\xi} \in \mathcal{A}) - \mathbb{P}_{\boldsymbol{\xi} \sim \mathcal{N}(\boldsymbol{v}, \sigma_p^2 \boldsymbol{I})}(\boldsymbol{\xi} \in \mathcal{A})|$$

$$\leq \frac{\|\boldsymbol{v}\|_2}{2\sigma_p} \leq 0.02,$$

where the first inequality is by Proposition 2.1 in Devroye et al. (2018) and the second inequality is by $\|\boldsymbol{v}\|_2 \leq 0.01\sigma_p$ according to Lemma C.6. Combined with that $\mathbb{P}(\mathcal{A}) = \mathbb{P}(-\mathcal{A})$, we finally achieve that $\mathbb{P}(\mathcal{A}) \geq 0.24$, corresponding to the second bullet result. Combined with Lemma 4.2, which establishes the first bullet point, this completes the proof of 3.1

$\square$

# D LABEL NOISE GD SUCCESSFULLY GENERALIZES WITH LOW SNR

## D.1 PROOF OF LEMMA 4.3

**Lemma D.1** (Lower bound on $\gamma_{j,r}^{(t)}$). *Under Assumption 3.1, during the first stage, where $0 \leq t \leq T_1 = \frac{nm \log(1/(\sigma_0 \sigma_p \sqrt{d}))}{\eta \sigma_p^2 d}$, there exists an lower bound for $\gamma_{j,r}^{(t)}$, for all $j$:*

$$\max_{r \in [m]} \gamma_{j,r}^{(t)} + |\langle \boldsymbol{w}_{j,r}^{(0)}, \boldsymbol{\mu} \rangle| \geq \exp\Big(\frac{C_0 \eta \|\boldsymbol{\mu}\|_2^2}{8m} t\Big) \max_{r \in [m]} |\langle \boldsymbol{w}_{j,r}^{(0)}, \boldsymbol{\mu} \rangle|.$$

*where $C_0$ is the lower bound on $|\tilde{\ell}'^{(t)}| \geq C_0$ is the first stage.*

*Proof of Lemma D.1.* If $\langle \boldsymbol{w}_{j,r}^{(t)}, \boldsymbol{\mu} \rangle \geq 0$, then

$$\gamma_{j,r}^{(t+1)} = \gamma_{j,r}^{(t)} - \frac{\eta}{nm} \sum_{i=1}^{n} \tilde{\ell}_i'^{(t)} \sigma'(\langle \boldsymbol{w}_{j,r}^{(t)}, y_i \boldsymbol{\mu} \rangle) \|\boldsymbol{\mu}\|_2^2 \epsilon_i^{(t)}$$

$$= \gamma_{j,r}^{(t)} - \frac{\eta}{nm} \Big[ \sum_{i \in \mathcal{S}_+^{(t)}} \tilde{\ell}_i'^{(t)} \sigma'(\langle \boldsymbol{w}_{j,r}^{(t)}, y_i \boldsymbol{\mu} \rangle) - \sum_{i \in \mathcal{S}_-^{(t)}} \tilde{\ell}_i'^{(t)} \sigma'(\langle \boldsymbol{w}_{j,r}^{(t)}, y_i \boldsymbol{\mu} \rangle) \Big] \|\boldsymbol{\mu}\|_2^2$$

$$= \gamma_{j,r}^{(t)} - \frac{2\eta}{nm} \Big[ \sum_{i \in \mathcal{S}_+^{(t)} \cap \mathcal{S}_1} \tilde{\ell}_i'^{(t)} - \sum_{i \in \mathcal{S}_-^{(t)} \cap \mathcal{S}_1} \tilde{\ell}_i'^{(t)} \Big] \langle \boldsymbol{w}_{j,r}^{(t)}, \boldsymbol{\mu} \rangle \|\boldsymbol{\mu}\|_2^2$$

$$\geq \gamma_{j,r}^{(t)} + \frac{2\eta}{nm} \big( C_0 |\mathcal{S}_+^{(t)} \cap \mathcal{S}_1| - |\mathcal{S}_-^{(t)} \cap \mathcal{S}_1| \big) \langle \boldsymbol{w}_{j,r}^{(t)}, \boldsymbol{\mu} \rangle \|\boldsymbol{\mu}\|_2^2.$$

Note that we have defined $\mathcal{S}_\pm^{(t)} = \{i : \epsilon_i^{(t)} = \pm 1\}$ and $\mathcal{S}_j = \{i : y_i = j\}$ in Lemma A.3.

On the other hand, when $\langle \boldsymbol{w}_{j,r}^{(t)}, \boldsymbol{\mu} \rangle < 0$,

$$\gamma_{j,r}^{(t+1)} = \gamma_{j,r}^{(t)} - \frac{2\eta}{nm} \Big[ \sum_{i \in \mathcal{S}_+^{(t)} \cap \mathcal{S}_{-1}} \tilde{\ell}_i'^{(t)} - \sum_{i \in \mathcal{S}_-^{(t)} \cap \mathcal{S}_{-1}} \tilde{\ell}_i'^{(t)} \Big] \langle -\boldsymbol{w}_{j,r}^{(t)}, \boldsymbol{\mu} \rangle \|\boldsymbol{\mu}\|_2^2$$

$$\geq \gamma_{j,r}^{(t)} + \frac{2\eta}{nm} \big( C_0 |\mathcal{S}_+^{(t)} \cap \mathcal{S}_{-1}| - |\mathcal{S}_-^{(t)} \cap \mathcal{S}_{-1}| \big) \langle -\boldsymbol{w}_{j,r}^{(t)}, \boldsymbol{\mu} \rangle \|\boldsymbol{\mu}\|_2^2.$$

By Lemma A.3, we have

$$\frac{|\mathcal{S}_+^{(t)} \cap \mathcal{S}_1|}{|\mathcal{S}_-^{(t)} \cap \mathcal{S}_1|}, \frac{|\mathcal{S}_+^{(t)} \cap \mathcal{S}_{-1}|}{|\mathcal{S}_-^{(t)} \cap \mathcal{S}_{-1}|} \geq \frac{(1-p)n - \sqrt{2n \log(8T^*/\delta)}}{pn + \sqrt{2n \log(8T^*/\delta)}},$$

$$|\mathcal{S}_+^{(t)} \cap \mathcal{S}_1|, |\mathcal{S}_+^{(t)} \cap \mathcal{S}_{-1}| \geq (1-p)n - \sqrt{2n \log(8T^*/\delta)}.$$

These hold with probability at least $1 - \delta$. This suggests that when $p < C_0/6, n \geq 72C_0^{-2} \log(8T^*/\delta)$, we have:

$$|\mathcal{S}_+^{(t)} \cap \mathcal{S}_1| \geq \frac{2}{C_0} |\mathcal{S}_-^{(t)} \cap \mathcal{S}_1|, \quad |\mathcal{S}_+^{(t)} \cap \mathcal{S}_{-1}| \geq \frac{2}{C_0} |\mathcal{S}_-^{(t)} \cap \mathcal{S}_{-1}|,$$

$$|\mathcal{S}_+^{(t)} \cap \mathcal{S}_1|, |\mathcal{S}_+^{(t)} \cap \mathcal{S}_{-1}| \geq \frac{n}{4}.$$

Hence, we have:

$$\gamma_{j,r}^{(t+1)} \geq \gamma_{j,r}^{(t)} + \frac{C_0\eta\|\boldsymbol{\mu}\|_2^2}{4m}\langle \boldsymbol{w}_{j,r}^{(t)}, \boldsymbol{\mu}\rangle = \gamma_{j,r}^{(t)} + \frac{C_0\eta\|\boldsymbol{\mu}\|_2^2}{4m}\big(\langle \boldsymbol{w}_{j,r}^{(0)}, \boldsymbol{\mu}\rangle + j\gamma_{j,r}^{(t)}\big), \quad \text{if } \langle \boldsymbol{w}_{j,r}^{(t)}, \boldsymbol{\mu}\rangle \geq 0$$

$$\gamma_{j,r}^{(t+1)} \geq \gamma_{j,r}^{(t)} - \frac{C_0\eta\|\boldsymbol{\mu}\|_2^2}{4m}\langle \boldsymbol{w}_{j,r}^{(t)}, \boldsymbol{\mu}\rangle = \gamma_{j,r}^{(t)} - \frac{C_0\eta\|\boldsymbol{\mu}\|_2^2}{4m}\big(\langle \boldsymbol{w}_{j,r}^{(0)}, \boldsymbol{\mu}\rangle + j\gamma_{j,r}^{(t)}\big), \quad \text{if } \langle \boldsymbol{w}_{j,r}^{(t)}, \boldsymbol{\mu}\rangle < 0.$$

When $j = 1$, due to the increase of $\gamma_{j,r}^{(t)}$, we have

$$\gamma_{1,r}^{(t+1)} \geq \gamma_{1,r}^{(t)} + \frac{C_0\eta\|\boldsymbol{\mu}\|_2^2}{4m}\big(\langle \boldsymbol{w}_{1,r}^{(0)}, \boldsymbol{\mu}\rangle + \gamma_{1,r}^{(t)}\big).$$

Let $B_j^{(t)} = \max_{r\in[m]}\{\gamma_{j,r}^{(t)} + j\langle \boldsymbol{w}_{j,r}^{(0)}, \boldsymbol{\mu}\rangle\}$, then we have

$$B_1^{t+1} \geq \big(1 + \frac{C_0\eta\|\boldsymbol{\mu}\|_2^2}{4m}\big)B_1^{(t)} \geq \big(1 + \frac{C_0\eta\|\boldsymbol{\mu}\|_2^2}{4m}\big)^{(t)}B_1^{(0)}$$

$$\geq \exp\big(\frac{C_0\eta\|\boldsymbol{\mu}\|_2^2}{8m}t\big)\max_r\langle \boldsymbol{w}_{1,r}^{(0)}, \boldsymbol{\mu}\rangle$$

$$\geq \exp\big(\frac{C_0\eta\|\boldsymbol{\mu}\|_2^2}{8m}t\big)\frac{\sigma_0\|\boldsymbol{\mu}\|_2}{2},$$

where we use the fact that $1 + x \geq \exp(x/2)$ for $x \leq 2$.

Similarly when $j = -1$, we have $\gamma_{-1,r}^{(t+1)} \geq \gamma_{-1,r}^{(t)} - \frac{C_0\eta\|\boldsymbol{\mu}\|_2^2}{4m}\big(\langle \boldsymbol{w}_{-1,r}^{(0)}, \boldsymbol{\mu}\rangle - \gamma_{-1,r}^{(t)}\big)$ and

$$B_{-1}^{(t+1)} \geq \big(1 + \frac{C_0\eta\|\boldsymbol{\mu}\|_2^2}{4m}\big)B_{-1}^{(t)} \geq \big(1 + \frac{C_0\eta\|\boldsymbol{\mu}\|_2^2}{4m}\big)^{(t)}B_{-1}^{(0)}$$

$$\geq \exp\big(\frac{C_0\eta\|\boldsymbol{\mu}\|_2^2}{8m}t\big)\max_r\langle -\boldsymbol{w}_{-1,r}^{(0)}, \boldsymbol{\mu}\rangle$$

$$\geq \exp\big(\frac{C_0\eta\|\boldsymbol{\mu}\|_2^2}{8m}t\big)\frac{\sigma_0\|\boldsymbol{\mu}\|_2}{2}.$$

Thus, we obtain $B_j^{(t)} \geq \exp\big(\frac{C_0\eta\|\boldsymbol{\mu}\|_2^2}{8m}t\big)\frac{\sigma_0\|\boldsymbol{\mu}\|_2}{2}, \forall j \in \{\pm 1\}$. □

**Lemma D.2.** *Let* $\bar{\beta} = \min_{i\in[n]}\max_{r\in[m]}\langle \boldsymbol{w}_{y_i,r}^{(0)}, \boldsymbol{\xi}_i\rangle$. *Suppose that* $\sigma_0 \geq 160n\sqrt{\frac{\log(4n^2/\delta)}{d}}(\sigma_p\sqrt{d})^{-1}\alpha d^{1/4}$. *Then we have that* $\bar{\beta}/d^{1/4} \geq 40n\sqrt{\frac{\log(4n^2/\delta)}{d}}\alpha$.

*Proof of Lemma D.2.* The proof follows from Lemma A.2. It is known that, with high probability, we have $\bar{\beta} \geq \sigma_0\sigma_p\sqrt{d}/4$. By substituting the condition for $\sigma_0$, we obtain

$$\bar{\beta}/d^{1/4} \geq 40n\sqrt{\frac{\log(4n^2/\delta)}{d}}\alpha.$$

□

**Lemma D.3** (Lower bound on $\bar{\rho}_{j,r,i}^{(t)}$)**.** *Let* $\bar{\beta} = \min_{i\in[n]}\max_{r\in[m]}\langle \boldsymbol{w}_{y_i,r}^{(0)}, \boldsymbol{\xi}_i\rangle$ *and* $A_{y_i,r,i}^{(t)} := \bar{\rho}_{j,r,i}^t + \langle \boldsymbol{w}_{j,r}^{(0)}, \boldsymbol{\xi}_i\rangle - 0.4\bar{\beta}/d^{1/4}$. *Under Assumption 3.1, if* $\langle \boldsymbol{w}_{j,r}^{(0)}, \boldsymbol{\xi}_i\rangle \geq \bar{\beta}$, *then at time step* $T_1 = \frac{nm\log(1/(\sigma_0\sigma_p\sqrt{d}))}{\eta\sigma_p^2 d}$, *with high probability, it holds that*

$$A_{y_i,r,i}^{(T_1)} \geq (1 + \frac{\eta C_0\sigma_p^2 d}{2nm})^{T_1}A_{y_i,r,i}^{(0)}.$$

*Proof of Lemma D.3.* First, consider $y_i = j$ as the case of $\bar{\rho}_{j,r,i}^{(t)}$. By Lemma B.1 and Lemma C.3, when $y_i = j$,

$$|\langle \boldsymbol{w}_{j,r}^{(t)}, \boldsymbol{\xi}_i\rangle - \langle \boldsymbol{w}_{j,r}^{(0)}, \boldsymbol{\xi}_i\rangle - \bar{\rho}_{j,r,i}^{(t)}| \leq 16n\sqrt{\frac{\log(4n^2/\delta)}{d}} \leq 0.4\bar{\beta}/d^{1/4}. \tag{10}$$

From the update of $\bar{\rho}_{j,r,i}^{(t)}$, when $\epsilon_i^{(t)} = 1$ and $\langle \boldsymbol{w}_{j,r}^{(t)}, \boldsymbol{\xi}_i \rangle > 0$,

$$\bar{\rho}_{j,r,i}^{(t+1)} = \bar{\rho}_{j,r,i}^{(t)} - \frac{2\eta}{nm} \tilde{\ell}_i^{\prime(t)} \langle \boldsymbol{w}_{j,r}^{(t)}, \boldsymbol{\xi}_i \rangle \|\boldsymbol{\xi}_i\|_2^2 \epsilon_i^{(t)} \geq \bar{\rho}_{j,r,i}^{(t)} + \frac{\eta C_0 \sigma_p^2 d}{nm} \left( \bar{\rho}_{j,r,i}^{(t)} + \langle \boldsymbol{w}_{j,r}^{(0)}, \boldsymbol{\xi}_i \rangle - 0.4\bar{\beta}/d^{1/4} \right),$$

On the other hand, when $\epsilon_i^{(t)} = -1$ and $\langle \boldsymbol{w}_{j,r}^{(t)}, \boldsymbol{\xi}_i \rangle > 0$,

$$\bar{\rho}_{j,r,i}^{(t+1)} = \bar{\rho}_{j,r,i}^{(t)} - \frac{2\eta}{nm} \tilde{\ell}_i^{\prime(t)} \langle \boldsymbol{w}_{j,r}^{(t)}, \boldsymbol{\xi}_i \rangle \|\boldsymbol{\xi}_i\|_2^2 \epsilon_i^{(t)} \geq \bar{\rho}_{j,r,i}^{(t)} - \frac{3\eta\sigma_p^2 d}{nm} \left( \bar{\rho}_{j,r,i}^{(t)} + \langle \boldsymbol{w}_{j,r}^{(0)}, \boldsymbol{\xi}_i \rangle + 0.4\bar{\beta}/d^{1/4} \right).$$

For simplification of notations, denote $\zeta = 0.8\bar{\beta}/d^{1/4}$. Let $A_{y_i,r,i}^{(t)} := \bar{\rho}_{j,r,i}^t + \langle \boldsymbol{w}_{j,r}^{(0)}, \boldsymbol{\xi}_i \rangle - 0.4\bar{\beta}/d^{1/4}$. Then when $\epsilon_i^{(t)} = 1$, we have

$$A_{y_i,r,i}^{(t+1)} \geq (1 + \frac{\eta C_0 \sigma_p^2 d}{nm}) A_{y_i,r,i}^{(t)},$$

and when $\epsilon_i^{(t)} = -1$, we have

$$A_{y_i,r,i}^{(t+1)} \geq (1 - \frac{3\eta\sigma_p^2 d}{nm}) A_{y_i,r,i}^{(t)} - \frac{3\eta\sigma_p^2 d\zeta}{nm}.$$

Here we prove when $\langle \boldsymbol{w}_{j,r}^{(0)}, \boldsymbol{\xi}_i \rangle \geq \bar{\beta}$, $A_{y_i,r,i}^{(t)} > \zeta$. The proof is by the induction method.

First it is clear that $A_{y_i,r,i}^{(0)} = \langle \boldsymbol{w}_{j,r}^{(0)}, \boldsymbol{\xi}_i \rangle - 0.5\zeta > \zeta$ because $d \gg \Theta(1)$. Then we consider when $t \leq \frac{2\log(4n/\delta)}{p^2}$ (where the condition for Lemma A.4 does not hold). In this case, $|\mathcal{S}_+^{(t)}| \geq (1-p)t - \sqrt{\frac{t}{2}\log(\frac{4n}{\delta})}, |\mathcal{S}_-^{(t)}| \leq pt + \sqrt{\frac{t}{2}\log(\frac{4n}{\delta})}$. In addition, the worst case lower bound is achieved by the case where all the $\mathcal{S}_-^{(t)}$ events happen at the first few iterations. This gives

$$A_{y_i,r,i}^{(t)} \geq (1 + \frac{\eta C_0 \sigma_p^2 d}{nm})^{(1-p)t - \sqrt{\frac{t}{2}\log(\frac{4n}{\delta})}} (1 - \frac{3\eta\sigma_p^2 d}{nm})^{pt + \sqrt{\frac{t}{2}\log(\frac{4n}{\delta})}} A_{y_i,r,i}^{(0)}$$

$$- (1 + \frac{\eta C_0 \sigma_p^2 d}{nm})^{(1-p)t - \sqrt{\frac{t}{2}\log(\frac{4n}{\delta})}} \left[ \sum_{s=0}^{pt + \sqrt{\frac{t}{2}\log(\frac{4n}{\delta})}} \left( 1 - \frac{3\eta\sigma_p^2 d}{nm} \right)^s \right] \frac{\zeta\eta\sigma_p^2 d}{3nm}$$

$$\geq (1 + \frac{\eta C_0 \sigma_p^2 d}{nm})^{(1-p)t - \sqrt{\frac{t}{2}\log(\frac{4n}{\delta})}} \left( (1 - \frac{3\eta\sigma_p^2 d}{nm})^{pt + \sqrt{\frac{t}{2}\log(\frac{4n}{\delta})}} A_{y_i,r,i}^{(0)} - \zeta \right)$$

$$\geq (1 + \frac{\eta C_0 \sigma_p^2 d}{nm})^{(1-p)t - \sqrt{\frac{t}{2}\log(\frac{4}{\delta})}} \zeta \geq \zeta,$$

where the last inequality follows from the fact that $d \gg \Theta(1)$. To see this, suppose there exists a $t \leq \frac{2\log(4n/\delta)}{p^2}$ such that

$$(1 - \frac{3\eta\sigma_p^2 d}{nm})^{pt + \sqrt{\frac{t}{2}\log(\frac{4n}{\delta})}} A_{y_i,r,i}^{(0)} \leq 2\zeta,$$

then we have

$$pt + \sqrt{\frac{t}{2}\log(\frac{4n}{\delta})} \geq \frac{\log(d^{1/4}/2)}{\log\left( \frac{1}{1 - \frac{3\eta\sigma_p^2 d}{nm}} \right)},$$

while $t \leq \frac{2\log(4/\delta)}{p^2}$ raises a contradiction by the choice of $d$. This proves for all $t \leq \frac{2\log(4/\delta)}{p^2}$, we have $(1 - \frac{3\eta\sigma_p^2 d}{nm})^{pt + \sqrt{\frac{t}{2}\log(\frac{4n}{\delta})}} A_{y_i,r,i}^{(0)} \geq 2\zeta$ and thus $A_{y_i,r,i}^{(t)} \geq \zeta$.

Then we consider the case when $t \geq \frac{2\log(4/\delta)}{p^2}$ where the condition for Lemma A.4 holds. Now suppose for all $s \leq t - 1$, we have $A_{y_i,r,i}^{(s)} \geq \zeta$, which clearly holds for $t = \frac{2\log(4n/\delta)}{p^2}$. For all

$s \leq t - 1$, we have $A_{y_i,r,i}^{(s)} \geq (1 - \frac{3\eta\sigma_p^2 d\zeta}{nm})A_{y_i,r,i}^{(s)}$ when $\epsilon_i^{(s)} = -1$. This leads to the following lower bound for $A_{y_i,r,i}^{(t)}$ as

$$A_{y_i,r,i}^{(t)} \geq (1 + \frac{\eta C_0 \sigma_p^2 d}{nm})^{(1-1.5p)t}(1 - \frac{3\eta\sigma_p^2 d}{nm})^{1.5pt} A_{y_i,r,i}^{(0)}$$

$$\geq (1 + \frac{\eta C_0 \sigma_p^2 d}{2nm})^t A_{y_i,r,i}^{(0)} \geq \zeta,$$

where the second last inequality follows from the choice of

$$p \leq \frac{2}{3} \frac{\log(1 + \frac{\eta C_0 \sigma_p^2 d}{nm}) - \log(1 + \frac{\eta C_0 \sigma_p^2 d}{2nm})}{\log(1 + \frac{\eta C_0 \sigma_p^2 d}{nm}) - \log(1 - \frac{3\eta\sigma_p^2 d}{nm})}.$$

We can verify that $p = \frac{C_0}{24}$ satisfies the above inequality. This concludes the proof that, for all $t$, we have $A_{y_i,r,i}^{(t)} \geq \zeta$ and thus for all $t$. Finally, we conclude that

$$A_{y_i,r,i}^{(t)} \geq (1 + \frac{\eta C_0 \sigma_p^2 d}{nm})^{(1-1.5p)t}(1 - \frac{2\eta\sigma_p^2 d}{3nm})^{1.5pt} A_{y_i,r,i}^{(0)}$$

$$\geq (1 + \frac{\eta C_0 \sigma_p^2 d}{2nm})^t A_{y_i,r,i}^{(0)}.$$

$\square$

With the above lemmas at hand, we are ready to prove Lemma 4.3:

*Proof of Lemma 4.3.* By Lemma D.3, at $t = T_1$, taking the maximum over $r$ yields

$$\max_r A_{y_i,r,i}^{(t)} \geq (1 + \frac{\eta C_0 \sigma_p^2 d}{2nm})^t 0.6\bar{\beta}$$

$$\geq (1 + \frac{\eta C_0 \sigma_p^2 d}{2nm})^t 0.15\sigma_0\sigma_p\sqrt{d}$$

$$\geq \exp\big(\frac{\eta C_0 \sigma_p^2 d}{4nm}t\big)0.15\sigma_0\sigma_p\sqrt{d},$$

where the first inequality is by $\max_r \langle w_{j,r}^{(0)}, \xi_i \rangle \geq \bar{\beta}$ and $0.4\bar{\beta}d^{-1/4} \leq 0.4\bar{\beta}$. In the last inequality, we use $(1 + z) \geq \exp(z/2)$ for $z \leq 2$.

Then we see $\max_r A_{y_i,r,i}^{(t)} \geq 1$ in at least $T_1 = \frac{\log(20/(\sigma_0\sigma_p\sqrt{d}))4nm}{\eta C_0 \sigma_p^2 d}$ and because $\max_{j,r} \bar{\rho}_{j,r,i}^{T_1} \geq A_{y_i,r,i}^{T_1} - \max_{j,r} |\langle w_{j,r}^{(0)}, \xi_i \rangle| + 0.4\bar{\beta} \geq 1$.

Besides, by Lemma C.2, we directly obtain the result that

$$|\underline{\rho}_{j,r,i}^{(T_1)}| \leq \frac{3\eta\sigma_p^2 dT_1}{nm}\sqrt{\log(8mn/\delta)}\sigma_0\sigma_p\sqrt{d} = \tilde{O}(\sigma_0\sigma_p\sqrt{d}).$$

Furthermore, Lemma C.1 yields

$$\gamma_{j,r}^{(T_1)} + |\langle w_{j,r}^{(0)}, \mu \rangle| \leq \exp\big(\frac{2\eta\|\mu\|_2^2}{m}\frac{4nm}{\eta\sigma_p^2 d}\log(1/(\sigma_0\sigma_p\sqrt{d}))\big)|\langle w_{j,r}^{(0)}, \mu \rangle| \leq 2|\langle w_{j,r}^{(0)}, \mu \rangle|,$$

where we have used the condition of low SNR, namely $n\text{SNR}^2 \leq 1/\log(20/(\sigma_0\sigma_p\sqrt{d}))$. By Lemma A.2, we conclude the proof for $\max_{j,r} \gamma_{j,r}^{(T_1)} = \tilde{O}(\sigma_0\|\mu\|_2)$.

Lastly, according to Lemma D.1, at the end of stage1, we have the lower bound on signal learning coefficient

$$\max_{r \in [m]} \gamma_{j,r}^{(t)} + |\langle w_{j,r}^{(0)}, \mu \rangle| \geq \exp\big(\frac{C_0\eta\|\mu\|_2^2}{8m}t\big) \max_{r \in [m]} |\langle w_{j,r}^{(0)}, \mu \rangle|$$

$$= \exp\big(\frac{C_0\eta\|\mu\|_2^2}{8m}\frac{\log(20/(\sigma_0\sigma_p\sqrt{d}))4nm}{\eta C_0 \sigma_p^2 d}\big) \max_{r \in [m]} |\langle w_{j,r}^{(0)}, \mu \rangle|$$

$$\geq \exp(n\text{SNR}^2\log(20/(\sigma_0\sigma_p\sqrt{d}))\sigma_0\|\mu\|_2 \geq \sigma_0\|\mu\|_2.$$

$\square$

### D.2 PROOF OF LEMMA 4.4

The key idea is to show $\overline{\rho}_{j,r,i}^{(t)}$ oscillates during the second stage, where the growth tends to offset the drop over a given time frame. This would suggest the $f(\boldsymbol{W}^{(t)}, \boldsymbol{x})$ is both upper and lower bounded by a constant, which is crucial to ensuring that $\gamma_{j,r}^{(t)}$ increases exponentially during the second stage.

Without loss of generality, for each $i$ with $\langle \boldsymbol{w}_{j,r,i}^{(t)}, \boldsymbol{\xi}_i \rangle > 0$ and $j = y_i = 1$, the evolution of $\overline{\rho}_{j,r,i}^{t+1}$ is written as

$$\overline{\rho}_{j,r,i}^{t+1} = \overline{\rho}_{j,r,i}^{(t)} - \frac{2\eta}{nm}\tilde{\ell}_i^{'(t)}\langle \boldsymbol{w}_{j,r}^{(t)}, \boldsymbol{\xi}_i \rangle \|\boldsymbol{\xi}_i\|^2 \epsilon_i^{(t)}$$

$$\approx \begin{cases} (1 + \frac{2\eta\|\boldsymbol{\xi}_i\|^2}{nm(1+\exp(f_i^{(t)}))})\overline{\rho}_{j,r,i}^{(t)}, & \text{if } \epsilon_i^{(t)} = 1 \\ (1 - \frac{2\eta\|\boldsymbol{\xi}_i\|^2}{nm(1+\exp(-f_i^{(t)}))})\overline{\rho}_{j,r,i}^{(t)} & \text{if } \epsilon_i^{(t)} = -1 \end{cases}$$

where we denote $f_i^{(t)} = f(\boldsymbol{W}^{(t)}, \boldsymbol{x}_i)$. Note that $f_i^{(t)} \approx \frac{1}{m}\sum_{r=1}^m (\overline{\rho}_{+1,r,i}^{(t)})^2$ when $\gamma_{j,r}^{(t)} \ll 1$.

To simplify the notation, we define that $\iota_i^{(t)} \triangleq \frac{1}{m}\sum_{r=1}^m \overline{\rho}_{+1,r,i}^{(t)}$. Then the dynamics can be approximated to

$$\iota_i^{(t+1)} \approx \begin{cases} (1 + \frac{2\eta\|\boldsymbol{\xi}\|_2^2}{nm(1+\exp((\iota_i^{(t)})^2))})\iota_i^{(t)} & \text{with prob } 1 - p \\ (1 - \frac{2\eta\|\boldsymbol{\xi}\|_2^2}{nm(1+\exp(-(\iota_i^{(t)})^2))})\iota_i^{(t)} & \text{with prob } p \end{cases}$$

**Lemma D.4** (Restatement of Lemma 4.4). *Under the same condition as Theorem 3.2, during $t \in [T_1, T_2]$ with $T_2 = T_1 + \log(6/(\sigma_0\|\boldsymbol{\mu}\|_2))4m(1 + \exp(c_2))\eta^{-1}\|\boldsymbol{\mu}\|_2^{-2}$, there exist a sufficient large positive constant $C_\iota$ and a constant $\iota_i^*$ depending on sample index $i$ such that the following results hold with high probability at least $1 - 1/d$:*

- $|\iota_i^{(t)} - \iota_i^*| \leq C_\iota$

- $\gamma_{j,r}^{(t)} \leq 0.1$ *for all* $j \in \{-1, 1\}$ *and* $r \in [m]$

- $\frac{1}{2m}(\sum_{r=1}^m \overline{\rho}_{y_i,r,i}^{(t)})^2 \leq f_i^{(t)} \leq \frac{2}{m}(\sum_{r=1}^m \overline{\rho}_{y_i,r,i}^{(t)})^2$

- $\max_{r \in [m]}(\gamma_{j,r}^{(t)} + |\langle \boldsymbol{w}_{j,r}^{(0)}, \boldsymbol{\mu}\rangle|) \geq \exp\left(\frac{\eta\|\boldsymbol{\mu}\|_2^2}{16m}(t - T_1)\right)\max_{r \in [m]}|\gamma_{j,r}^{(T_1)} + \langle \boldsymbol{w}_{j,r}^{(0)}, \boldsymbol{\mu}\rangle|$.

*Proof of Lemma D.4.* The proof is based on the method of induction. Without loss of generality, we consider all $i$ with $y_i = 1$. We first check that at time step $t = T_1$, by Lemma 4.3, there exists a constant $C$ such that

$$\left|\frac{1}{m}\sum_{r=1}^m \overline{\rho}_{+1,r,i}^{(T_1)} - \iota_i^*\right| \leq C.$$

Besides, by Lemma 3.2, it is straightforward to check that $\gamma_{j,r}^{(T_1)} \leq 1$ for all $j \in \{-1, 1\}$ and $r \in [m]$, and $\max_j \gamma_{j,r}^{(T_1)} \geq 0$. Next, we can show the following result at time $t = T_1$:

$$f_i^{(T_1)} = F_{+1}(\boldsymbol{W}_{+1}^{(T_1)}, \boldsymbol{x}_i) - F_{-1}(\boldsymbol{W}_{-1}^{(T_1)}, \boldsymbol{x}_i)$$

$$= \frac{1}{m}\sum_{r=1}^m \sigma\big(\langle \boldsymbol{w}_{+1,r}^{(0)}, \boldsymbol{\mu}\rangle + \gamma_{+1,r}^{(T_1)}\big) + \frac{1}{m}\sum_{r=1}^m \sigma\Big(\langle \boldsymbol{w}_{+1,r}^{(0)}, \boldsymbol{\xi}_i\rangle + \overline{\rho}_{+1,r,i}^{(T_1)} + \sum_{i' \neq i}\frac{\langle \boldsymbol{\xi}_i, \boldsymbol{\xi}_{i'}\rangle}{\|\boldsymbol{\xi}_{i'}\|_2^2}\rho_{+1,r,i'}^{(T_1)}\Big)$$

$$- \frac{1}{m}\sum_{r=1}^m \sigma\big(\langle \boldsymbol{w}_{-1,r}^{(0)}, \boldsymbol{\mu}\rangle - \gamma_{-1,r}^{(T_1)}\big) - \frac{1}{m}\sum_{r=1}^m \sigma\Big(\langle \boldsymbol{w}_{-1,r}^{(0)}, \boldsymbol{\xi}_i\rangle + \underline{\rho}_{-1,r,i}^{(T_1)} + \sum_{i' \neq i}\frac{\langle \boldsymbol{\xi}_i, \boldsymbol{\xi}_{i'}\rangle}{\|\boldsymbol{\xi}_{i'}\|_2^2}\rho_{-1,r,i'}^{(T_1)}\Big)$$

$$\geq -\tilde{\Omega}(\sigma_0^2\|\boldsymbol{\mu}\|_2^2) - \tilde{\Omega}(\sigma_0\sigma_p\sqrt{d}) + \frac{1}{m}(\sum_{r=1}^m \overline{\rho}_{+1,r,i}^{T_1)} - \beta - 16\sqrt{\frac{\log(4n^2/\delta)}{d}}n\alpha)^2$$

$$\geq \frac{1}{2m}(\sum_{r=1}^m \overline{\rho}_{+1,r,i}^{(T_1)})^2,$$

where the first inequality is by Lemma 4.4, Proposition 4.1, and Lemma A.1, The second inequality follows from the condition on $\sigma_0$ and $d$ in Assumption 3.1. Similarly, we have

$$
\begin{aligned}
f_i^{(T_1)} &= F_{+1}(\boldsymbol{W}_{+1}^{(T_1)}, \boldsymbol{x}_i) - F_{-1}(\boldsymbol{W}_{-1}^{(T_1)}, \boldsymbol{x}_i) \\
&\leq \tilde{O}(\sigma_0^2 \|\boldsymbol{\mu}\|_2^2) + \tilde{O}(\sigma_0 \sigma_p \sqrt{d}) + \frac{1}{m}\big(\sum_{r=1}^{m} \overline{\rho}_{+1,r,i}^{(T_1)} + \beta + 16\sqrt{\frac{\log(4n^2/\delta)}{d}} n\alpha\big)^2 \\
&\leq \frac{2}{m}\big(\sum_{r=1}^{m} \overline{\rho}_{+1,r,i}^{(T_1)}\big)^2.
\end{aligned}
$$

Next, we assume that all the results hold for $T_1 < t \leq T$. By the induction hypothesis, we can bound $c_1 \leq f_i^{(T)} \leq c_2$ for all $i \in [n]$. Then we can show that $\gamma_{j,r}^{(T+1)}$ continues to exhibit exponential growth:

$$
\begin{aligned}
\gamma_{j,r}^{(T+1)} &= \gamma_{j,r}^{(T)} - \frac{2\eta}{nm}\big(\sum_{i \in \mathcal{S}_+^{(T)} \cap \mathcal{S}_1} \tilde{\ell}_i^{'(t)} - \sum_{i \in \mathcal{S}_-^{(T)} \cap \mathcal{S}_1} \tilde{\ell}_i^{'(t)}\big)\langle \boldsymbol{w}_{j,r}^{(T)}, \boldsymbol{\mu}\rangle \|\boldsymbol{\mu}\|_2^2 \\
&\geq \gamma_{j,r}^{(T)} + \frac{2\eta}{nm}\big(|\mathcal{S}_+^{(T)}|\frac{1}{1+\exp(c_2)} - |\mathcal{S}_-^{(T)}|\frac{1}{1+\exp(-c_2)}\big)\langle \boldsymbol{w}_{j,r}^{(T)}, \boldsymbol{\mu}\rangle \|\boldsymbol{\mu}\|_2^2 \\
&\geq \gamma_{j,r}^{(T)} + \frac{2\eta}{m}\big(\frac{2-3p}{4}\frac{1}{1+\exp(c_2)} - \frac{3p}{4}\frac{1}{1+\exp(-c_2)}\big)\langle \boldsymbol{w}_{j,r}^{(T)}, \boldsymbol{\mu}\rangle \|\boldsymbol{\mu}\|_2^2 \\
&= \gamma_{j,r}^{(T)} + \frac{\eta}{m}\big(\frac{1}{1+\exp(c_2)} - \frac{3p}{2}\big)(\langle \boldsymbol{w}_{j,r}^{(T)}, \boldsymbol{\mu}\rangle + j\gamma_{j,r}^{(T)})\|\boldsymbol{\mu}\|_2^2 \\
&\geq \gamma_{j,r}^{(T)} + \frac{\eta\|\boldsymbol{\mu}\|_2^2}{2m(1+\exp(c_2))}(\langle \boldsymbol{w}_{j,r}^{(T)}, \boldsymbol{\mu}\rangle + j\gamma_{j,r}^{(T)}),
\end{aligned}
$$

where the last inequality is by $\frac{3}{2}p \leq \frac{1}{2}\frac{1}{1+\exp(c_2)}$. Next, define $B^{(t)} = \max_{r \in [m]}(\gamma_{j,r}^{(t)} + |\langle \boldsymbol{w}_{j,r}^{(0)}, \boldsymbol{\mu}\rangle|)$, we have:

$$
\begin{aligned}
B^{(T+1)} &\geq B^{(T)}(1 + \frac{\eta\|\boldsymbol{\mu}\|_2^2}{2m(1+\exp(c_2))}) \\
&\geq \exp(\frac{\eta\|\boldsymbol{\mu}\|_2^2}{4m(1+\exp(c_2))}(t-T_1))B^{(T_1)} \\
&\geq \exp\big(\frac{\eta\|\boldsymbol{\mu}\|_2^2}{16m}(t-T_1)\big)B^{(T_1)}.
\end{aligned}
$$

At the same time, there exists an upper bound on the signal learning:

$$
\gamma_{j,r}^{(T)} + |\langle \boldsymbol{w}_{j,r}^{(0)}, \boldsymbol{\mu}\rangle| \leq \exp\big(\frac{2\eta\|\boldsymbol{\mu}\|_2^2}{m}(T-T_1)\big)|\gamma_{j,r}^{(T_1)} + \langle \boldsymbol{w}_{j,r}^{(0)}, \boldsymbol{\mu}\rangle| \leq 0.01,
$$

where we used the condition that $T < T_2$.

To show that $\iota_i^{(T+1)}$ remains within a constant range, we define $M_i^{(t)} \triangleq (\iota_i^{(t)} - \iota_i^*)^2$ where $\iota_i^*$ is a sufficiently large constant depending on $i$. Using the relation $\frac{1}{2m}(\sum_{r=1}^{m} \overline{\rho}_{y_i,r,i}^{(T)})^2 \leq f_i^{(T)} \leq \frac{2}{m}(\sum_{r=1}^{m} \overline{\rho}_{y_i,r,i}^{(T)})^2$ we have:

$$
\begin{aligned}
\mathbb{E}[\iota_i^{(T+1)}|\iota_i^{(T)}] &\geq (1-p)\Big(1 + \frac{2\eta\|\boldsymbol{\xi}_i\|_2^2}{(1+2\exp((\iota_i^{(T)})^2))nm}\Big)\iota_i^{(T)} \\
&\quad + p\Big(1 - \frac{2\eta\|\boldsymbol{\xi}_i\|_2^2}{(1+1/2\exp(-(\iota_i^{(T)})^2))nm}\Big)\iota_i^{(T)}.
\end{aligned}
$$

At the same time,

$$
\mathbb{E}[(\iota_i^{(T+1)})^2|\iota_i^{(T)}] \leq (1-p)\Big(1 + \frac{2\eta\|\boldsymbol{\xi}_i\|_2^2}{(1+1/2\exp((\iota_i^{(T)})^2))nm}\Big)^2 (\iota_i^{(T)})^2
$$

$$+ p\Big(1 - \frac{2\eta\|\boldsymbol{\xi}_i\|_2^2}{(1 + 2\exp(-(\iota_i^{(T)})^2))nm}\Big)^2 (\iota_i^{(T)})^2.$$

Then we show that

$$\mathbb{E}[M_i^{(T+1)}|\iota_i^{(T)}] = \mathbb{E}[(\iota_i^{(T+1)})^2|\iota_i^{(T)}] - 2\iota^*\mathbb{E}[\iota_i^{(T+1)}|\iota_i^{(T)}] + (\iota^*)^2$$

$$\leq (1-p)\Big(1 + \frac{2\eta\|\boldsymbol{\xi}_i\|_2^2}{(1 + 1/2\exp((\iota_i^{(T)})^2))nm}\Big)^2 (\iota_i^{(T)})^2$$

$$+ p\Big(1 - \frac{2\eta\|\boldsymbol{\xi}_i\|_2^2}{(1 + 2\exp(-(\iota_i^{(T)})^2))nm}\Big)^2 (\iota_i^{(T)})^2$$

$$- 2\iota^*(\iota_i^{(T)} + \frac{2\eta\|\boldsymbol{\xi}_i\|^2}{nm}\Big(\frac{1}{1 + 2\exp((\iota_i^{(T)})^2))} - p\Big)\iota_i^{(T)}) + (\iota^*)^2.$$

Subtracting $M_i^{(T)}$ yields

$$\mathbb{E}[M_i^{(T+1)}|\iota_i^{(T)}] - M_i^{(T)}$$

$$\leq (1-p)\Big[\frac{4\eta\|\boldsymbol{\xi}_i\|_2^2}{(1 + 1/2\exp((\iota_i^{(T)})^2))nm} + \Big(\frac{2\eta\|\boldsymbol{\xi}_i\|_2^2}{(1 + 1/2\exp((\iota_i^{(T)})^2))nm}\Big)^2\Big](\iota_i^{(T)})^2$$

$$+ p\Big[-\frac{4\eta\|\boldsymbol{\xi}_i\|_2^2}{(1 + 2\exp(-(\iota_i^{(T)})^2))nm} + \Big(\frac{2\eta\|\boldsymbol{\xi}_i\|_2^2}{(1 + 2\exp(-(\iota_i^{(T)})^2))nm}\Big)^2\Big](\iota_i^{(T)})^2$$

$$- 2\iota^*\frac{2\eta\|\boldsymbol{\xi}_i\|^2}{nm}\Big(\frac{1}{1 + 2\exp((\iota_i^{(T)})^2)} - p\Big)\iota_i^{(T)}$$

$$= -p\frac{4\eta\|\boldsymbol{\xi}_i\|_2^2(\iota_i^{(T)})^2}{(1 + 2\exp(-(\iota_i^{(T)})^2))nm} + (1-p)\frac{4\eta\|\boldsymbol{\xi}_i\|_2^2(\iota_i^{(T)})^2}{(1 + 1/2\exp((\iota_i^{(T)})^2))nm}$$

$$- 2\iota^*\frac{2\eta\|\boldsymbol{\xi}_i\|^2}{nm}\Big(\frac{1}{1 + 2\exp((\iota_i^{(T)})^2)} - p\Big)\iota_i^{(T)} + O(\eta^2)$$

$$= \frac{4\eta\|\boldsymbol{\xi}_i\|_2^2}{nm}\left[\frac{1 - p(1 + 1/2\exp((\iota_i^{(T)})^2))}{1 + 1/2\exp((\iota_i^{(T)})^2)}(\iota_i^{(T)})^2 - \frac{1 - p(1 + 2\exp((\iota_i^{(T)})^2))}{1 + 2\exp((\iota_i^{(T)})^2)}\iota_i^{(T)}\iota^*\right] + O(\eta^2)$$

$$\leq 0,$$

where the final inequality is by $\iota_i^{(T)} \leq 4\iota^*$ and $p < 1/(1 + 2\exp((\iota_i^{(T)})^2))$ and condition the learning rate from Assumption 3.1, which confirms that $\{M_i^{(t)}\}_{t\in[T_1,T+1]}$ is a super martingale. By one-sided Azuma inequality, with probability at least $1 - \delta$, for any $\tau > 0$, it holds that

$$P(M_i^{(T+1)} - M_i^{(T_1)} \geq \tau) \leq \exp\left(-\frac{\tau^2}{\sum_{k=1}^t c_k^2}\right),$$

where,

$$c_k = |M_i^{(k)} - M_i^{(k-1)}| = |(\iota_i^{(t)} - \iota^*)^2 - (\iota_i^{(t-1)} - \iota^*)^2|$$

$$= |(\iota_i^{(t)} - \iota_i^{(t-1)})(\iota_i^{(t)} + \iota_i^{(t-1)} - 2\iota_i^*)| \leq \eta C_2.$$

Taking the upper bound of $c_k \leq \eta C_2$ yields

$$P((\iota_i^{(T+1)} - \iota_i^*)^2 - C_0^2 \geq \tau) \leq \exp\left(-\frac{\tau^2}{t\eta^2 C_2^2}\right),$$

where we define $C_0 \triangleq (\iota_i^{(T_1)} - \iota_i^*)^2 > 0$. Therefore, we conclude with probability at least $1 - \delta$,

$$|\iota_i^{(T+1)} - \iota_i^*| \leq \sqrt{C_0^2 + \sqrt{\eta^2 t C_2^2 \log(1/\delta)}} \leq C_\iota,$$

where the last inequality is by $\eta \leq \tilde{O}(\sigma_p^{-2} d^{-1})$ and $T < T_2$.

Finally, we check that

$$f_i^{(T+1)} = F_{+1}(\boldsymbol{W}_{+1}^{(T+1)}, \boldsymbol{x}_i) - F_{-1}(\boldsymbol{W}_{-1}^{(T+1)}, \boldsymbol{x}_i)$$

$$= \frac{1}{m} \sum_{r=1}^{m} \sigma\big(\langle \boldsymbol{w}_{+1,r}^{(0)}, \boldsymbol{\mu} \rangle + \gamma_{+1,r}^{(T+1)}\big) + \frac{1}{m} \sum_{r=1}^{m} \sigma\Big(\langle \boldsymbol{w}_{+1,r}^{(0)}, \boldsymbol{\xi}_i \rangle + \overline{\rho}_{+1,r,i}^{(T+1)} + \sum_{i' \neq i} \frac{\langle \boldsymbol{\xi}_i, \boldsymbol{\xi}_{i'} \rangle}{\|\boldsymbol{\xi}_{i'}\|_2^2} \rho_{+1,r,i'}^{(T+1)}\Big)$$

$$\quad - \frac{1}{m} \sum_{r=1}^{m} \sigma\big(\langle \boldsymbol{w}_{-1,r}^{(0)}, \boldsymbol{\mu} \rangle - \gamma_{-1,r}^{(T+1)}\big) - \frac{1}{m} \sum_{r=1}^{m} \sigma\Big(\langle \boldsymbol{w}_{-1,r}^{(0)}, \boldsymbol{\xi}_i \rangle + \underline{\rho}_{-1,r,i}^{(T+1)} + \sum_{i' \neq i} \frac{\langle \boldsymbol{\xi}_i, \boldsymbol{\xi}_{i'} \rangle}{\|\boldsymbol{\xi}_{i'}\|_2^2} \rho_{-1,r,i'}^{(T+1)}\Big)$$

$$\geq -\tilde{\Omega}(\sigma_0^2 \|\boldsymbol{\mu}\|_2^2) - 0.01 + \big(\frac{1}{m} \sum_{r=1}^{m} \overline{\rho}_{+1,r,i}^{(T+1)} - \beta - 16\sqrt{\frac{\log(4n^2/\delta)}{d}} n\alpha\big)^2$$

$$\geq \frac{1}{2}\big(\frac{1}{m} \sum_{r=1}^{m} \overline{\rho}_{+1,r,i}^{(T+1)}\big)^2,$$

where the first inequality is by Lemma 4.3 and the induction claim, and the second inequality is by condition on $d$ from Assumption 3.1. Similarly, by the same argument, we conclude that:

$$f_i^{(T+1)} = F_{+1}(\boldsymbol{W}_{+1}^{(t)}, \boldsymbol{x}_i) - F_{-1}(\boldsymbol{W}_{-1}^{(t)}, \boldsymbol{x}_i)$$

$$\leq \tilde{O}(\sigma_0^2 \|\boldsymbol{\mu}\|_2^2) + 0.01 + \tilde{O}(\sigma_0 \sigma_p \sqrt{d}) + \big(\frac{1}{m} \sum_{r=1}^{m} \overline{\rho}_{+1,r,i}^{(t)} + \beta + 16\sqrt{\frac{\log(4n^2/\delta)}{d}} n\alpha\big)^2$$

$$\leq 2\big(\frac{1}{m} \sum_{r=1}^{m} \overline{\rho}_{+1,r,i}^{(t)}\big)^2.$$

Let $T_2 = T_1 + \log(6/(\sigma_0 \|\boldsymbol{\mu}\|_2)) 4m(1 + \exp(c_2))\eta^{-1}\|\boldsymbol{\mu}\|_2^{-2}$, then by lemma 4.3 we can show that

$$\gamma_{j,r}^{(T_2)} \geq \exp\big(\frac{\eta \|\boldsymbol{\mu}\|_2^2}{4m(1 + \exp(c_2))} t\big) \gamma_{j,r}^{(T_1)}$$

$$= \exp\big(\frac{\eta \|\boldsymbol{\mu}\|_2^2}{4m(1 + \exp(c_2))} \log(6/(\sigma_0 \|\boldsymbol{\mu}\|_2)) 4m(1 + \exp(c_2))\eta^{-1}\|\boldsymbol{\mu}\|_2^{-2}\big) \gamma_{j,r}^{(T_1)}$$

$$= C_0/(\sigma_0 \|\boldsymbol{\mu}\|_2) \gamma_{j,r}^{(T_1)}$$

$$\geq 0.01.$$

$\square$

## D.3 PROOF OF THEOREM 3.2

*Proof of Theorem 3.2.* For the population loss, we expand the expression as follows:

$$\mathcal{L}_{\mathcal{D}}^{0-1}(\boldsymbol{W}^{(t)}) = \mathbb{E}_{(\boldsymbol{x},y) \sim \mathcal{D}}[y \neq f(\boldsymbol{W}^{(t)}, \boldsymbol{x})] = \mathbb{P}(yf(\boldsymbol{W}^{(t)}, \boldsymbol{x}) < 0)$$

$$= \mathbb{P}\Big(\frac{1}{m} \sum_{r=1}^{m} \sigma(\langle \boldsymbol{w}_{-y,r}^{(t)}, \boldsymbol{\xi} \rangle) - \frac{1}{m} \sum_{r=1}^{m} \sigma(\langle \boldsymbol{w}_{y,r}^{(t)}, \boldsymbol{\xi} \rangle) \geq$$

$$\frac{1}{m} \sum_{r=1}^{m} \sigma(\langle \boldsymbol{w}_{y,r}^{(t)}, y\boldsymbol{\mu} \rangle) - \frac{1}{m} \sum_{r=1}^{m} \sigma(\langle \boldsymbol{w}_{-y,r}^{(t)}, y\boldsymbol{\mu} \rangle)\Big).$$

Recall the weight decomposing

$$\boldsymbol{w}_{j,r}^{(t)} = \boldsymbol{w}_{j,r}^{(0)} + j\gamma_{j,r}^{(t)} \|\boldsymbol{\mu}\|_2^{-2} \boldsymbol{\mu} + \sum_{i=1}^{n} \overline{\rho}_{j,r,i}^{(t)} \|\boldsymbol{\xi}_i\|_2^{-2} \boldsymbol{\xi}_i + \sum_{i=1}^{n} \underline{\rho}_{j,r,i}^{(t)} \|\boldsymbol{\xi}_i\|_2^{-2} \boldsymbol{\xi}_i.$$

From this, we obtain:

$$\langle \boldsymbol{w}_{-y,r}^{(t)}, y\boldsymbol{\mu} \rangle = \langle \boldsymbol{w}_{-y,r}^{(0)}, y\boldsymbol{\mu} \rangle - \gamma_{-y,r}^{(t)},$$

$$\langle \boldsymbol{w}_{y,r}^{(t)}, y\boldsymbol{\mu} \rangle = \langle \boldsymbol{w}_{y,r}^{(0)}, y\boldsymbol{\mu} \rangle + \gamma_{y,r}^{(t)}.$$

By Lemma 4.4, we conclude that

$$\langle \boldsymbol{w}_{y,r}^{(t)}, y\boldsymbol{\mu} \rangle = \Theta(1), \quad \langle \boldsymbol{w}_{-y,r}^{(t)}, y\boldsymbol{\mu} \rangle = -\Theta(\gamma_{y,r}^{(t)}) < 0.$$

Therefore, it holds that

$$\frac{1}{m}\sum_{r=1}^{m}\sigma(\langle \boldsymbol{w}_{y,r}^{(t)}, y\boldsymbol{\mu} \rangle) - \frac{1}{m}\sum_{r=1}^{m}\sigma(\langle \boldsymbol{w}_{-y,r}^{(t)}, y\boldsymbol{\mu} \rangle)$$

$$= \frac{1}{m}\sum_{r=1}^{m}\sigma(\langle \boldsymbol{w}_{y,r}^{(t)}, y\boldsymbol{\mu} \rangle)$$

$$= \frac{1}{m}\sum_{r=1}^{m}\sigma(\langle \boldsymbol{w}_{y,r}^{(0)}, y\boldsymbol{\mu} \rangle + \gamma_{y,r}^{(t)})$$

$$= \Theta(1),$$

where the last inequity is by Lemma 4.4.

Next, we provide the bound for the noise memorization part. Define that $g(\boldsymbol{\xi}) = \sum_{r=1}^{m}\sigma(\langle \boldsymbol{w}_{-y,r}^{(t)}, \boldsymbol{\xi} \rangle)$. By Theorem 5.2.2 in Vershynin (2018), for any $\tau > 0$, it holds

$$\mathbb{P}(g(\boldsymbol{\xi}) - \mathbb{E}[g(\boldsymbol{\xi})] \geq \tau) \leq \exp(-\frac{c\tau^2}{\sigma_p^2\|g\|_{\mathrm{Lip}}^2}),$$

where $c$ is a constant and $\|g\|_{\mathrm{Lip}}$ is the Lipschitz norm of function $g(\boldsymbol{\xi})$, which can be calculated as follows:

$$|g(\boldsymbol{\xi}) - g(\boldsymbol{\xi}')| = |\sum_{r=1}^{m}\sigma(\langle \boldsymbol{w}_{-y,r}^{(t)}, \boldsymbol{\xi} \rangle) - \sum_{r=1}^{m}\sigma(\langle \boldsymbol{w}_{-y,r}^{(t)}, \boldsymbol{\xi}' \rangle)|$$

$$\leq \sum_{r=1}^{m}|\sigma(\langle \boldsymbol{w}_{-y,r}^{(t)}, \boldsymbol{\xi} \rangle) - \sigma(\langle \boldsymbol{w}_{-y,r}^{(t)}, \boldsymbol{\xi}' \rangle)|$$

$$\leq 2\sum_{r=1}^{m}|\langle \boldsymbol{w}_{-y,r}^{(t)}, \boldsymbol{\xi} \rangle| \cdot |\langle \boldsymbol{w}_{-y,r}^{(t)}, \boldsymbol{\xi} - \boldsymbol{\xi}' \rangle|$$

$$\leq 2\sum_{r=1}^{m}\|\boldsymbol{w}_{-y,r}^{(t)}\|_2^2 \cdot \|\boldsymbol{\xi}\|_2 \cdot \|\boldsymbol{\xi} - \boldsymbol{\xi}'\|_2$$

$$\leq 3\sum_{r=1}^{m}\|\boldsymbol{w}_{-y,r}^{(t)}\|_2^2\sigma_p\sqrt{d}\|\boldsymbol{\xi} - \boldsymbol{\xi}'\|_2,$$

where the first inequality is by the triangle inequality, the second inequality follows from the the convexity of the activation function, the third inequality is by the Cauchy-Schwarz inequality, and the last inequality follows from A.1. Therefore we conclude that

$$\|g\|_{\mathrm{Lip}} \leq 3\sum_{r=1}^{m}\|\boldsymbol{w}_{-y,r}^{(t)}\|_2^2\sigma_p\sqrt{d}.$$

Furthermore, given that $\langle \boldsymbol{w}_{-y,r}^{(t)}, \boldsymbol{\xi} \rangle \sim \mathcal{N}(0, \sigma_p^2\|\boldsymbol{w}_{-y,r}^{(t)}\|_2^2)$ we have:

$$\mathbb{E}[g(\boldsymbol{\xi})] = \sum_{r=1}^{m}\mathbb{E}[\sigma(\langle \boldsymbol{w}_{-y,r}^{(t)}, \boldsymbol{\xi}' \rangle)] = \sum_{r=1}^{m}\sigma_p^2/2\|\boldsymbol{w}_{-y,r}^{(t)}\|_2^2.$$

To obtain the the upper bound of $g(\boldsymbol{\xi})$, we show that:

$$\|\boldsymbol{w}_{-y,r}^{(t)}\|_2^2 = \|\sum_{i=1}^{n}\rho_{j,r,i}^{(t)}\|\boldsymbol{\xi}_i\|_2^{-2}\boldsymbol{\xi}_i\|_2^2$$

$$= \sum_{i=1}^{n} (\rho_{j,r,i}^{(t)})^2 \|\boldsymbol{\xi}_i\|_2^{-2} \boldsymbol{\xi}_i + 2 \sum_{i=1}^{n} \sum_{j \neq i} \rho_{j,r,i}^{(t)} \rho_{j,r,j}^{(t)} \|\boldsymbol{\xi}_i\|_2^{-2} \|\boldsymbol{\xi}_j\|_2^{-2} \langle \boldsymbol{\xi}_i, \boldsymbol{\xi}_j \rangle$$

$$\leq 3nC(\sigma_p^2 d)^{-1} + 2n^2(\sigma_p^2 d)^{-2} \sigma_p^2 \sqrt{d \log(4n^2/\delta)}$$

$$\leq 4nC(\sigma^2 d)^{-1},$$

where the first inequality is by Lemma A.1, and the second inequality is by the condition on $d$ in Assumption 3.1. With the results above, we conclude that

$$\mathcal{L}_{\mathcal{D}}^{0-1}(\boldsymbol{W}^{(t)}) = \mathbb{E}_{(\boldsymbol{x},y)\sim\mathcal{D}}[y \neq f(\boldsymbol{W}^{(t)}, \boldsymbol{x}))] = \mathbb{P}(yf(\boldsymbol{W}^{(t)}, \boldsymbol{x}) < 0)$$

$$\leq \mathbb{P}(\sum_{r=1}^{m} \sigma(\langle \boldsymbol{w}_{-y,r}^{(t)}, \boldsymbol{\xi} \rangle) \geq \sum_{r=1}^{m} \sigma(\langle \boldsymbol{w}_{y,r}^{(t)}, y\boldsymbol{\mu} \rangle))$$

$$= \mathbb{P}(g(\boldsymbol{\xi}) - \mathbb{E}[g(\boldsymbol{\xi})] \geq \sum_{r=1}^{m} \sigma(\langle \boldsymbol{w}_{y,r}^{(t)}, y\boldsymbol{\mu} \rangle) - \sum_{r=1}^{m} \sigma_p^2/2 \|\boldsymbol{w}_{-y,r}^{(t)}\|_2^2)$$

$$\leq \exp\left(-\frac{c(\sum_{r=1}^{m} \sigma(\langle \boldsymbol{w}_{y,r}^{(t)}, y\boldsymbol{\mu} \rangle) - \sum_{r=1}^{m} \sigma_p^2/2 \|\boldsymbol{w}_{-y,r}^{(t)}\|_2^2)^2}{\sigma_p^2 (3 \sum_{r=1}^{m} \|\boldsymbol{w}_{-y,r}^{(t)}\|_2^2 \sigma_p \sqrt{d})^2}\right)$$

$$\leq \exp\left(-\left(\frac{C_1 - \sigma_p^2/2 \cdot 4nC(\sigma_p^2 d)^{-1}}{3\sigma_p^2 \sqrt{d} 4nC(\sigma_p^2 d)^{-1}}\right)^2\right)$$

$$\leq \exp(\frac{1}{36d}) \exp(-\frac{C_1^2 d}{12^2 n^2 C^2})$$

$$\leq 2 \exp\left(-\frac{C_1^2 d}{12^2 n^2 C^2}\right),$$

which corresponds to the second bullet point of Theorem 3.2. Combined with Lemma 4.4, which establishes the first bullet point, this completes the proof of Theorem 3.2. □

# E ADDITIONAL EXPERIMENTS

In this section, we provide additional experiments to further support our theoretical findings.

## E.1 DEEPER NEURAL NETWORK

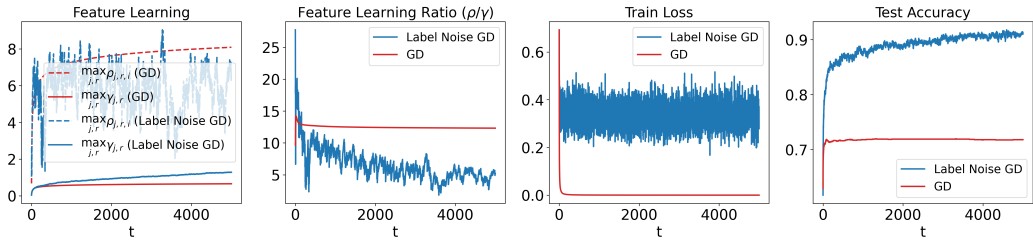

Figure 3: Performance of a 3-layer ReLU neural network: The ratio of noise memorization to signal learning, along with training loss and test accuracy, for standard GD and label noise GD.

We have conducted additional experiments using a 3-layer neural network with ReLU activation. The network is defined as $f(\boldsymbol{W}, \boldsymbol{x}) = F_{+1}(\boldsymbol{W}_{+1}, \boldsymbol{W}, \boldsymbol{x}) - F_{-1}(\boldsymbol{W}_{-1}, \boldsymbol{W}, \boldsymbol{x})$, where

$$F_j(\boldsymbol{W}_j, \boldsymbol{W}, \boldsymbol{x}) = \frac{1}{m} \sum_{r=1}^{m} \sum_{p=1}^{2} \sigma(\langle \boldsymbol{w}_{j,r}, \boldsymbol{z}^{(p)} \rangle), \quad \boldsymbol{z}^{(p)} = \sigma(\boldsymbol{W}^\top \boldsymbol{x}^{(p)}),$$

in which $\sigma(\cdot)$ is the ReLU activation, $\boldsymbol{W} \in \mathbb{R}^{d \times m}$ denotes the weight in the first layer, and $\boldsymbol{W}_{\pm 1} \in \mathbb{R}^{m \times m}$ are weights in the second layer. The last layer is fixed.

Specifically, we train the first two layers. The number of training samples is $n = 200$, and the number of test samples is $n_{\text{test}} = 2000$. The input dimension was set to $d = 2000$. We set the width to $m = 20$, the learning rate to $\eta = 0.5$, and the noise flip rate to $p = 0.1$. The data model follows our theoretical setting, where $\boldsymbol{\mu} = [1, 0, 0, \cdots, 0]$ and the noise strength is $\sigma_p = 1$. The experimental results, shown in Figure 3, are consistent with our original findings: compared to standard gradient descent, label noise GD boosts signal learning (as shown in the first plot) and achieves better generalization (as shown in the last plot).

## E.2  REAL WORLD DATASET

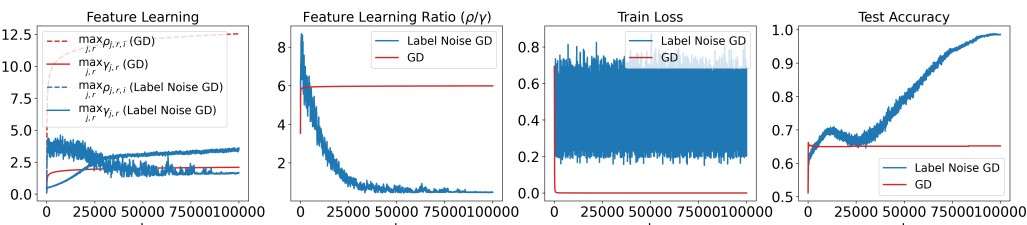

Figure 4: Performance on the modified MNIST dataset: The ratio of noise memorization to signal learning, along with training loss and test accuracy, for standard GD and label noise GD.

We conducted an experiment using the MNIST dataset, in which Gaussian noise was added to the borders of the images while retaining the digits in the middle. The noise level was set to $\sigma_p = 5$. Moreover, the original pixel values of the digits ranged from 0 to 255, and we chose a normalization factor of 80. In this setup, the added noise formed a "noise patch" and the digits formed a "signal patch". We focused on the digits '0' and '1', using $n = 100$ samples for training and 200 samples for testing. The learning rate was set to $\eta = 0.001$, and the width was set to $m = 20$, with a label noise level of $p = 0.15$. The results, shown in Figure 4, were consistent with our theoretical conclusions, reinforcing the insights derived from our analysis.

To assess the sensitivity of the methods to the choice of noise parameters and signal normalization, we conducted additional experiments on a modified MNIST dataset. The signal normalization values were varied from 60 to 140, while the noise levels ranged from 4 to 8. For each combination of noise level and signal normalization, we trained the neural network for 200,000 steps with a learning rate $\eta = 0.001$, using either standard gradient descent (GD) or label noise GD.

The resulting test errors are visualized in Figure 5. Notably, label noise GD (right) consistently achieves higher test accuracy than standard GD (left) across all configurations. This demonstrates the robustness of label noise GD to variations in noise and signal normalization parameters.

The motivation behind using MNIST was its clearer signal, which allows us to more directly observe the effects of label noise without other confounding factors. However, we also conducted experiments on a subset of CIFAR-10, using two classes: *airplane* and *automobile*. Gaussian noise was added to a portion of the images, following a similar setup to MNIST. For these experiments, we set $q = 2$, the number of neurons $m = 20$, the learning rate $\eta = 0.001$, the signal norm signal_norm $= 64$, the noise level noise_level $= 5$, the number of samples $n = 100$, the label noise probability $p = 0.15$, and the input dimension $d = 6144$.

The results shown in Figure 6 indicate that label noise GD continues to provide benefits in terms of generalization compared to standard GD. We believe these extended experiments help establish a broader applicability of our findings to more complex benchmarks.

## E.3  DIFFERENT TYPE OF LABEL NOISE

To validate the robustness of label noise GD under different noise forms, we varied $p$ across different values. For example, we show the results for $p = 0.3$ in Figure 7 and $p = 0.4$ in Figure 8. The results consistently indicate that label noise helps reduce overfitting and boost generalization, especially in low SNR settings.

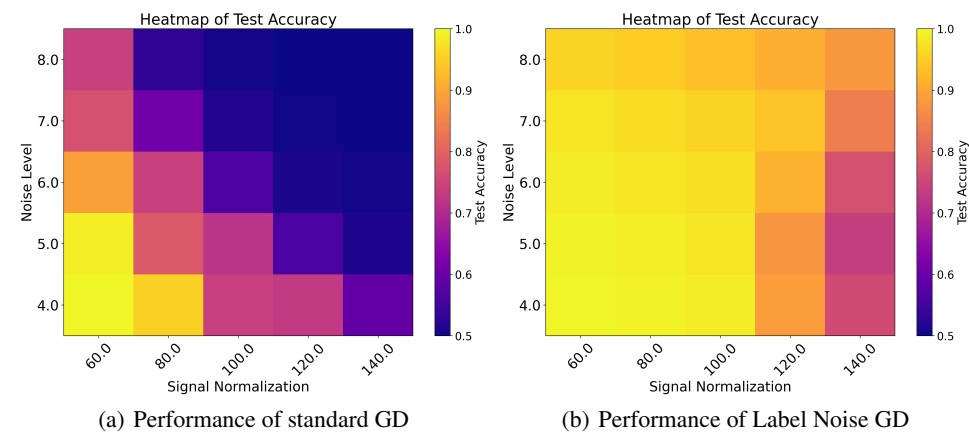

Figure 5: Test accuracy heatmap of standard GD (left) and Label Noise GD (right) after training on modified MNIST dataset.

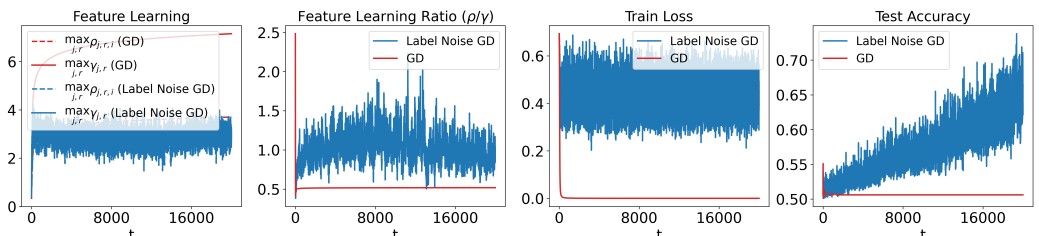

Figure 6: Performance on the modified CIFAR-10 dataset: The ratio of noise memorization to signal learning, along with training loss and test accuracy, for standard GD and label noise GD.

In addition, we extended our empirical analysis to include Gaussian noise and uniform distribution noise added to the labels. For Gaussian noise, we used two examples, namely $\epsilon_i^{(t)} \sim \mathcal{N}(1,1)$ and $\epsilon_i^{(t)} \sim \mathcal{N}(1,1)$, with the results shown in Figures 9 and 10, respectively. Furthermore, for the uniform distribution, we simulated the noise with $\epsilon_i^{(t)} \sim \text{unif}[-1,2]$ and $\epsilon_i^{(t)} \sim \text{unif}[-2,3]$. The results are shown in Figures 11 and 12, respectively.

Our results indicate that label noise GD still performs effectively, achieving better generalization compared to standard GD, providing further evidence of the robustness of label noise GD under different noise forms.

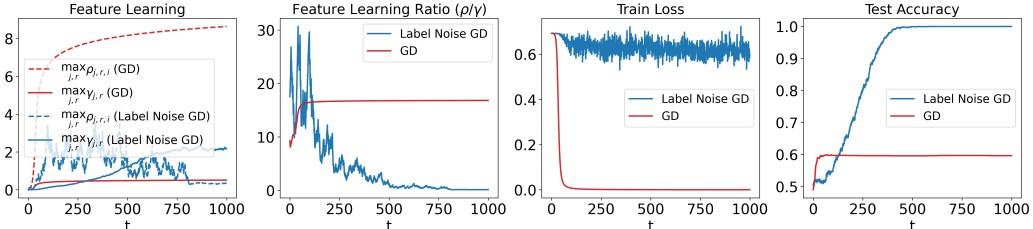

Figure 7: Performance with flip noise $p = 0.3$: The ratio of noise memorization to signal learning, training loss, and test accuracy of standard GD and label noise GD.

### E.4 HIGHER ORDER POLYNOMIAL RELU

In this work, we set the activation function as squared ReLU. This choice makes $q = 2$ a particularly interesting and challenging case to analyze, as it allows us to study the interaction between signal and noise in a setting that closely resembles practical two-layer ReLU networks.

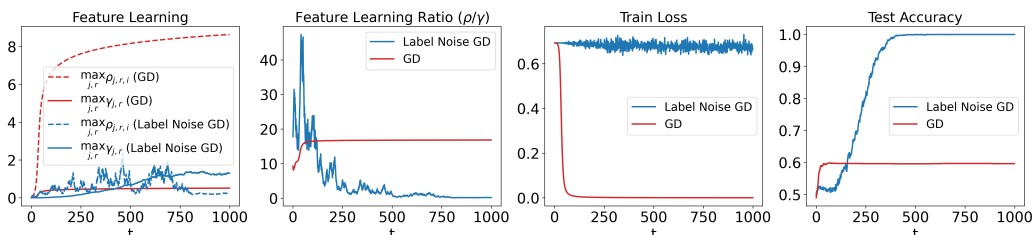

Figure 8: Performance with flip noise $p = 0.4$: The ratio of noise memorization to signal learning, training loss, and test accuracy of standard GD and label noise GD.

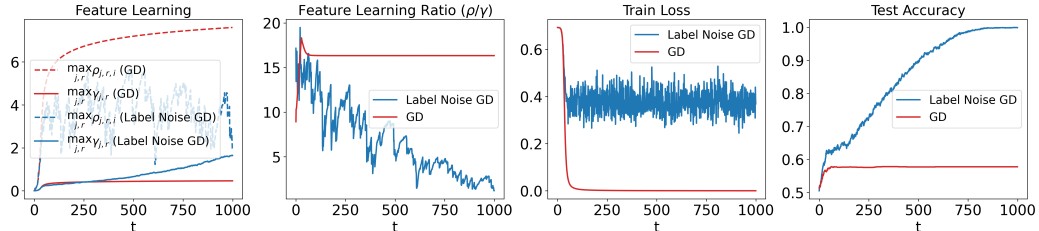

Figure 9: Performance with Gaussian noise $\mathcal{N}(1,1)$: The ratio of noise memorization to signal learning, training loss, and test accuracy of standard GD and label noise GD.

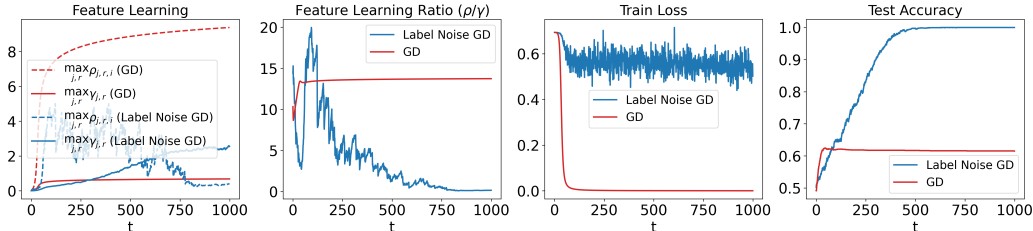

Figure 10: Performance with Gaussian noise $\mathcal{N}(0.6,1)$: The ratio of noise memorization to signal learning, training loss, and test accuracy of standard GD and label noise GD.

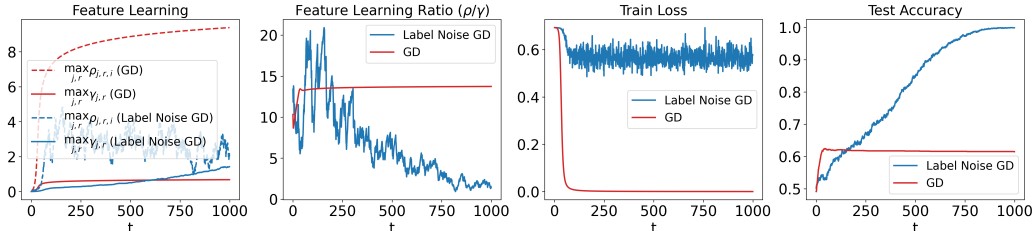

Figure 11: Performance with uniform distribution noise $\mathrm{unif}[-1,2]$: The ratio of noise memorization to signal learning, training loss, and test accuracy of standard GD and label noise GD.

For higher values of $q$, we also conducted experiments with $q = 3$ and $q = 4$. For $q = 3$, we set the learning rate $\eta = 0.5$, the number of neurons $m = 20$, the number of samples $n = 200$, the signal mean $\boldsymbol{\mu} = [2, 0, 0, \cdots, 0]$, and the noise strength $\sigma_p = 0.5$. The results are shown in Figure 13. For $q = 4$, the parameters were set as $\eta = 0.1$, $m = 20$, $n = 50$, $\boldsymbol{\mu} = [5, 0, 0, \cdots, 0]$, and $\sigma_p = 0.5$. The results are shown in Figure 14.

In all these cases, the experimental results consistently show that using a higher polynomial ReLU activation helps label noise GD suppress noise memorization while enhancing signal learning. This ultimately leads to improved test accuracy compared to standard GD.

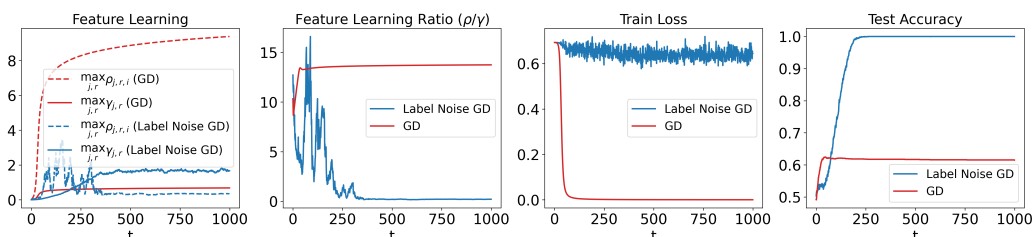

Figure 12: Performance with uniform distribution noise $\mathrm{unif}[-2, 3]$: The ratio of noise memorization to signal learning, training loss, and test accuracy of standard GD and label noise GD.

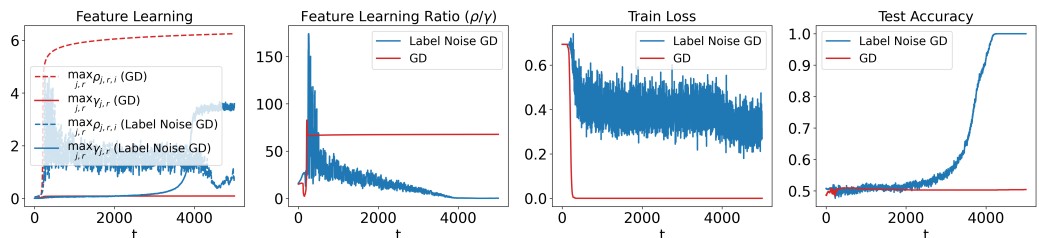

Figure 13: Performance with $q = 3$ for polynomial ReLU: The ratio of noise memorization to signal learning, training loss, and test accuracy of standard GD and label noise GD.

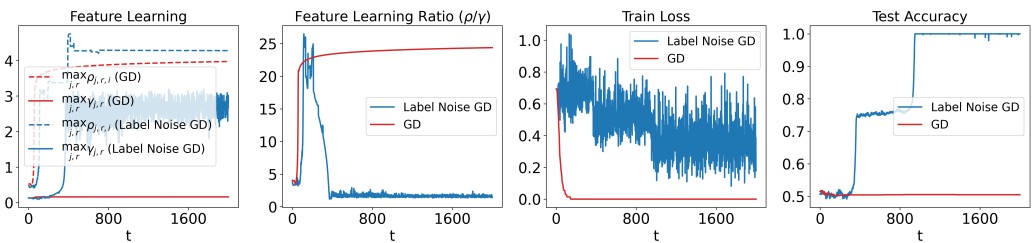

Figure 14: Performance with $q = 4$ for polynomial ReLU: The ratio of noise memorization to signal learning, training loss, and test accuracy of standard GD and label noise GD.

