# OpenReview forum: "Label Noise Gradient Descent Improves Generalization in the Low SNR Regime"
_ICLR.cc/2025/Conference — Submitted to ICLR 2025_

### Official Review · Reviewer_y9sU · 2024-10-20

**Soundness:** 1
**Presentation:** 3
**Contribution:** 1
**Rating:** 3
**Confidence:** 5

**Summary:**

This paper presents a very specific example where there is large noise in the data and the number of samples is smaller than the dimensions, so that standard GD cannot be better than random guess, but GD with label noise can achieve 100% accuracy. The authors proved for this specific scenario that label noise GD can achieve small error while standard GD cannot, and validated it with a synthetic experiment.

**Strengths:**

1. Reproducibility: I have done the experiment myself and the results can be reproduced
2. Intellectually, there exists such an example is a fun fact to know
3. The authors proved their theorems with complicated techniques, so there might be technical novelties though I cannot check the proof carefully

**Weaknesses:**

Over the years I have seen a number of papers claiming to "prove" some big or surprising results, by means of a very specific, sometimes even toyish, example. I am not a big fan of this class of papers. Unfortunately, this paper belongs to this class. My main concern of this type of papers is that they cannot show whether the results they prove only work for the very specific example they study, and whether these results can be generalized to a more, even slightly, general situation. After all, one can prove that something does not work by giving a counter-example, but one cannot prove that something works by just giving one example. Thus, the conclusions these papers make are usually misleading, and since they make big claims, these papers are often much more misleading than others.

In the case of this particular submission, I did some experiments and came to the conclusion that the claim of this paper "label noise GD can be better than standard GD" only works for very limited situations. Even for the very example the authors provided, changing just one parameter can break the whole story.

I used the exact same setting as the authors in Section 5. I always used the parameters in the code in the supplementary material if it disagrees with the paper. For instance, the paper used $\eta=0.5$ (line 465) while in the code $\eta = 1.0$, so I used $\eta = 1.0$. The paper didn't say what flip probability was used, and $p=0.1$ was used in the code so I used that too. My code: https://pastecode.io/s/r6epwvx7

I always used fixed random seeds so my results are fully reproducible. I encourage the authors and my fellow reviewers to play with my code on their own.

First, I reproduced the results in the paper, using the exact same set of parameters. In the following plots, the blue curves are the training loss, while the orange curves are the test accuracy.

- Standard GD: https://i.postimg.cc/YC0LGM7z/1.png
- Label noise GD: https://i.postimg.cc/C1TRQ8qc/2.png

I managed to reproduce the exact same results as the authors.

Next, when I stared at the code, the first thing that struck me was that the model architecture looked quite strange. First, I couldn't understand what the authors mean by "two patches" $x = \{ x^{(1)}, x^{(2)} \}$ (line 144), and I couldn't understand why there should be a "patch" that is just Gaussian noise. Second, I don't think the model in lines 172-174 is a "CNN" because I don't see any convolution here. Third, the squared ReLU activation is not usually used. Anyway, I tried to change the model a little bit and see if it could help, and the very first thing I did was to use a smaller $m$ (that is, making the network narrower), which I believe is a very reasonable and very natural thought. Here are the results:

- Standard GD, with $m=3$: https://i.postimg.cc/br6NpSZc/3.png
- Label noise GD, with $m=3$: https://i.postimg.cc/7Ydw9k7V/4.png

Standard GD now achieves 100% test accuracy. Thus, by simply changing $m=20$ to $m=3$, the problem is immediately solved.

Plus, when $m=3$, standard GD still works with a larger noise. I changed $\sigma_p = 0.5$ to $\sigma_p = 2.0$, and ran the two methods again with $m=3$. Here are the results:

- Standard GD, with $m=3, \sigma_p = 2.0$: https://i.postimg.cc/HLCHG968/5.png
- Label noise GD, with $m=3, \sigma_p = 2.0$: https://i.postimg.cc/zDPrDrdr/6.png

In this case, standard GD can achieve 90-ish% test accuracy, while the performance of label noise GD is miserable. Thus, I've shown a case where label noise GD is much worse than standard GD.

The authors might argue: "Our theory only works when $m$ is not too small (Assumption 3.1 (ii))." I don't think this is a valid argument, because one is free to choose whatever model architecture they want. And if a narrower network can work so well, why bother using a wider one that is much worse? Moreover, when we couldn't get expected results in deep learning, we should always start with trying something straightforward like changing the network architecture, optimizer, loss, etc., before trying something very unnatural like "adding label noise during training".

To sum up, the experimental results I present here establish the fact that the claim of this paper, that is "label noise GD is better than GD", is not very robust and not very sound. I am not convinced that label noise can improve generalization in any practical regime.

As a final remark, I am not saying that the authors should not get credit for constructing this particular example in the paper. I think it is a fun fact to know that such an example exists. However, I think the conclusion of this submission is unsound, and the paper is quite misleading due to its catchy claim "label noise GD improves generalization". The claim can be easily invalidated by just changing one parameter. Imagine a PhD student who read this paper, got attracted by the idea, and tried to add label noise on a number of tasks, only to find out that it didn't work at all and lots of time and work went down the drain. For the sake of that student and the learning theory community, I must insist on rejecting this submission.

**Questions:**

If the authors really want to argue that label noise GD has its merits in some situations, they need to provide general sufficient or necessary conditions under which label noise GD is better than standard GD, instead of just a toy example. If you can provide a sufficient condition that is weak enough, then the results you established will be much more useful.

I also think that the authors ought to do experiments on some real-world experiments, to show that your theory can be at least useful in some situations. You also need to show that simple solutions such as changing the network width cannot solve the problem, and thus using label noise is necessary.

---

> ### Author Response · Authors · 2024-11-21
> **Responses (Part I) to Reviewer y9sU**
>
> We thank the reviewer for the helpful feedback. We address the technical concerns below.
>
> ---------
>
> > one is free to choose whatever model architecture they want
>
> We do not claim that standard GD always fails to learn this problem regardless of the architecture and hyperparameter – such an unconditional lower bound is extremely difficult to establish. We are not interested in the statistical complexity of solving this problem under some notion of “optimal architecture” — in fact, it is clear that the target function in our signal-noise data model (Definition 2.1) can be expressed by a linear classifier, so if the motivation is to simply learn this binary classification task, many well-known and influential papers on feature learning theory [Allen-Zhu & Li (2020)](https://openreview.net/pdf?id=Uuf2q9TfXGA),[Jelassi et al. 2022](https://proceedings.mlr.press/v162/jelassi22a/jelassi22a.pdf),[Wen 2021](https://proceedings.mlr.press/v139/wen21c/wen21c.pdf),[Shen et al. (2022)](https://proceedings.mlr.press/v162/shen22a/shen22a.pdf),[Kou et al. 2023](https://openreview.net/pdf?id=qmwtMuRh1j) are meaningless and ought to be rejected, because there is no point of using a (wide) neural network when the task can be solved by a linear model.
>
> In other words, the signal-noise model has been extensively studied not because it represents an intrinsically challenging function class that requires extensive network width to learn; instead, the setting provides a tractable model for the nonlinear optimization dynamics, so that one can reason about how simple modifications of the learning algorithm affect the order of signal learning vs. overfitting. More specifically, it is well known that deep neural networks are capable of overfitting to noise in the training set, yet gradient-based optimization tends to avoid such memorization solutions. The theory community introduced the signal-noise data model to capture this phenomenon, where the key features are that (i) the data contains a noise component that potentially leads to overfitting, and (ii) the neural network is expressive enough to memorize noise. Consequently, prior works  [Allen-Zhu & Li (2020)](https://openreview.net/pdf?id=Uuf2q9TfXGA),[Jelassi et al. 2022](https://proceedings.mlr.press/v162/jelassi22a/jelassi22a.pdf),[Cao et al. 2022](https://proceedings.neurips.cc/paper_files/paper/2022/file/a12c999be280372b157294e72a4bbc8b-Supplemental-Conference.pdf),[Wen 2021](https://proceedings.mlr.press/v139/wen21c/wen21c.pdf),[Shen et al. (2022)](https://proceedings.mlr.press/v162/shen22a/shen22a.pdf),[Chen et al. 2023](https://openreview.net/pdf?id=eozEoAtjG8) and our analysis requires a minimum width of the neural network so that noise memorization is feasible. Setting the width to be very small (e.g., m=3) limits the network capacity and therefore rules out the possibility of overfitting — this explicit capacity control is tangential to our investigation where harmful overfitting should be avoided by algorithmic regularization.
>
> While we do not argue that all our theoretical assumptions are justified simply because prior works also consider similar restrictions, we would like to point out that these well-cited papers in feature learning theory were not dismissed by the theory community because "the model architecture looked quite strange" or the studied learning problem is not statistically hard; rather, they serve as an important first step toward understanding more complicated feature learning dynamics in more realistic data settings.
>
> We hope the reviewer can re-evaluate our contributions in light of this perspective, and reconsider the statement that “for the sake of that student and the learning theory community, I must insist on rejecting this submission,” which we believe is inappropriate.

---

> > ### Author Response · Authors · 2024-11-21
> > **Responses (Part II) to Reviewer y9sU**
> >
> > > My main concern of this type of papers is that they cannot show whether the results they prove only work for the very specific example they study
> >
> > We agree that our theoretical results are limited to an idealized setting, as stated in the abstract: “we consider the learning of a two-layer NN with a simple label noise gradient descent (GD) algorithm, in an idealized signal-noise data setting.”  As remarked in the Introduction section, this signal-noise model has been extensively studied in the neural network theory literature, as a sandbox to understand how gradient-based training prioritizes learning of the underlying signal instead of overfitting to noise. We make the following clarifications.
> >
> > - *I couldn't understand why there should be a "patch" that is just Gaussian noise*
> >
> > As explained in the Problem Setting section (starting line 160), this problem setting is designed to reflect the scenario where data contains a mix of relevant and irrelevant features – see ( [Allen-Zhu & Li (2020)](https://openreview.net/pdf?id=Uuf2q9TfXGA), Appendix A) for discussions. The Gaussian noise component models features that (i) do not correlate with the ground truth signal, (ii) can be utilized by the neural network to achieve low training error (hence overfitting). Such a setting has also been analyzed in [Jelassi et al. 2022](https://proceedings.mlr.press/v162/jelassi22a/jelassi22a.pdf),[Cao et al. 2022](https://proceedings.neurips.cc/paper_files/paper/2022/file/a12c999be280372b157294e72a4bbc8b-Supplemental-Conference.pdf),[Wen 2021](https://proceedings.mlr.press/v139/wen21c/wen21c.pdf),[Shen et al. (2022)](https://proceedings.mlr.press/v162/shen22a/shen22a.pdf),[Chen et al. 2023](https://openreview.net/pdf?id=eozEoAtjG8).
> >
> >
> > - *I don't see any convolution here.*
> >
> > The CNN terminology in this problem setting originates from the influential paper of [Cao et al. 2022](https://proceedings.neurips.cc/paper_files/paper/2022/file/a12c999be280372b157294e72a4bbc8b-Supplemental-Conference.pdf), and has been used in many subsequent works [Shen et al. (2022)](https://proceedings.mlr.press/v162/shen22a/shen22a.pdf),[Chen et al. 2023](https://openreview.net/pdf?id=eozEoAtjG8),[Kou et al. 2023](https://openreview.net/pdf?id=qmwtMuRh1j),[Chen et al. 2023](https://openreview.net/pdf?id=3WAnGWLpSQ). Convolution refers to the local structure of the predictor, where the “filters” apply to the signal and noise patches separately. We have clarified this terminology in the revision.
> >
> > - *the squared ReLU activation is not usually used*
> >
> > As discussed in Remark 2.1, the squared ReLU activation mimics the 2-homogenous dynamics of jointly training both layers. High-order local growth of the activation function has been assumed in many theoretical works on gradient-based feature learning, see [Allen-Zhu & Li (2020)](https://openreview.net/pdf?id=Uuf2q9TfXGA),[Cao et al. 2022](https://proceedings.neurips.cc/paper_files/paper/2022/file/a12c999be280372b157294e72a4bbc8b-Supplemental-Conference.pdf),[Shen et al. (2022)](https://proceedings.mlr.press/v162/shen22a/shen22a.pdf),[Lu et al. 2023](https://openreview.net/pdf?id=wYmvN3sQpG), [Zou et al. 2021](https://openreview.net/pdf?id=iUYpN14qjTF).
> >
> > - *... before trying something very unnatural like "adding label noise during training".*
> >
> > We respectfully disagree that adding label noise during training is very unnatural. As discussed in the introduction (starting line 71), this approach has been employed to mitigate overfitting, both in theoretical [Blanc et al. 2020](https://arxiv.org/pdf/1904.09080),[Damian et al. 2021](https://proceedings.neurips.cc/paper/2021/file/e6af401c28c1790eaef7d55c92ab6ab6-Paper.pdf),[Huh et al. 2024](https://proceedings.mlr.press/v238/eun-huh24a/eun-huh24a.pdf) and empirical studies [Shallue et al. 2019](https://arxiv.org/pdf/1811.03600),[Szegedy et al. 2016](https://www.cv-foundation.org/openaccess/content_cvpr_2016/papers/Szegedy_Rethinking_the_Inception_CVPR_2016_paper.pdf),[Wen et al. 2019](https://arxiv.org/pdf/1902.08234v3). But to our knowledge, we are the first to characterize the optimization dynamics of this method in the context of signal-noise model and gradient-based feature learning,

---

> ### Author Response · Authors · 2024-11-21
> **Responses (Part III) to Reviewer y9sU**
>
> - *the paper is quite misleading due to its catchy claim "label noise GD improves generalization"*
>
> Our primary goal is to provide a theoretical understanding of the differences in generalization between label noise GD and standard GD in the well-studied signal-noise data model. To ensure the soundness of our theoretical claims, we have explicitly detailed our data model, neural network structure, and provided proof sketches.
> We do not claim that label noise GD is universally superior to standard GD in all settings (in fact, benign overfitting of standard GD has been established in many prior works in the high-SNR regime). Our analysis is specific to the signal-noise setting, and we argue that the benefit of label noise injection is prominent in the low SNR setting (see response below on the experimental setup), which is known to be a challenging regime to establish benign overfitting.
>
> We believe that the mechanism presented in Theorem 3.2 (regularization effect through label noise) is nontrivial and fairly robust in the studied signal-noise data model – see our response to the experimental settings.
>
> ----------
>
> > I did some experiments and came to the conclusion that the claim of this paper "label noise GD can be better than standard GD" only works for very limited situations.
>
> As discussed in our previous response, decreasing the network width reduces the model capacity and hence its ability to memorize noise. Hence it is not surprising that overfitting is less of an issue regardless of the optimization algorithm. Our analysis considers the setting where the network has enough capacity to memorize the training labels, yet optimization provides additional regularization that prevents overfitting.
> Nevertheless, we respectfully disagree that “the claim can be easily invalidated by just changing one parameter”. We make the following clarifications.
>
> - First, based on the plots you provided, we noticed that the training loss of label noise GD appears to be diverging, which indicates that the learning rate is too large. To investigate this further, we adjusted the learning rate by using a smaller value in your provided code. The updated results, which can be found https://postimg.cc/py7B7BXB, show that label noise GD performs at least as well as standard GD in achieving high test accuracy, even though our theoretical analysis excludes such narrow-width settings.
>
> - Second, we conducted additional experiments for the narrow-width model ($m = 3$) under smaller signal-to-noise ratio (SNR) $\sigma_p = 10$. Specifically, we used the following parameter settings: $\eta = 0.002$, $ m = 3$, $\mu[0]$ = 2, and $\sigma_p = 10$. The results, which can be found: standard GD (https://postimg.cc/vgZzRqV0); label noise GD (https://postimg.cc/D8HYrTXn), show that label noise GD outperforms standard GD under these conditions. These simulations were based on the code you provided, with the only modifications being the value of $\sigma_p$ and setting the learning rate to 0.002. These findings suggest that even for narrow models, label noise GD can still be advantageous when the SNR is low.
>
> - Third, to further confirm that label noise GD can consistently outperform standard GD in the low SNR regime, we conducted an additional experiment involving a hyper-parameter search on the learning rate. This time we used a larger width $m = 10$, which is more aligned with our theoretical requirement on the network width compared to $m = 3$. The specific settings were: $m = 10$, $\sigma_p = 2$, and $\mu[0] = 2$, while keeping other hyper-parameters unchanged. We conducted a search over different learning rates from the set [0.01, 0.05, 0.1, 0.5, 1.0]. The results show that standard GD failed to achieve high test accuracy across all learning rates (https://postimg.cc/d74292xr), while label noise GD achieved almost 100% test accuracy when using $\eta = 0.1$ (https://postimg.cc/zHRT59XF). These results corroborate the claim that label noise GD has significant advantage in the low SNR regime, particularly when using an appropriate network width that meets the theoretical conditions.
>
> - Lastly, following your suggestion, we conducted an experiment on a real-world dataset using MNIST. We introduced noise to the dataset to align with our theoretical setting and compared the performance of standard GD and label noise GD under this setting. The results are consistent with our theoretical findings—label noise GD demonstrated improved generalization compared to standard GD, especially in noisy environments. These additional findings are now included in **Appendix E.2** of the revised manuscript.

---

> > ### Comment · Reviewer_y9sU · 2024-11-22
> >
> > I thank the authors for the rebuttal. Here is my response.
> >
> > > many well-known and influential papers on feature learning theory (a bunch of citations) are meaningless
> >
> > I don't think this is an apple-to-apple comparison. The main difference between your paper and the five papers you cited is that these papers are either trying to understand some empirically observed phenomena, or providing counter arguments using simple data models, while you are claiming that label noise GD can improve generalization, which has not been widely observed in real applications and is therefore unverified and unsound.
> >
> > For example, it is widely observed in real applications that momentum SGD leads to better generalization. Thus, Jelassi & Li (2022) asked if momentum can always improve generalization. They constructed an example where momentum actually hurts generalization, and thus their conclusion was "momentum does not always lead to a higher generalization in deep learning". This conclusion is sound, because like I said in my review, one can prove that something does not work by showing a counter-example. Similarly, Allen-Zhu & Li (2019) used a counter-example to argue that "ensemble in deep learning is very different from ensemble in random feature mapping", and this is a sound conclusion.
> >
> > However, in your case, you are trying to generalize an unverified conclusion from a very specific setting to the general "low SNR regime", and such generalization is questionable. To justify such generalization, you either need to prove that your conclusion is robust w.r.t. the parameters, which is shown to be false by my experiments; or you can provide empirical evidence by doing experiments on **real** datasets.
> >
> > > (Adding label noise) has been employed to mitigate overfitting
> >
> > I think you are confusing among three things: flipping the labels randomly (your paper), label smoothing (the empirical papers), and implicit regularization in SGD (the theoretical papers).
> >
> > All the empirical works you cited used label smoothing. I am well aware that label smoothing works in a couple of real applications. Label smoothing typically works with exp-tailed loss functions such as cross entropy, because they can prevent the weights from getting too large. However, I am not aware of any real applications where people **flip** the labels, like you did in your setting. I am happy to admit that I am wrong if you could point me to one real application, where flipping the labels during training is actually preferable in practice. At this point, there is no empirical evidence showing that your definition of "label noise GD" actually works in real tasks. Alternatively, you can try to extend your results to label smoothing.
> >
> > Regarding the three theoretical papers you cited here, they were studying the implicit noise induced by SGD. However, none of the three cited papers suggested that flipping the labels during training is a good idea. I am very familiar with this line of research on SGD. One of the main arguments is that SGD can be better than GD due to the stochasticity, which I think is a beautiful result, but this is drastically different from flipping the labels.
> >
> > > We do not claim that label noise GD is universally superior to standard GD in all settings
> >
> > Right now your title is "Label Noise Gradient Descent Improves Generalization in the Low SNR Regime", which is claiming that something works ("improves") in a fairly general setting ("low SNR regime"). This claim is unsound and misleading, because (i) there is no empirical evidence that flipping labels really works in real tasks, and (ii) like I said in my review, you cannot prove that something works by giving a specific example. I'd suggest you change your title to "on the implicit regularization effect of label noise gradient descent", similar to the previous papers on SGD.
> >
> > I should also point out that SGD has better generalization than GD is widely observed in practice, but none of the papers you cited in your rebuttal claimed that they "proved" SGD is better than GD in any general sense. They were only trying to understand this baffling phenomenon, since most people would imagine that SGD is worse due to its stochasticity. I have some reservations for some of these papers, but in general I think their conclusions are fine. In your case, I am not convinced that your conclusion is sound.
> >
> > **My experiment**: The only purpose of my experiment is to show that your result is not robust w.r.t. the parameters. Choosing a smaller m is the very first thing I thought of. I am sure that there are other ways to break your result, but it is not my job to find them. Instead, it is the authors' job to make sure that all claims are sound and robust, and cannot be broken by such a simple tweak. Thus, I suggest you do more sensitivity analysis and ablation studies, to show that label noise GD really works and is irreplaceable, if this is what you really want to argue.

---

> ### Author Response · Authors · 2024-11-24
> **Further Response to Reviewer y9sU**
>
> We thank the reviewer for the response. We address the additional concerns below.
>
> --------
>
> > Regarding the three theoretical papers you cited here, they were studying the implicit noise induced by SGD. However, none of the three cited papers suggested that flipping the labels during training is a good idea. I am very familiar with this line of research on SGD. One of the main arguments is that SGD can be better than GD due to the stochasticity, which I think is a beautiful result, but this is drastically different from flipping the labels.
>
> As mentioned in Line 210 of the main text, our label-flipping procedure was introduced in [Damian et al. 2021](https://proceedings.neurips.cc/paper/2021/file/e6af401c28c1790eaef7d55c92ab6ab6-Paper.pdf), which the reviewer should be very familiar with. This extension is needed since the additive label noise used in the regression analysis is not suited for binary classification – see their Section 5.1. Note that all experiments on CIFAR-10 + ResNet in [Damian et al. 2021](https://proceedings.neurips.cc/paper/2021/file/e6af401c28c1790eaef7d55c92ab6ab6-Paper.pdf) were conducted using this label-flipping algorithm.
> Besides, [Haochen et al. 2021](https://proceedings.mlr.press/v134/haochen21a/haochen21a.pdf) trained a VGG19 model on CIFAR100 with flipping noise, and found adopting flipping label noise achieves better test error (around 15% improvement over baseline). Similar label flipping regularization appeared in [Xie et al. 2016.](https://arxiv.org/pdf/1605.00055), and it has been argued that this approach can be cast as label smoothing methods [Li et al. 2020](​​https://proceedings.mlr.press/v108/li20e/li20e.pdf).
>
> We have added these references in the revised manuscript to clarify the role of label flipping and how it relates to existing works on regularization.
>
> -----------
>
> > Thus, Jelassi & Li (2022) asked if momentum can always improve generalization. They constructed an example where momentum actually hurts generalization, and thus their conclusion was "momentum does not always lead to a higher generalization in deep learning". This conclusion is sound, because like I said in my review, one can prove that something does not work by showing a counter-example.
>
> We respectfully disagree. The main contribution of Jelassi & Li (2022) is the analysis of GD vs. GD+M in the signal-noise data model where the conclusion is that GD+M indeed improves generalization – this is in the exact same vein as our comparison of SGD vs. label noise GD.
> The claim that "momentum does not always lead to a higher generalization in deep learning" is merely confirmed in a toy experiment to motivate their introduction of the signal-noise data model, in which GD+M provably avoids overfitting.
> We are not aware of any existing works on the signal-noise data model where the contribution is only a counterexample that a specific algorithm does not work. If anything in Jelassi & Li (2022) counts as a “counterexample”, it would be the negative result that GD fails to generalize (Theorem 4.1), which is established under similar hyperparameters (width, learning rate, etc.) as our analysis.
>
> -----------
>
>
> > The only purpose of my experiment is to show that your result is not robust w.r.t. the parameters. Choosing a smaller m is the very first thing I thought of.
>
> Please refer to our Part I response. Choosing a network width to be small introduces additional capacity control which reduces overfitting without algorithmic regularization. As mentioned earlier, setting $m=3$ will also break the conclusion in prior works on the comparison of algorithms in the signal-noise model (e.g., [Jelassi and Li 2022](https://proceedings.mlr.press/v162/jelassi22a/jelassi22a.pdf), [Chen et al. 2023](https://openreview.net/pdf?id=eozEoAtjG8)). In this regard, we do not think that the claims in any of these works are more “sound and robust” than ours.
>
> -----------
>
> > Suggestion on changing the Title
>
> We have revised our title to:"How Does Label Noise Gradient Descent Improve Generalization in the Low SNR Regime?"
> This title suggests an investigative nature of our study (rather than presenting a universally applicable claim), analogous to the title of [Jelassi and Li 2022](https://proceedings.mlr.press/v162/jelassi22a/jelassi22a.pdf).
> We hope this addresses the reviewer's concern on the presented scope of our results.

---

> > ### Comment · Reviewer_y9sU · 2024-11-26
> >
> > I thank the authors for the response.
> >
> > >  The main contribution of Jelassi & Li (2022) is the analysis of GD vs. GD+M in the signal-noise data model
> >
> > Let me make this very clear then. The problem this paper studied was why momentum can improve in deep learning, which is widely observed in real applications. The authors showed three things:
> > - They first used some experiments to empirically show that momentum consistently improves generalization, similar to what has been widely observed
> > - Then, they used a toy example to demonstrate that momentum does not always improve generalization
> > - Finally, they provided one possible explanation why momentum could improve generalization (GD+M) using an example
> >
> > Why do I think their conclusion is more sound than yours?
> > - First, the effect of momentum has been widely observed, and indeed momentum is used by default in real applications. Moreover, Jelassi & Li also did experiments to provide further empirical evidence. In your case, however, there is no empirical evidence that the method you are proposing really works in real applications. None of the papers you provided in your new response empirically showed that flipping labels could be preferable in real applications. For example, HaoChen et al. (2021) argued that SGD + flipping labels is better than SGD + Gaussian noise, but I doubt that they showed that SGD + flipping labels is better than simple SGD on CIFAR-100 in a realistic setting. In fact, none of the methods on the CIFAR-100 leaderboard uses such kind of label noise (https://paperswithcode.com/sota/image-classification-on-cifar-100). If they really showed that flipping labels can lead to a 15% improvement like you claimed, then either their baseline was too weak (in Figure 1 they used the weaker small batch baseline), or their experiment was flawed.
> > - In the second point, they proved something does not always work with a counter-example, which is valid
> > - In the third point, they proposed one possible explanation why GD+M is better than GD using an example. Sure one can argue that their explanation might only work on their specific example and might be unable to generalize. However, Jelassi & Li emphasized that their theory only works for small-margin data, and they empirically validated it with small-margin data in CIFAR-10. They were not claiming that their theory would work in a very general setting like "low SNR regime". Also in their conclusion they wrote "it would be interesting to understand whether this phenomenon is the only reason", that is they admitted that their explanation might not be unique or generalizable. In your case, you only did experiments on your specific example, so there is no guarantee that your theory can be generalized to anything other than the specific case you studied. And even with your new title, "low SNR regime" is still too general. Quantitative finance is a typical application in the "low SNR regime". Why would any folks in quantitative finance risk their money to test flipping labels during training?
> >
> > > we do not think that the claims in any of these works are more “sound and robust” than ours.
> >
> > I am not interested in comparing other works with your paper. I am specifically focusing on the flaws of your paper, and I am pointing out that your theory might not be generalizable to even a slightly more general setting. Sure you can argue that your theory requires $m$ to be sufficiently large. But what about other parameters? What if there are more than two patches and one of them is noise? What if $\sigma$ does not satisfy Assumption 3.1 (iv)? Have you done sensitivity analysis for these parameters? If your theory can be broken by changing any one of these parameters, then how robust could it be?
> >
> > On a side note, as of today, I don't think a paper getting accepted or getting 100 citations is a strong signal of the paper having high quality. It is well known that ICML has given its best paper award to wrong/trivial papers three years in a row, and many such wrong/trivial papers get lots of citations. The variance of review is extremely high today, and this is a good example of low SNR regime. As a reviewer, I can only make sure that my reviews provide strong signals.
> >
> > My purpose is not to reject your paper. I spent so much time reviewing your paper because I found the specific example you constructed quite interesting, and I feel that it has the potential of becoming a great paper. Like I suggested, you could do more in-depth analysis on label smoothing, which is at least being used in real applications. You could also do more sensitive analysis and provide empirical evidence that your theory is robust. At this juncture, I feel that this paper is making overclaims and thus quite misleading, and thus is below my bar of acceptance, which is the standard a few years ago but might be higher than the average bar today.

---

> ### Author Response · Authors · 2024-12-03
>
> We thank the reviewer for the response.
>
> > Jelassi & Li emphasized that their theory only works for small-margin data, and they empirically validated it with small-margin data in CIFAR-10. They were not claiming that their theory would work in a very general setting like "low SNR regime".
>
> The term “small margin” is defined in the theoretical binary classification setting in the signal-noise data model, parallel to our definition of “low SNR”. Similar to Jelassi & Li 2022, we have also emphasized that our theory applies to the particular single-noise model in the abstract and introduction.
> If the reviewer thinks that our title is too general and constitutes overclaiming, we do not see why the claims in Jelassi & Li 2022 are sound instead of misleading, as their title does not include the qualifier “small margin data”.
>
> Regarding the modified CIFAR-10 experiment in Jelassi & Li 2022, the margin (or SNR) of real-world data is not known a priori, and there is no explanation of why shuffling the RGB channels corresponds to modulating the margin.
> Nevertheless, we have conducted experiments on the CIFAR-10 dataset where a VGG-16 model is optimized by full-batch GD vs. label-noise GD (analogous to Figure 1 in Jelassi & Li 2022).  We trained the model on 10,000 samples, and to isolate the effect of label noise, we disabled data augmentation and batch normalization. The learning rate was set to 0.02, and the label flipping rate was 0.2.
> The results are available [here](https://postimg.cc/K341g2RK), where we observe ~2% improvement in the test accuracy due to label noise, even though it is not clear if this dataset has low SNR. We also note that our baseline for full-batch GD achieves higher accuracy than that reported in Jelassi & Li 2022. This indicate that, in real-world scenarios, label-noise GD can outperform standard GD, which is consistent with the findings in Haochen et al. 2021.
>
> We will provide additional updates on synthetic experiments where the SNR can be explicitly controlled.
>
> ----
>
> > None of the papers you provided in your new response empirically showed that flipping labels could be preferable in real applications.
>
> We do not argue that label noise is an indispensable component for state-of-the-art algorithms in practical datasets such as ImageNet – such SOTA performance is not achieved by standard SGD training (without preconditioning, data augmentation, etc.).
> This being said, it has been empirically observed in Haochen et al. 2021 that label flipping noise closes the gap between small- and large-batch training, and the benefit (due to implicit regularization) of such label noise has been studied in Damian et al. 2022.
> We believe that this provides sufficient justification to study label noise GD in the signal-noise data model and to characterize how this algorithmic regularization contributes to benign overfitting.
>
> ----
> Finally, while we appreciate the reviewer’s constructive feedback, we find it frustrating that a substantial part of our response has been consistently overlooked. We summarize these points as follows.
>
> 1. The reviewer stated in the original review that the primary concern was that “*the experimental results I present here establish the fact that the claim of this paper, that is "label noise GD is better than GD", is not very robust and not very sound*”. We have clarified  in our previous response that
>    - The experiment where GD achieves comparable generalization performance does not contradict our theoretical findings. Moreover, we have shown that by further decreasing the SNR in this narrow-width setting, the benefit of label noise is once again observed.
>    - The experiment where “*label noise GD is much worse than standard GD*” is due to large learning rate that leads to non-convergence.
> 2. The reviewer wrote in the followup comment that “*I suggest you do more sensitivity analysis and ablation studies*”. Note that in our first round of response, we have already provided such a learning rate sweep for GD in our signal-noise data model. We have also included additional experiments on modified MNIST and CIFAR where the SNR can be modulated (see Appendix E).
> 3. Regarding the reviewer’s concern on our theoretical setup, we have explained the role of different components of our model (width, noise patch, activation, etc.) and referred to earlier works where similar conditions were introduced.
> 4. We have clarified claims that the reviewer made on the literature that we find misleading in our second round of response, including
>    - The review claimed “*none of the cited papers suggested that flipping the labels during training is a good idea*”, when we explicitly mentioned in the main text that our label flipping procedure originated from one of the cited papers.
>    - The reviewer claimed that the conclusion of Jelassi and Li (2022) was "*momentum does not always lead to a higher generalization in deep learning*", while this is not the central contribution of the work.

---

### Official Review · Reviewer_tiYy · 2024-10-31

**Soundness:** 3
**Presentation:** 3
**Contribution:** 3
**Rating:** 6
**Confidence:** 4

**Summary:**

This paper analyzes a data model introduced in prior work on 2-layer convolutional network training dynamics, and shows that training such networks using label noise can allow for generalization even in low signal-to-noise (SNR) settings. The main contribution of this work is to show that previous negative results on noise memorization in low SNR settings can be avoided with very little additional computational overhead.

**Strengths:**

**Significance:** This paper addresses the important problem of learning in low SNR settings, and shows that introducing a small amount of label noise can mitigate noise memorization issues, which is an observation with potentially significant practical ramifications given the ease with which label noise can be introduced.

**Quality:** Overall the results and techniques in this paper are interesting, and the work seems sound as it operates in a now well-established theoretical setting and appropriately builds off/draws comparisons to the prior related work.

**Clarity:** The paper is well-written; the key ideas are introduced properly and the results are contextualized well.

**Originality:** Although the paper is working in an identical setting to prior work [1], the label noise analysis is new and uses techniques that did not appear in the previous results.

[1] Cao, Yuan et al. “Benign Overfitting in Two-layer Convolutional Neural Networks.” ArXiv abs/2202.06526 (2022).

**Weaknesses:**

Overall, I find the paper to be successful in what it sets out to do and thus favor acceptance; however, I do feel the paper could be stronger with a more practical verification of the introduced ideas. Additionally, some aspects related to clarity could be improved -- these are outlined below.

**Experiments:** The paper only contains synthetic experiments, which serve to verify the theory directly. However, several of the prior works in this direction (analyzing conv nets on multi-feature/multi-view data) have included experiments at least on image classification benchmarks to show that their theoretical observations still transfer (to an extent). For example, it seems to me that one could practically mimic Definition 2.1 by concatenating random noise to images from standard datasets (randomly to either the front or back).

**Clarity of Theory:** Theorem 3.1 has an analogous result in [1] (Theorem 4.4) and it would be helpful to compare the two more directly. It would also be useful to provide more motivation for using squared ReLU as opposed to other modifications -- for what pieces of the analysis is this necessary? Lastly, it would be nice to sketch some intuition for proving the supermartingale property in Lemma D.4; I looked at the proof but it's hard to get a sense from the calculations themselves.

**Questions:**

- My main question is regarding the choice of activation (as mentioned above) -- in prior work, using higher power polynomials of ReLU was important for making sure noise was suppressed while the signal was learned, and I'm curious how important the choice of using squared ReLU is here?

---

> ### Author Response · Authors · 2024-11-21
> **Responses (Part I) to Reviewer tiYy**
>
> We thank the reviewer for the thoughtful comment and constructive feedback. We address the technical concerns below.
>
> ----------------------------
>
> > W1: The paper only contains synthetic experiments, which serve to verify the theory directly. It seems to me that one could practically mimic Definition 2.1 by concatenating random noise to images from standard datasets (randomly to either the front or back).
>
> **R1**: We appreciate your suggestion regarding practical verification. In response, we have conducted an additional experiment using the MNIST dataset, where we concatenated random noise to each image (either at the front or back), in a manner similar to Definition 2.1. This setup aims to mimic the low SNR setting by introducing irrelevant noisy features while retaining the core signal in the original images.
>
> The noise level was set to $\sigma_p = 5$. Moreover, the original pixel values of the digits ranged from 0 to 255, and we chose a normalization factor of 80. In this setup, the added noise formed a "noise patch," and the digits formed a "signal patch." We focused on the digits '0' and '1', using $n = 100$ samples for training and 200 samples for testing. The learning rate was set to $\eta = 0.001$, and the width was set to $m=20$, with a label noise level of $p = 0.15$. The results, shown in Figure 4, were consistent with our theoretical conclusions, reinforcing the insights derived from our analysis. We show the corresponding result in **Appendix E.2**.
>
> ----------------------------
>
> > W2: Theorem 3.1 has an analogous result in [1] (Theorem 4.4) and it would be helpful to compare the two more directly. It would also be useful to provide more motivation for using squared ReLU as opposed to other modifications -- for what pieces of the analysis is this necessary? Lastly, it would be nice to sketch some intuition for proving the supermartingale property in Lemma D.4; I looked at the proof but it's hard to get a sense from the calculations themselves.
>
> **R2**: The main difference between our Theorem 3.1 and Theorem 4.4 from [1] lies in the condition under which harmful overfitting occurs for standard GD. In [1] (Theorem 4.4), the condition that leads to harmful overfitting is $n^{-1} SNR^{-q} = \tilde{\Omega}(1)$, where $q \ge 3$. In contrast, our work shows that even under a  different condition, we require  $n^{-1} SNR^{-2} = \tilde{\Omega}(1)$ harmful overfitting can still occur.
>
> The choice of squared ReLU activation is motivated by the fact that we do not optimize the second-layer parameters. We aim for the 2-homogeneous squared ReLU activation to mimic the behavior of training both layers simultaneously in a standard ReLU network. This higher-order homogeneity amplifies feature learning (e.g., see [Chizat & Bach, 2020](https://proceedings.mlr.press/v125/chizat20a/chizat20a.pdf); [Glasgow, 2023](https://openreview.net/pdf?id=HgOJlxzB16)) and helps create a more significant gap between signal learning and noise memorization, which is crucial for our analysis. A similar effect can be achieved by a smoothed ReLU activation with local polynomial growth as in [Allen-Zhu & Li (2020)](https://openreview.net/pdf?id=Uuf2q9TfXGA); [Shen et al. (2022)](https://proceedings.mlr.press/v162/shen22a/shen22a.pdf)
>
> The supermartingale property in Lemma D.4 is established to demonstrate how the introduced label noise keeps the noise memorization coefficient oscillating within a range of constant values. More specifically, we define:$ \iota^{(t)}\_i = \frac{1}{m} \sum\_{r=1}^m  \overline{\rho}^{(t)}\_{y\_i, r,i} $, where $\iota^{(t)}\_i$​ represents the noise memorization coefficient at iteration t. Under label noise GD, its evolution can be approximated as:
> $$
>     \iota^{(t+1)}\_{i}  \approx
>         (1 + \frac{\eta \sigma^2\_p d}{(1+\exp((\iota^{(t)}\_{i} )^2))nm}) \iota^{(t)}\_{i},   \text{ with prob } 1-p.
> $$
>
>
> $$
> \iota^{(t+1)}\_{i}  \approx     (1 - \frac{\eta \sigma^2\_p d}{(1+\exp(-(\iota^{(t)}\_{i} )^2))nm}) \iota\_{i},   \text{ with prob } p.
> $$
>
> From this equation, we observe that the probability of an increase ($1−p$) is greater than the probability of a decrease ($p$). On the other hand, the magnitude of an increase is smaller than that of a decrease. This implies that while there is a higher chance of an increase, each increase is relatively small; conversely, decreases are less frequent but larger in magnitude. This balancing effect results in the oscillating behavior of the noise memorization coefficients.
>
> We use a one-sided Azuma inequality based on the supermartingale property because the noise memorization coefficient originates from a small order value, and the inequality provides a quantitative tool to characterize the oscillatory nature of noise memorization. This helps demonstrate that the noise remains controlled.

---

> > ### Author Response · Authors · 2024-11-21
> > **Responses (Part II) to Reviewer tiYy**
> >
> > > Q1: My main question is regarding the choice of activation (as mentioned above) -- in prior work, using higher power polynomials of ReLU was important for making sure noise was suppressed while the signal was learned, and I'm curious how important the choice of using squared ReLU is here?
> >
> > **A1**: Thank you for your question regarding the choice of activation function. The squared ReLU activation is the simplest polynomial function that can effectively mimic the exponential growth in both signal learning and noise memorization. This amplification effect allows the gap between signal learning and noise memorization to widen as training progresses, making it easier to distinguish between the two.
> >
> > Higher-order polynomial activations, as used in prior works [Allen-Zhu & Li (2020)](https://openreview.net/pdf?id=Uuf2q9TfXGA); [Shen et al. (2022)](https://proceedings.mlr.press/v162/shen22a/shen22a.pdf); [Cao et al. 2022](https://proceedings.neurips.cc/paper_files/paper/2022/file/a12c999be280372b157294e72a4bbc8b-Supplemental-Conference.pdf)
> >  , serve a similar role in amplifying the useful signal while controlling noise. While the polynomial order differs, the underlying mechanism is consistent—using a polynomial activation helps amplify learning while mitigating the effects of noise. In our case, squared ReLU is sufficient for achieving this goal.

---

> > > ### Comment · Reviewer_tiYy · 2024-11-22
> > >
> > > Thank you for the clarifications and the added experiments. The modified MNIST plots are encouraging, but I have a few follow-up questions: how sensitive are the plots to the choice of noise parameters and normalization? The normalization factor of 80 seems pretty arbitrary to me so it would be nice to have some further justification for the setting of those experiments. Additionally, I understand the motivation for introducing noise into MNIST is because MNIST has a clearer "signal" than other image classification datasets, but given the simplicity of the modification do the same results appear in harder benchmarks like CIFAR-10/CIFAR-100?
> > >
> > > Also, regarding the choice of activation -- my apologies, I should have framed my question more precisely. If I recall correctly, the results of Cao et al. 2022 worked for $\max(x, 0)^q$ for any $q > 2$. Do your results hold for $q \ge 2$ or is the choice of squared ReLU specifically necessary?

---

> ### Author Response · Authors · 2024-11-24
> **Further Response to Reviewer tiYy**
>
> Thank you for your follow-up questions and for your interest in our work. We are glad that you found the modified MNIST experiments encouraging. Below, we address your concerns in detail.
>
> --------------
>
> > how sensitive are the plots to the choice of noise parameters and normalization
>
> **R**:  To assess the sensitivity of our methods to the choice of noise parameters and signal normalization, we conducted additional experiments on a modified MNIST dataset. Specifically, we varied the signal normalization values from 60 to 140 and the noise levels from 4 to 8. For each combination of noise level and signal normalization, we trained the neural network for 200,000 steps with a learning rate of $\eta = 0.001$, using either standard gradient descent (GD) or label noise GD.
>
> The results consistently show that label noise GD achieves higher test accuracy compared to standard GD across all tested configurations. This demonstrates the robustness of label noise GD to variations in both noise and signal normalization parameters, highlighting its effectiveness in handling different training conditions. We have updated **Appendix E.2** to include these results.
>
> --------------
>
> > Given the simplicity of the modification do the same results appear in harder benchmarks like CIFAR-10/CIFAR-100
>
> **R**: We have also extended our experiments to a subset of CIFAR-10, specifically using two classes: airplane and automobile. Gaussian noise was added to a portion of the images, following a similar setup as the MNIST experiments.
>
>
> The results indicate label noise GD consistently outperformed standard GD, demonstrating its robustness even in more challenging CIFAR-10 dataset. We believe that these extended experiments help establish the broader applicability of our findings to more complex benchmarks.
>
> We have updated **Appendix E.2** to include these results.
>
> --------------
>
> > Do your results hold for q≥2 or is the choice of squared ReLU specifically necessary
>
> **R**: Our theoretical results regarding label noise GD hold for $q=2$, which is the minimal value for which both the signal and noise exhibit exponential growth. This makes  a particularly interesting and challenging case to analyze, as it allows us to study the interaction between signal and noise in a setting that closely resembles practical two-layer ReLU networks.
>
> We conducted additional experiments with $q=3$ and $q=4$. In both cases, the experimental results consistently show that using a higher polynomial ReLU activation helps label noise GD suppress noise memorization while enhancing signal learning, ultimately leading to improved test accuracy compared to standard GD. These findings suggest that the benefits of label noise GD generalize well to higher-order polynomial ReLU activations. We believe our current theoretical analysis can be extended to higher higher-order polynomial ReLU activations.
>
> We have updated **Appendix E.4** to include the experimental results for $q=3$ and $q=4$.

---

> > ### Comment · Reviewer_tiYy · 2024-11-25
> >
> > Thank you for the further clarifications. I feel my original questions have been adequately answered, but I am following the discussion with Reviewer y9sU and will update if I have any follow-up questions.

---

> > > ### Author Response · Authors · 2024-12-03
> > >
> > > Thank you for your response and for letting us know that your original questions have been addressed. Please don’t hesitate to reach out if further clarification or additional details are needed!

---

### Official Review · Reviewer_TMo9 · 2024-11-03

**Soundness:** 3
**Presentation:** 3
**Contribution:** 3
**Rating:** 6
**Confidence:** 4

**Summary:**

The paper investigates the performance of a variant of Gradient Descent (GD), termed Label Noise Gradient Descent (LNGD), in scenarios with low signal-to-noise ratios.
LNGD introduces label flipping as a stochastic process before performing GD iterations, maintaining a computational cost comparable to standard GD.
The authors demonstrate a scenario where GD fails while LNGD is effective.
I find the paper to contain intriguing concepts, but some sections necessitate further elucidation.
With proper clarification of these points, I would recommend acceptance.

**Strengths:**

1. The authors establish a theoretical separation between the performance of GD and LNGD under specific signal-to-noise ratio (SNR) settings.
2. The proposed LNGD method does not introduce additional computational costs.
3. The paper is well-structured and presented clearly.
4. The synthetic experiments corroborate the theoretical findings well.

**Weaknesses:**

1. [Major Concern] Theorem 3.1 suggests that GD fails with a threshold of $t = \Theta(m^3 n /d)$, where $m$ is the number of neurons, while Theorem 3.2 indicates that LNGD succeeds with a threshold of $t = \Theta(m)$. This discrepancy raises concerns about the failure conditions and the comparative performance when both algorithms are evaluated within the same time frame. Additionally, it is unclear how varying the number of iterations for GD and LNGD affects their success or failure. To summarize, it requires:
- Clarify if the different time thresholds for GD and LNGD are directly comparable, and if not, explain why.
- Provide a more detailed comparison of how GD and LNGD perform when run for the same number of iterations.
- Analyze or discuss how the performance of both algorithms changes as the number of iterations varies.
2. How does LNGD perform under different forms of noise? I am concerned that LNGD might only be effective in specific noise regimes. The authors should provide empirical or theoretical evidence to address this concern.
3. For larger values of $m$, it appears that reaching the generalizable region as per Theorem 3.2 becomes slower. Could the authors offer insights into this phenomenon?

**Questions:**

See above

---

> ### Author Response · Authors · 2024-11-21
> **Responses to Reviewer TMo9**
>
> We thank the reviewer for the thoughtful comment and constructive feedback. We address the technical concerns below.
>
> ----------------------------
>
> > W1 Clarify if the different time thresholds for GD and LNGD are directly comparable, and if not, explain why; Provide a more detailed comparison of how GD and LNGD perform when run for the same number of iterations; Analyze or discuss how the performance of both algorithms changes as the number of iterations varies.
>
> **R1**: The time thresholds for standard GD and label noise GD are not directly comparable, as they correspond to different evaluation criteria. Specifically, the time threshold for standard GD represents the time required for the training loss to converge below a certain threshold $\epsilon$, whereas the time threshold for LNGD pertains to achieving sufficiently low 0-1 test loss. Therefore, comparing the running time for the two algorithms directly is not fair, as they are optimizing different objectives.
>
> However, to facilitate a meaningful comparison, we have derived a ratio between the time thresholds under Assumption 4.1. By setting $m^2 = \log(6/(\sigma_0 \\| \boldsymbol{\mu} \\|_2))/\epsilon$. we can obtain the ratio between the time thresholds as follows:
>
> $$\frac{n \\| \boldsymbol{\mu} \\|^2_2}{ \sigma^2_p d} =  n \mathrm{SNR}^2$$
>
> According to Assumption 4.1, we assume that $n  \mathrm{SNR}^2  \ll 1$, which implies that label noise GD requires more time to achieve good test performance compared to the convergence time of training loss for standard GD, which is consistent with the empirical results demonstrated in Figure 1.
>
> Regarding performance as the number of iterations varies:
> - For standard GD, the 0-1 test loss does not vary significantly with the number of iterations; the main focus is on training loss convergence, which requires sufficient iterations.
> - For LNGD, it is the opposite: the training loss does not vary significantly with the number of iterations, while the 0-1 test loss benefits from increased training iterations as the label noise regularizes and prevents overfitting.
> ----------------------------
>
> > W2 How does LNGD perform under different forms of noise? I am concerned that LNGD might only be effective in specific noise regimes. The authors should provide empirical or theoretical evidence to address this concern.
>
> **R2**. To validate the robustness of label noise GD under different noise forms, we varied $p$ across different values. For example, we show the results for $p = 0.3$ in Figure 5 and $p = 0.4$ in Figure 6. The results consistently indicate that label noise helps reduce overfitting and boost generalization, especially in low SNR settings.
>
> In addition, we extended our empirical analysis to include Gaussian noise and uniform distribution noise added to the labels. For Gaussian noise, we used two examples, namely $\epsilon^{(t)}_i \sim \mathcal{N}(1,1)$ and $\epsilon^{(t)}_i \sim \mathcal{N}(0.6,1)$, with the results shown in Figures 7 and 8, respectively. Furthermore, for the uniform distribution, we simulated the noise with $\epsilon^{(t)}_i \sim \mathrm{unif}[-1, 2]$ and $\epsilon^{(t)}_i \sim \mathrm{unif}[-2, 3]$. The results are shown in Figures 9 and 10, respectively.
>
> Our results indicate that label noise GD still performs effectively, achieving better generalization compared to standard GD, providing further evidence of the robustness of label noise GD under different noise forms. We update the corresponding results in **Appendix E.3**.
>
> ----------------------------
>
> > W3 For larger values of m, it appears that reaching the generalizable region as per Theorem 3.2 becomes slower. Could the authors offer insights into this phenomenon?
>
> **R3** The parameter $m$ represents the number of neurons in the hidden layer. In our analysis, we utilize $F_{j}= \frac{1}{m} \sum\_{r = 1}^m \sum\_{p = 1}^2 \sigma  ( \langle w\_{j,r}, x^{(p)} \rangle )$ for output function, and similarly, $\gamma^{(t+1)}\_{j,r} = \gamma\_{j,r}^{(t)} - \frac{\eta}{nm} \sum_{i=1}^n \ell_i \sigma'(\langle w_{j,r}^{(t)}, y_i \mu \rangle)  \| \mu\|^{2}_2 \epsilon_i^{(t)}$ for signal learning. Here $1/m$ is the scaling factor.
>
> As the network is trained, different neurons converge at different rates. The training process often exhibits an "exponential separation" effect, where the neuron with the fastest convergence rate can significantly outpace the others. Importantly, while the convergence rate of the fastest neuron remains relatively stable, the overall outcome is effectively averaged across all m neurons, resulting in an average impact of approximately $\max_r\gamma^{(t)}_{r}/m$.
>
> As  $m$ increases, the contribution of each individual neuron becomes smaller, which means that the network as a whole requires more time to achieve similar overall performance.

---

> > ### Author Response · Authors · 2024-12-03
> >
> > Thank you for your thoughtful review and feedback. As the discussion phase is nearing its conclusion, we would like to confirm if our responses have adequately addressed your concerns. If you have any remaining questions or additional feedback, please don’t hesitate to let us know—we would be happy to provide further clarification!

---

### Official Review · Reviewer_YCSq · 2024-11-04

**Soundness:** 3
**Presentation:** 3
**Contribution:** 3
**Rating:** 8
**Confidence:** 2

**Summary:**

The paper explores the effects of label noise gradient descent (GD) on improving generalization in deep learning models under low signal-to-noise ratio (SNR) conditions. The authors argue that neural networks trained with standard gradient descent tend to overfit noise in the low SNR regime, leading to poor generalization. In contrast, label noise GD, where random label noise is introduced during training, mitigates noise memorization and leads to improved generalization performance. The authors present theoretical analysis and empirical experiments to support their claims, demonstrating that label noise GD effectively enhances signal learning while controlling overfitting in noisy settings.

**Strengths:**

Theoretical Contributions: The authors offer rigorous theoretical analysis, demonstrating that label noise GD can effectively balance the trade-off between noise memorization and feature learning in challenging settings. The use of supermartingale arguments and concentration inequalities adds depth to the theoretical framework.

**Weaknesses:**

Practical Applications: While the theoretical analysis and synthetic experiments are thorough, adding experiments on more complicated simulations or real-world datasets would significantly strengthen the practical relevance of the findings.

**Questions:**

Data Generating Process Complexity: The current data generating process appears to be relatively simple, which may limit the generalizability of the results. I suggest exploring more complex data generation functions, such as staircase functions and parity functions, to test the robustness of the proposed method in capturing more intricate relationships.

---

> ### Author Response · Authors · 2024-11-21
> **Responses to Reviewer YCSq**
>
> We thank the reviewer for the thoughtful comment and constructive feedback. We address the technical concerns below.
>
> ----------------------------
>
> > W1: Practical Applications: While the theoretical analysis and synthetic experiments are thorough, adding experiments on more complicated simulations or real-world datasets would significantly strengthen the practical relevance of the findings.
>
> **R1** Thank you for your suggestion regarding the practical applications of our work. To address this, we have conducted two additional experiments to extend our findings to more complex simulations and real-world datasets.
> First, we conducted additional experiments using a 3-layer neural network with ReLU activation. Specifically, we fixed the last layer and trained the first two layers. The experimental results are consistent with our original findings: compared to standard GD, label noise GD boosts signal learning (as shown in the first plot) and achieves better generalization (as shown in the last plot). We have updated our manuscript to include this new result, with more details provided in **Appendix E.1**. We hope that this extension will address your concerns about the practical implications of our work.
>
> Second, We conducted an experiment using the MNIST dataset, in which Gaussian noise was added to the borders of the images while retaining the digits in the middle. The noise level was set to $\sigma_p = 5$. Moreover, the original pixel values of the digits ranged from 0 to 255, and we chose a normalization factor of 80. In this setup, the added noise formed a "noise patch," and the digits formed a "signal patch." We focused on the digits '0' and '1', using $n = 100$ samples for training and 200 samples for testing. The learning rate was set to $\eta = 0.001$, and the width was set to $m=20$, with a label noise level of $p = 0.15$. The results, shown in Figure 4, were consistent with our theoretical conclusions, reinforcing the insights derived from our analysis. We show the corresponding result in **Appendix E.2**.
>
> These additional experiments help demonstrate the practical applicability of our approach and provide further validation of our theoretical findings.
>
> ----------------------------
>
> > Q1: Data Generating Process Complexity: The current data generating process appears to be relatively simple, which may limit the generalizability of the results. I suggest exploring more complex data generation functions, such as staircase functions and parity functions, to test the robustness of the proposed method in capturing more intricate relationships.
>
> **A1**: Although we focus on simplified data setting, they successfully capture the primary factors necessary to quantitatively distinguish the training dynamics and generaliztion of both standard GD and label noise GD. Furthermore, we believe the data model setup is suitable for theoretical studies, given the non-linear, two layer structure and the non-convex nature of neural network optimization.
>
> We agree that exploring more complex data models, such as staircase functions and parity functions, may provide additional insights into the robustness of the proposed method in capturing more intricate relationships. This is indeed an interesting direction, and we plan to explore these more complex data models in future work.

---

> > ### Comment · Reviewer_YCSq · 2024-12-03
> >
> > I appreciate the authors’ response and the additional experiments presented in Appendix E. Based on this, I have increased my score by 1 and now recommend acceptance.

---

> > > ### Author Response · Authors · 2024-12-03
> > >
> > > Thank you for your updated feedback and for taking the time to review our additional experiments in Appendix E. We appreciate your thoughtful evaluation and are pleased to hear that the new insights contributed to your recommendation for acceptance.

---

### Official Review · Reviewer_M9Ej · 2024-11-07

**Soundness:** 3
**Presentation:** 2
**Contribution:** 2
**Rating:** 5
**Confidence:** 3

**Summary:**

The paper analyses generalisation error of gradient descent with and without label noise in a binary classification problem setting where the input is split into patches, with only one patch being dependent the class label and the other patches being random noise. The results mainly apply to the setting where the input norm in a noisy patch is much larger than the signal patch and where the number of training points is small relative to the dimension of an input patch. The model is a simple one hidden layer MLP, which is applied separately to all patches and added.


The main result of the paper shows that gradient descent with artificial label noise acts as a regulariser and generalises in low SNR setting while standard gradient descent fails.

**Strengths:**

The setting and the failure of gradient descent to generalise in such low SNR settings is well known, the viewpoint of label noise as regulariser is less studied however and the paper gives a detailed analysis and a reasonable proof sketch. It also demonstrates the result with a simple synthetic setup and demonstrates the message of the theorems. The analysis of the signal and noise coefficients with label noise is technically complex and intricate piece of work.

**Weaknesses:**

1. The theorem and results are quite believable, as they are also demonstrated with simple examples. But the setup of the results in the main paper seems to lack rigour. As evidenced in the haphazard way the order notation is used.

e.g Assumption 3.1 says n is Omega(1) but d is Omega(n^2). I am not sure how that is possible if Omega and O and Theta only hide absolute universal constants (like 2 and pi and not problem dependent constants like d,n, m etc). There are several other places where this issue pops up, like Theorem 3.2 has some constants C1 and C2 inside the Theta notation. As written now, the Theorems 3.1 and 3.2 sound vacuous, even though the underlying message is likely true. I would recommend replacing O, Theta and Omega with explicit absolute constants in the main paper, and give these constants in the appendix.


2. The two main results are Theorems 3.1 and 3.2. But Theorem 3.1 proves the same result as Cao et al.for q=2 while they prove for q>2. This is not a significant extra contribution.


3. The final weakness has to with the artificiality and extreme simplicity of the entire setup. While it is true that several papers have been published under similar setups (two layer networks with fixed last layer) I believe we need more realistic setups to make progress and for practitioners to actually take notice of the work done by theoreticians.

**Questions:**

I would appreciate a simplified version of Assumption 3.1 and Theorems 3.1 and 3.2 (even if it is less general) where several of the constants (like $\sigma_p, \mu, m, n $ ) are fixed and only the dimension $d$ is present as a symbol. The assumptions and theorem statement should be capable of being computed instead of just being given in order terms.

The label noise fraction seems to be a crucial parameter and is not given sufficient detail in the assumptions (it is bounded by some undefined C) and in the proofs and also in the synthetic experiments. Some details on this is needed.

---

> ### Author Response · Authors · 2024-11-21
> **Responses (Part I) to Reviewer M9Ej**
>
> We thank the reviewer for the thoughtful comment and constructive feedback. We address the technical concerns below.
>
> -----------------
>
> > W1 & Q1: Concern with the big-O notation.  Appreciate a simplified version of Assumption 3.1 and Theorems 3.1 and 3.2 (even if it is less general).
>
> **R1**: Thank you for the suggestion; we have revised the manuscript to enhance the clarity of our presentation. We believe that the big-O notations are appropriate for our purposes as they enhance clarity while maintaining mathematical rigor, as these notations are commonly used to concisely describe asymptotic relationships without explicitly stating all constants, which keeps the presentation clean and focused on key insights. This approach follows from recent works on gradient-based feature learning, such as [Cao et al. 2022](https://proceedings.neurips.cc/paper_files/paper/2022/file/a12c999be280372b157294e72a4bbc8b-Supplemental-Conference.pdf), [Jelassi et al. 2022](https://proceedings.mlr.press/v162/jelassi22a/jelassi22a.pdf), [Chen et al. 2023](https://openreview.net/pdf?id=3WAnGWLpSQ), [Frei et al. 2023](https://proceedings.mlr.press/v195/frei23a/frei23a.pdf), [Allen-Zhu et al. 2023](https://openreview.net/pdf?id=Uuf2q9TfXGA). We make the following clarifications.
>
> -----------------
> - *Simplified Version of Assumption 3.1*:
>
> To improve clarity, we provide a simplified version of Assumption 3.1 below:
>
> Given a failure probability $\delta \in (0,0.5)$,with a large enough universal constant $C$ we make the following assumptions on the parameters:
>
> - data dimension $d \ge C  \log(n/\delta) \max \\{ n^2, n \\| \boldsymbol{\mu} \\|^2_2/\sigma^2_p \\}$
>
> - network width $m \ge C\log(d/\delta)$
>
> - number of training samples $n \ge C\log(d/\delta) $.
>
> - learning rate $\eta \le 1/(C\log(d/\delta)) \sigma^{-2}_p d^{-1} $.
>
> - initialization variance $ C\log(d/\delta) n \sigma^{-1}_p d^{-3/4} \le \sigma_0 \le  1/(C\log(d/\delta)) \min \\{  \\| \boldsymbol{\mu}  \\|^{-1}_2 d^{-5/8}, \sigma^{-1}_p d^{-1/2} \\}   $.
>
> - flipping rate of label noise $0 < p < 1/C$
>
> -----------------
>
> - *Assumption 3.1 says n is Omega(1) but d is Omega(n^2). I am not sure how that is possible*
>
> We would like to clarify that this requirement does not imply a contradiction. Specifically, the relation can be understood as:
>  $$  d \ge  n^2 \log(n)  \ge  C \mathrm{poly}(\log(d)) $$
> Note that similar constraints also appeared in [Cao et al. 2022](https://proceedings.neurips.cc/paper_files/paper/2022/file/a12c999be280372b157294e72a4bbc8b-Supplemental-Conference.pdf),[Chen et al. 2023](https://openreview.net/pdf?id=3WAnGWLpSQ),[Kou et al. 2023](https://openreview.net/pdf?id=qmwtMuRh1j).
>
> -----------
>
> - *Theorem 3.2 has some constants C1 and C2 inside the Theta notation*
>
> Thank you for pointing this out. In the revision we have removed these constants from the Θ notation
>
> ------------
> - *Appreciate a simplified version of Assumption 3.1 and Theorems 3.1 and 3.2*
>
> To verify the consistency of the conditions in Assumption 3.1, we provide a concrete choice of hyperparameters:  $\sigma_0 =  d^{-11/16}, \eta<  d^{-1}, \\| \boldsymbol{\mu} \\|_2 = d^{-1/3}$,  $\sigma_p = C$, $n = d^{1/3}$, here $C$ is a constant. In this case, the constant does not need to be specified as d is sufficiently large.
>
> ------------
> - *like 2 and pi and not problem dependent constants like d,n, m etc*
>
> We would like to clarify that certain parameters, such as $m$ and $n$, cannot be treated as constants. In our analysis, $m$ and $n$ need to be at least polylogarithmic in the dimension $d$, as indicated by the use of the \tilde{} notation to hide polylogarithmic terms.
>
> --------------
>
> > W2: The two main results are Theorems 3.1 and 3.2. But Theorem 3.1 proves the same result as Cao et al. for q=2 while they prove for q>2. This is not a significant extra contribution.
>
> **R2**: First we note that the case where $q=2$ is not a straightforward extension of previous results for higher-order homogeneity. As remarked in Section 2, the choice of squared ReLU activation is motivated by the fact that we do not optimize the second-layer parameters. We aim for the 2-homogeneous squared ReLU activation to mimic the behavior of training both layers simultaneously in a standard ReLU network. Besides, $q=2$ is the smallest integer for which both the signal and noise exhibit exponential growth. This characteristic makes the analysis more challenging compared to $q>2$, where the gap between signal learning and noise memorization can be more easily characterized.
>
> In addition, the purpose of presenting Theorem 3.1 is also to enable a direct comparison label noise GD and standard GD. Therefore, the significance lies not in Theorem 3.1 itself as a standalone contribution, but rather in the comparison it allows. This comparison clearly demonstrates that incorporating label noise into gradient descent updates improves generalization in the low SNR regime, which is the core contribution of our work.

---

> ### Author Response · Authors · 2024-11-21
> **Responses (Part II) to Reviewer M9Ej**
>
> > W3: artificiality and extreme simplicity of the entire setup.
>
> **R3**: We agree that the problem setting (signal-noise data model, network architecture, etc.) has been studied in many prior works — as explained in the Introduction, this signal-noise model has been extensively studied in the neural network theory literature, as a sandbox to understand how gradient-based training prioritizes learning of the underlying signal instead of overfitting to noise. However, to the best of our knowledge, we are the first to study label noise GD in this setup and demonstrate efficient learning and benign overfitting in the challenging low SNR regime; moreover, the analysis of label noise GD is already rather nontrivial as discussed in Section 4.3. Extending this result to more complicated data generating process and architecture is an interesting future direction.
>
> In light of your feedback, we have also conducted additional experiments using a more complicated architecture of 3-layer neural network with ReLU activation. Specifically, we fixed the last layer and trained the first two layers. The experimental results are consistent with our original findings: compared to standard GD, label noise GD boosts signal learning (as shown in the first plot) and achieves better generalization (as shown in the last plot). We have updated our manuscript to include this new result, with more details provided in **Appendix E.1**.
>
> ----------------------------
>
> > Q2: The label noise fraction seems to be a crucial parameter and is not given sufficient detail in the assumptions (it is bounded by some undefined C) and in the proofs and also in the synthetic experiments. Some details on this are needed.
>
> **A2**: In our theoretical analysis, the label noise fraction ($1/C$) is used as an abstraction to maintain the correctness of asymptotic analysis. This approach is consistent with other recent works in the field, such as Condition 4.1 in [Kou et al. 2023](https://openreview.net/pdf?id=qmwtMuRh1j), Assumption (A3) in [Xu et al. 2023](https://openreview.net/pdf?id=BxHgpC6FNv), Condition 3.1 in [Chen et al. 2023](https://openreview.net/pdf?id=3WAnGWLpSQ). Specifically, there are two main sources that impose constraints on the label noise fraction:
>
> (1) Signal Learning Coefficient: To guarantee the lower bound of the signal learning coefficient, we need to impose an upper bound on the flipping rate ($p$). This is necessary for maintaining effective learning of the signal and corresponds to Lemma D.1 and the third bullet point in Lemma 4.3.
>
>  (2) Noise Memorization Coefficient: Similarly, to guarantee the lower bound of the noise memorization coefficient during the initial phase of training, we also need to limit the flipping rate ($p$). This constraint corresponds to Lemma D.3 and the first bullet point in Lemma 4.3.
>
> For the experimental section, we indeed used a fixed value of $p = 0.1$ for the label noise fraction. We have updated our manuscript to include this information explicitly, ensuring that the experimental details are clearly presented for reproducibility.

---

> > ### Comment · Reviewer_M9Ej · 2024-11-25
> > **Reply to rebuttal**
> >
> > Thanks for the response. Does the result of Cao et al. apply only to integer values of $q$? If it applies to real values of $q>2$ couldn't Theorem 2.1 potentially be derived via an approximation argument, like using ReLU^{2.0001} and arguing this is close to ReLU squared?
> >
> > I am not sure that the new experiments in Section E1 can be called a deeper neural net, as there is only one layer which has an activation. It is more like an amalgamation of the deep linear network with the signal noise model. I understand the authors intention to show that the message could be applied to other architectures, but I am not sure this section achieves anything.
> >
> > Overall, I agree that it is interesting to see how label flipping (or effectively label smoothing as each data point is seen T times with a p fraction of them being flipped labels) can help in regularisation. Can the authors add a paragraph/theorem/experiments on explicit label smoothing, as opposed to the simulated label smoothing that is implemented in the algorithm here? Would that also fail similar to standard SGD? Experiments with this baseline would also be appreciated.
> >
> > I will stick to my score as of now, but am following the points made by reviewer y9su (I agree with the spirit of the argument made by the reviewer) and the discussion below.

---

> ### Author Response · Authors · 2024-12-03
>
> We thank the reviewer for the further questions. Below, we address your concerns:
>
> --------------
>
> > Does the result of Cao et al. apply only to integer values of q? If it applies to real values of q>2couldn't Theorem 2.1 potentially be derived via an approximation argument, like using ReLU^{2.0001} and arguing this is close to ReLU squared?
>
> While it is theoretically possible to use an approximation argument with $\text{ReLU}^{2.0001}$ to analyze the setup, the resulting separation condition would differ significantly from that of $\text{ReLU}^2$. Specifically, with $\text{ReLU}^{2.0001}$, the separation condition would scale as $n \cdot \text{SNR}^{2.0001} $. The ratio $\frac{n\text{SNR}^{2.0001}}{n \text{SNR}^2} \rightarrow 0$ as $n \rightarrow \infty$, given $n \text{SNR}^2 \le 1/\log(d)$.
>
> More generally, if we consider $q = 2 + \epsilon$, letting $\epsilon \rightarrow 0$ introduces significant challenges. Specifically, in the condition of 4.2 from Cao et al. (2022), they require
> $\sigma_0 \le \tilde{O}(m^{-2/(q-2)}) n^{-[1/(q-2)] \lor 1} $, where $\sigma_0$ represents the initialization variance. As $\epsilon \rightarrow 0$, the term $m^{2/(q-2)}$ diverges, causing the constraint on $\sigma_0$​ to become impractical.
>
> -----
>
> > I am not sure that the new experiments in Section E1 can be called a deeper neural net, as there is only one layer which has an activation. It is more like an amalgamation of the deep linear network with the signal noise model. I understand the authors intention to show that the message could be applied to other architectures, but I am not sure this section achieves anything.
>
> There is a typo in the previous description: the first layer in the experiment is defined as $z^{(p)} = \sigma(W^\top x^{(p)})$, where $\sigma(\cdot)$ represents the ReLU activation function. In the experiment, this transformation is indeed applied alongwith ReLU activation. We have updated the manuscript to clarify this.
>
> -----
>
> >  Can the authors add a paragraph/theorem/experiments on explicit label smoothing, as opposed to the simulated label smoothing that is implemented in the algorithm here? Would that also fail similar to standard SGD? Experiments with this baseline would also be appreciated.
>
> In this work, we focus on the label flipping flipping which has been studied both empirically and theoretically in prior works. For instance, [Damian et al. 2021](https://proceedings.neurips.cc/paper/2021/file/e6af401c28c1790eaef7d55c92ab6ab6-Paper.pdf) where they use CIFAR-10 + ResNet  with flipping label noise algorithm. Besides, [Haochen et al. 2021](https://proceedings.mlr.press/v134/haochen21a/haochen21a.pdf) trained a VGG19 model on CIFAR100 with flipping noise, and found adopting flipping label noise achieves better test error compared to the baseline full-batch GD algorithm.
>
> We have also conducted experiments on the CIFAR-10 dataset where a VGG-16 model is optimized by full-batch GD vs. label-noise GD (analogous to Figure 1 in Jelassi & Li 2022).  We trained the model on 10,000 samples using both algorithms, and to isolate the effect of label noise, we disabled data augmentation and batch normalization. The learning rate was set to 0.02, and the label flipping rate was 0.2. The results are available [here](https://postimg.cc/K341g2RK), where we observe a ~2% improvement in the test accuracy due to label noise, even though it is not clear if this dataset exhibits low SNR.
>
> Lastly, we remark that label smoothing and label flipping is equivalent in expectation. In particular, label smoothing transform the original label $y_k$ to
> $ y’_k = y_k(1-\alpha) + \alpha/2 y_k +  \alpha/2 (-y_k)  =  (1-\alpha) y_k,$
> where $\alpha$ is the smoothing parameter. On the hand, taking expectation of label flipping, the expected label
> is $y’_k =   (1-p)y_k - p y_k = (1-2p) y_k.$
> Thus, when $\alpha=2p$, label smoothing is mathematically equivalent to label flipping in expectation; this connection has been discussed in [Li et al. 2020](​​https://proceedings.mlr.press/v108/li20e/li20e.pdf). However, we note that this equivalence in expectation does not imply closeness in the training dynamics due to the stochasticity introduced in the label flipping procedure.
>
> We will include the above clarifications in the revised manuscript.

---

### Meta-Review · Area_Chair_avbw · 2024-12-22

**Metareview:**

In this paper, the authors address the challenge of noise memorization in deep learning models, which can harm generalization, particularly in low signal-to-noise ratio (SNR) settings. They explore whether introducing label noise to gradient updates can improve test performance in such regimes. Focusing on a two-layer neural network (NN) trained with a label noise gradient descent (GD) algorithm in an idealized signal-noise data setting, they demonstrate that adding label noise during training suppresses noise memorization, allowing signal growth to dominate while controlling overfitting. This approach achieves better generalization despite low SNR. In contrast, the study establishes that standard GD is prone to overfitting under the same conditions, with a non-vanishing lower bound on test error, highlighting the advantage of label noise injection in gradient-based training.

The reviewers collectively identified the following strengths and weaknesses:

Strengths:

+Rigorous theoretical analysis showing label noise GD improves generalization by suppressing noise memorization in low SNR settings.
Clear, well-structured presentation with novel insights.

+Validation through synthetic experiments closely aligned with theoretical findings.

+Low computational overhead and potential for practical relevance.

+Highlights a counterintuitive and underexplored idea in regularization.

Weaknesses:

-Limited real-world experiments; mostly synthetic validation.

-Overgeneralized claims based on an idealized setup.

-Simplistic models may not reflect practical scenarios.

-Lack of comprehensive comparisons with related techniques like label smoothing.

-Insufficient exploration of performance under diverse noise types and conditions.

-Ambiguities in theoretical rigor and notation.

The authors rebuttal addressed a number of these concerns and some reviewers increased their score. However, some others such as overgeneralization, simplistic models, clarity of analysis, and insufficient practical validation seems to remain. See next box for more detail of the summary of the reviewers responses and reactions to the authros rebuttal.  Ultimately even during the discussion period the reviewers didn't agree on the merits of the paper with one reviewer strongly in favor and another strongly against. I think the paper has interesting ideas however in my opinion some of the claims still require further experimentation and additional points to be clarified. I therefore can not recommend acceptance at this time. I do encourage the authors to resubmit after a thorough revision.

**Additional Comments On Reviewer Discussion:**

Reviewer Positions Post-Rebuttal:

Reviewer y9sU (Reject):
Maintained concerns about the impracticality of the setup and lack of generalization to broader cases. Argued that flipping labels in low SNR settings is counterintuitive and unlikely to work in real-world scenarios.

Reviewer M9Ej (Marginal Reject):
Acknowledged improvements in the manuscript but remained skeptical about the theoretical rigor and broader contributions compared to prior work.

Reviewer tiYy and TMo9 (Marginal Accept):
Recognized the theoretical contributions and novel insights but agreed the paper would benefit from broader experiments and clearer comparisons to related methods.

Reviewer YCSq (Accept):
Increased their score after reviewing additional experiments, noting the counterintuitive nature of the results and potential to inspire future work.

---

### Decision · Program_Chairs · 2025-01-22

Reject